# ARID1A loss induces polymorphonuclear myeloid-derived suppressor cell chemotaxis and promotes prostate cancer progression

Ni Li [1,7] ✉, Qiuli Liu [2,7], Ying Han[1,7], Siyu Pei[1,7], Bisheng Cheng[3], Junyu Xu[4], Xiang Miao[1], Qiang Pan[1], Hanling Wang[1], Jiacheng Guo[1], Xuege Wang[1], Guoying Zhang[1], Yannan Lian[1], Wei Zhang[1], Yi Zang[1], Minjia Tan [4], Qintong Li [5], Xiaoming Wang [6], Yichuan Xiao [1], Guohong Hu [1], Jun Jiang [2] ✉, Hai Huang [3] ✉ & Jun Qin [1,2] ✉

Chronic inflammation and an immunosuppressive microenvironment promote prostate cancer (PCa) progression and diminish the response to immune checkpoint blockade (ICB) therapies. However, it remains unclear how and to what extent these two events are coordinated. Here, we show that ARID1A, a subunit of the SWI/SNF chromatin remodeling complex, functions downstream of inflammation-induced IKKβ activation to shape the immunosuppressive tumor microenvironment (TME). Prostate-specific deletion of *Arid1a* cooperates with *Pten* loss to accelerate prostate tumorigenesis. We identify polymorphonuclear myeloid-derived suppressor cells (PMN-MDSCs) as the major infiltrating immune cell type that causes immune evasion and reveal that neutralization of PMN-MDSCs restricts the progression of *Arid1a*-deficient tumors. Mechanistically, inflammatory cues activate IKKβ to phosphorylate ARID1A, leading to its degradation via β-TRCP. ARID1A downregulation in turn silences the enhancer of *A20* deubiquitinase, a critical negative regulator of NF-κB signaling, and thereby unleashes CXCR2 ligand-mediated MDSC chemotaxis. Importantly, our results support the therapeutic strategy of anti-NF-κB antibody or targeting CXCR2 combined with ICB for advanced PCa. Together, our findings highlight that the IKKβ/ARID1A/NF-κB feedback axis integrates inflammation and immunosuppression to promote PCa progression.

[1]CAS Key Laboratory of Tissue Microenvironment and Tumor, CAS Center for Excellence in Molecular Cell Science, Shanghai Institute of Nutrition and Health, Chinese Academy of Sciences, University of Chinese Academy of Sciences, 320 Yueyang Road, Shanghai 200031, China. [2]Department of Urology, Institute of Surgery Research, Daping Hospital, Army Medical University, Chongqing 400042, China. [3]Department of Urology, Guangdong Provincial Key Laboratory of Malignant Tumor Epigenetics and Gene Regulation, Guangdong Provincial Clinical Research Center for Urological Diseases, Sun Yat-sen Memorial Hospital, Sun Yat-sen University, Guangzhou 510120, China. [4]State Key Laboratory of Drug Research, Shanghai Institute of Materia Medica, Chinese Academy of Sciences, Shanghai 201203, China. [5]Department of Obstetrics, Gynecology and Pediatrics, West China Second University Hospital, Key Laboratory of Birth Defects and Related Diseases of Women and Children, Ministry of Education, Sichuan University, 20 Renmin South Road, Chengdu 610041, China. [6]Department of Immunology, Jiangsu Key Lab of Cancer Biomarkers, Prevention and Treatment, Collaborative Innovation Center for Personalized Cancer Medicine, Nanjing Medical University, 101 Longmian Ave, Nanjing 211166, China. [7]These authors contributed equally: Ni Li, Qiuli Liu, Ying Han, Siyu Pei. ✉e-mail: lini@sibs.ac.cn; jiangjun_64@163.com; huangh9@mail.sysu.edu.cn; qinjun@sibs.ac.cn

Prostate cancer (PCa) remains a leading cause of cancer-related death in men worldwide. Genetic and epigenetic dysregulations, including but not limited to alterations in AR, PTEN, TP53, and RB1, promote PCa progression and disease relapse[1]. In human PCa, ~30% of primary tumors and up to 70% of metastatic cancers exhibit loss of heterozygosity at the *PTEN* gene locus[2,3]. Loss of PTEN expression correlates with a higher Gleason score (GS), poorer prognosis, and higher metastatic incidence[3]. Mouse prostate-specific *Pten* deletion produces prostatic intraepithelial neoplasia (PIN) and only evolves into adenocarcinoma after a long latency[4], suggesting that a "second hit" is required for the metastatic transformation of *Pten*-null indolent tumors. In support of this finding, dual inactivation of *Pten* and *TrpS3*, *Smad4* or *Setd2* produces metastatic PCa[5–7]. Together, patient specimens and genetically engineered mouse models (GEMMs) reinforce the central role of PTEN in prostate tumorigenesis; therefore, the identification of therapeutic targets in *PTEN*-deficient tumors is of high translational value.

The interaction between immune, stromal, and cancer cells shapes the tumor microenvironment (TME) into tumor-permissive "soil" to potentiate PCa progression and relapse[3,8]. Tumor-infiltrating myeloid cells, and in particular myeloid-derived suppressor cells (MDSCs), are demonstrated to be a particularly abundant population in preclinical models and patients that endow tumor growth and, moreover, could account for the limited success of immunotherapeutic strategies[9,10]. The abundance of circulating MDSCs correlates with PSA levels and metastasis in patients[11]. High infiltration of MDSCs causes immune evasion, as these cells suppress the functions of T and natural killer (NK) cells[12,13]. Moreover, MDSCs protect proliferating tumor cells from senescence[9] and drive the development of castration-resistant prostate cancer (CRPC) by secreting interleukin 23 (IL-23)[14]. Thus, targeting MDSCs or neutralizing MDSC recruiter cytokines restricts tumor progression in several PCa models[14–17]. Compared to WT or *Pten* deleted lesions, *Pten/TrpS3* double KO (DKO) mouse tumors exhibit the high enrichment of MDSCs, which promote PCa progression and castration resistance via suppression of anti-tumor immunity and secretions of cytokine or growth-related factors[18]. Importantly, primary and metastatic castration-resistant PCa (CRPC) showed synergistic responses when immune checkpoint blockade (ICB) was combined with MDSC-targeted therapy[14,16–19]. Gene alterations in cancer cells (e.g., YAP, CHD1 and ZBTB7α) are implicated in the localized expansions of MDSCs[16,17,19]. Nevertheless, the signaling circuits orchestrating the immunosuppressive TME remain largely unclear.

Epigenetic regulation modulates cell plasticity to instruct PCa metastasis[20]. The SWI/SNF complexes reposition nucleosomes and bind to DNA regulatory regions to govern transcription in an ATP-dependent manner[20]. SWI/SNF are macromolecular complexes comprising 12–15 subunits: a catalytic ATPase subunit, SMARCA4/BRG1 or SMARCA2/BRM, and several core subunits[20,21]. AT-rich interactive domain-containing proteins 1A and 1B (ARID1A and ARID1B) are mutually exclusive subunits and potentiate SWI/SNF complex activity via recruitment of the ATPase catalytic subunit[22]. Many components of SWI/SNF complexes are altered in different malignancies, adding up to ~20% of cancers bearing mutations in SWI/SNF genes[23,24]. ARID1A is the most frequently mutated chromatin remodeling gene and is present in many cancers, including but not limited to ovarian clear cell carcinomas, endometrioid carcinomas, bladder cancer and colorectal cancer[25]. The majority of ARID1A mutations are frameshift or nonsense mutations that occur throughout the coding region[24,26], suggesting that ARID1A functions as a tumor suppressor. Although SWI/SNF function is described at both promoters and enhancers, studies show that ARID1A targets SWI/SNF complexes to enhancers, where they coordinate with transcription factors to enable chromatin remodeling and gene activation[27,28]. ARID1A is less frequently mutated in PCa specimens (~2% in The Cancer Genome Atlas (TCGA) dataset).

However, the results showing that patients bearing low ARID1A expression had adverse disease outcomes prompted us to investigate its functions in PCa (Fig. 1b).

Here, we show that deletion of *Arid1a* in *Pten*-null PCa models promotes tumor progression in conjunction with dramatic remodeling of the TME toward a protumor immune profile. Our studies highlight that a combination of immune checkpoint blockade (ICB) and inhibition of NF-κB signaling show synergistic efficacy against advanced PCa with ICB resistance.

## Results

### Arid1a deletion cooperates with Pten loss to promote PCa progression

To assess ARID1A in PCa, we first compared ARID1A expression in prostate lysates derived from WT mice or mice harboring *Probasin*^Cre/+-mediated deletion of *Pten* (*Pten*^PC–/–) or *Pten/Trp53* (*Pten*^PC–/–; *Trp53*^PC–/–), representing indolent tumors or metastatic PCa, respectively. ARID1A levels were highest in WT prostates, and its expression was negatively correlated with tumor malignancy, with higher expression in *Pten*^PC–/– mice than in *Pten*^PC–/–; *Trp53*^PC–/– mice (Fig. 1a). Interestingly, the ARID1A level and NF-κB signaling (reflected by p-IKKβ versus total IKKβ and p-P65 levels), a major inflammatory insult that promotes PCa progression and therapeutic resistance, were inversely correlated (Fig. 1a). Encouraged by these observations, we performed immunohistochemistry (IHC) by using pre-evaluated anti-ARID1A antibodies in a human PCa tissue microarray composed of 140 samples (Supplementary Table 1). Examination of prostate specimens showed lower ARID1A expression in tumors than in adjacent normal tissues (Supplementary Fig. 1a). ARID1A immunostaining intensity was negatively associated with the GS and PSA level in prostate tumors (Supplementary Fig. 1b, c). More importantly, patients with low ARID1A expression exhibited a higher risk of biochemical recurrence (BCR) than those with high ARID1A levels (Fig. 1b). These results suggest a tumor suppressor role of ARID1A in PCa.

To explore *Arid1a* function, *Arid1a*-floxed mice were crossed with *Probasin*^Cre/+ mice to generate mice with deletion of *Arid1a* in the prostate epithelium (hereafter referred to as *Arid1a*^PC–/–; Supplementary Fig. 1d). *Arid1a*^PC–/– mice were healthy without overt defects in growth and reproduction. During 12 months of follow-up, 7 out of 10 *Arid1a*^PC–/– mice exhibited normal prostatic histology in a manner similar to that of *Arid1a*-floxed mice. The remaining mice (3 out of 10) developed hyperplasia or multifocal low-grade PIN (LGPIN) exhibiting a cribriform-like structure reminiscent of human PIN (Supplementary Fig. 1e). Nevertheless, the development of adenocarcinoma was not detected, indicating that *Arid1a* loss alone is insufficient to produce PCa, so cooperation with other oncogenic insults might be necessary.

Given that *PTEN* is frequently mutated in PCa, we crossed *Arid1a*^PC–/– mice with *Pten*-floxed mice to generate *Pten*^PC–/–; *Arid1a*^PC–/– mice. All *Pten*^PC–/–; *Arid1a*^PC–/– mice died by 22 weeks of age largely due to bladder outlet obstruction, likely causing hydronephrosis and renal failure (Fig. 1c). Quantitation of 3- and 4-month-old prostatic volumes suggested that *Arid1a* deletion expedited *Pten*-loss induced PCa progression (Fig. 1d). Likewise, magnetic resonance imaging (MRI) analysis quantified an ~3–5-fold increase in average prostatic tumor volume in 16-week-old *Pten*^PC–/–; *Arid1a*^PC–/– mice compared to that in *Pten*^PC–/– mice (Supplementary Fig. 1f). *Pten*^PC–/– mice developed localized high-grade PIN, and most survived beyond 1 year of age. In contrast, all 12-week-old *Pten*^PC–/–; *Arid1a*^PC–/– mice exhibited focally invasive PCa with a prominent stromal reaction (Fig. 1e and Supplementary Fig. 1g). The tumor malignance progressed over time, showing that the percent of adenocarcinoma further increased in 4-month-old *Arid1a*-deleted prostates relative to that in 3-month-old mice (Fig. 1e, f). Compared to the indolent lesions developed in *Pten*^PC–/– mice, *Arid1a*-deficient tumors were endowed with proliferative advantages, as evidenced by a

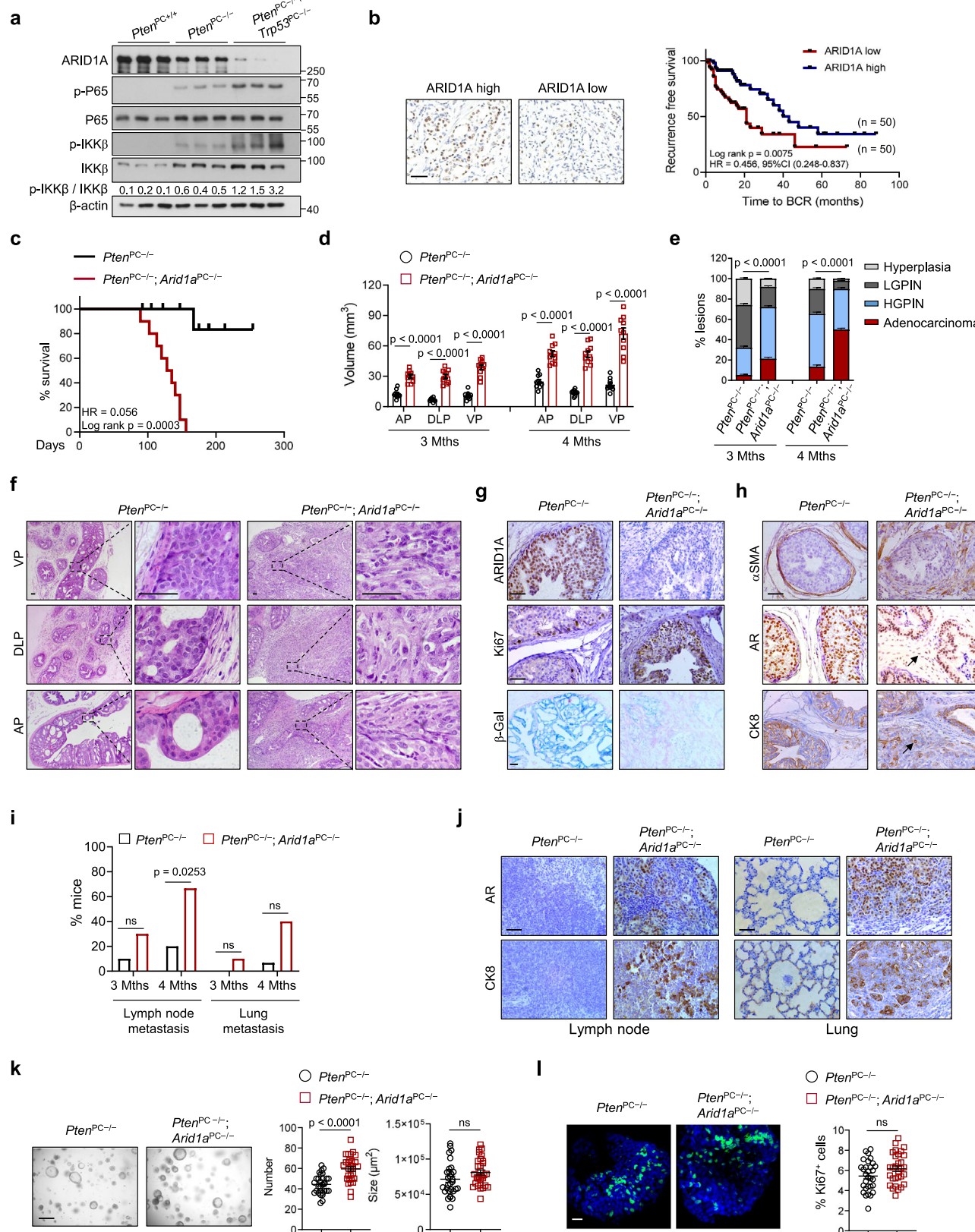

higher Ki67+ staining index and a decrease in oncogene-induced senescence-associated β-galactosidase staining (Fig. 1g). The smooth muscle sheaths surrounding the tumors (indicated by SMAα staining) in 4-month-old Pten^PC−/−; Arid1a^PC−/− mouse prostates were either discontinuous or disappeared in subset prostatic areas (Fig. 1h). As such, tumor cells with a luminal origin (AR+ and CK8+ cells, arrow) were

found to invade the stromal compartment (Fig. 1h), supporting that focally invasive cancer develops in Arid1a-deficient prostates. In contrast to limited metastatic incidences observed in 3-month-old Pten/Arid1a DKO mice (Fig. 1i and Supplementary Fig. 1h), metastatic spreading of tumor cells to distant organs, including the lymph nodes and lungs, was found with high penetrance in Pten^PC−/−; Arid1a^PC−/− mice

**Fig. 1 | ARID1A is clinically and functionally important for PCa progression. a** IB analysis of the indicated proteins in 3-month-old mouse prostate tissues with the indicated genotypes ($n = 3$). **b** Representative IHC staining and Kaplan–Meier plot of recurrence based on ARID1A expression in the TMA with PCa ($n = 100$). HR, hazard ratio. Scale bar, 50 μm. **c** Kaplan–Meier survival plots of $Pten^{PC-/-}$ and $Pten^{PC-/-}$; $Arid1a^{PC-/-}$ mice ($n = 10$ for each group). **d** Quantitation of 3- or 4-month-old prostatic volumes of indicated mice ($n = 10$). **e** Quantification of histological grade at the indicated time points ($n = 15$). **f** H&E-stained sections in the anterior (AP), dorsolateral (DLP) and ventral (VP) prostates of 16-week-old $Pten^{PC-/-}$ and $Pten^{PC-/-}$; $Arid1a^{PC-/-}$ prostates ($n = 15$, representative data are shown). Scale bars, 50 μm. **g** Immunohistochemical analysis of ARID1A and Ki67 expression and β-Gal staining in prostate sections. Scale bar, 50 μm. **h** IHC for SMAα, AR and CK8 in prostate sections. Scale bars, 50 μm. **i** Quantification of the metastatic incidence as indicated ($n = 10$ for 3 months, $n = 15$ for 4 months). **j** AR and CK8 staining of lymph nodes and lungs from 16-week-old mice. Scale bar, 50 μm. **k** Representative organoid images derived from 4-month-old $Pten^{PC-/-}$ and $Pten^{PC-/-}$; $Arid1a^{PC-/-}$ mice and quantification ($n = 10$ fields from three mice per group). Scale bar, 500 μm. **l** Immunofluorescence staining and quantification of the organoids with Ki67+ cells ($n = 10$ fields from three mice per group). Scale bar, 100 μm. **d, e, k, l** Data represent the mean ± SEM. Statistical significance was determined by the log-rank test (**b, c**), two-tailed unpaired $t$-test (**d, k, l**), $\chi^2$-test (**e**) and Fisher's exact test (**i**). **g, h** Experiments were repeated at least three times independently with similar results; data from one representative experiment are shown. Source data are provided as a Source Data file. ns, no significance.

at 4 months of ages (Fig. 1i, j). At this time points, $Pten^{PC-/-}$; $Arid1a^{PC-/-}$ mice showed lymph node metastasis in 10 of 15 cases and lung metastasis in 6 of 15 cases at 16 weeks of age (Fig. 1i). This observation contrasted with the results for $Pten^{PC-/-}$ mice, which exhibited a minimal metastatic incidence. Notably, previous studies indicate that specialized SWI/SNF complexes are associated with neuroendocrine or small cell PCa and may play a role in therapy resistance[29]. Nevertheless, we found that $Arid1a$ loss did not alter the markers of neuroendocrine lineage ($Chga$, $Eno2$ and $Syp$), basal ($Trp63$, $Ck5$) and luminal cells ($Ck8$ and $Ck18$) (Supplementary Fig. 1i). In addition, neuroendocrine lineage regulators $Ezh2$, $Sox2$, $Ascl1$, and androgen signal reflected by $Ar$ as well as its target, $Tmprss2$ remained unaffected upon $Arid1a$ loss (Supplementary Fig. 1i). Together, $Arid1a$ inactivation with $Pten$ deficiency expedites PCa progression.

We further carried out an organoid assay to determine whether $Arid1a$ functions in an autonomous manner. Luminal cells from 4-month-old $Pten^{PC-/-}$ mice and $Pten^{PC-/-}$; $Arid1a^{PC-/-}$ mice were isolated. At this time point, $Pten^{PC-/-}$; $Arid1a^{PC-/-}$ tumors already advanced to invasive PCa. In contrast to the phenotype observed in mice, $Arid1a$ silencing only slightly enhanced the capacity of tumor cells to form organoids without significant alteration in proliferation (Fig. 1k, l), suggesting that TME-mediated mechanisms account for enhanced tumor progression in vivo.

## $Arid1a$ loss produces an immunosuppressive TME in $Pten$-deficient PCa

To understand how $Arid1a$ ablation promotes prostate tumorigenesis, we conducted gene expression analysis using prostate epithelium (EpCAM+; CD45− cells) collected from 3-month-old $Pten^{PC-/-}$ and $Pten^{PC-/-}$; $Arid1a^{PC-/-}$ mice. We chose the earlier time point to characterize expression changes; therefore, the genes altered might be more directly related to the regulatory mechanism influenced by $Arid1a$ loss rather than secondary to malignance differences. Gene set enrichment analysis (GSEA) revealed significant enrichment of inflammatory pathways, including NF-κB and IL-6/STAT3 signaling, suggesting possible immune dysregulation in the absence of $Arid1a$ (Fig. 2a). To assess the spectrum of infiltrating immune cells in tumors, we performed mass cytometry (CyTOF) immunophenotyping to catalog immune cells in tumors from 12-week-old $Pten^{PC-/-}$ and $Pten^{PC-/-}$; $Arid1a^{PC-/-}$ mice (Fig. 2b). CD45+ infiltrating leukocytes were significantly increased in $Pten^{PC-/-}$; $Arid1a^{PC-/-}$ tumors compared to $Pten^{PC-/-}$ tumors (Supplementary Fig. 2a). We employed a 9-marker antibody panel to specifically identify 7 immune cell types and viSNE, a visualization tool for high-dimensional single-cell data based on the t-SNE algorithm; the results revealed that polymorphonuclear MDSCs (CD45+ CD11b+ F4/80low Ly6G+ Ly6Clow) were the most enriched immune cells in prostates from $Pten^{PC-/-}$; $Arid1a^{PC-/-}$ mice compared to $Pten^{PC-/-}$ mice (Fig. 2b). In line with the notion that PMN-MDSCs cells suppress T-cell proliferation, $Arid1a$ deletion led to reduced total and IFNγ+ CD8+ T cells, while other tumor-infiltrating leukocytes, including CD4+ T cells and macrophages, did not differ significantly (Fig. 2b).

Fluorescence-activated cell sorting (FACS) analysis verified the increased PMN-MDSCs and Tregs and decreased total and IFNγ+ CD8+ T cells, but not CD4+ T cells and macrophages in $Arid1a$-deleted tumors (Fig. 2c and Supplementary Fig. 2b). In mice, both neutrophils and PMN-MDSCs are characterized as CD11b+ Ly6G+ Ly6Clow. Thus, we compared gene expressions of Ly6G+ cells in peripheral blood of WT mice and $Pten^{PC-/-}$; $Arid1a^{PC-/-}$ tumors. We showed that Ly6G+ cells in tumors exhibited the high level of immunosuppressive genes such as $Nos2$, $Arg1$, $Nox2$, $S100a9$ manifested the features of PMN-MDSCs, whereas the Ly6G+ cells in peripheral blood predominantly expressed the pro-inflammatory genes including $Tnf$, $Il6$, $Cxcl4$ and among others (Supplementary Fig. 2c). In addition, standard T cell proliferation co-culture assay revealed that Ly6G+ cells from tumors suppressed CD3 and CD28 antibody-induced T cell proliferation and activation, establishing that they are mainly PMN-MDSCs rather than neutrophils (Supplementary Fig. 2d, e). Immunoprofiling data were confirmed by IHC staining of PMN-MDSC and CD8+ T-cell markers (Fig. 2d). To extend these observations, we employed synergistic models including Myc-CaP and $Pten$; $Trp53$; $Smad4$ triple KO (TKO) cells derived from a c-Myc transgenic mouse or $Probasin^{Cre/+}$; $Pten^{PC-/-}$; $Trp53^{flox/flox}$; $Smad4^{flox/flox}$ mouse with PCa. Organoid assays indicated that $Arid1a$ loss did not boost organoid formation, and its loss even inhibited the growth of Myc-CaP cells (Supplementary Fig. 2f). Paradoxically, $Arid1a$ ablation exhibited the inhibitory effects when tumor cells were engrafted in $Rag1$ null mice (Fig. 2e), suggesting that possible tumor or innate immune cell mediated mechanisms account for a negative role of ARID1A loss on tumor growth. In any event, $Arid1a$-deficient cells inoculated into immunocompetent FVB or C57/BL6 mice produced significantly larger tumors than $Arid1a$-intact cells (Fig. 2e), supporting that the effect of ARID1A loss to promote PCa progression depends on T-cell functionality.

To address whether PMN-MDSC enrichment indeed mediates $Arid1a$ loss effects, we adopted a well-characterized anti-Ly6G neutralizing monoclonal antibody (clone 1A8) to deplete PMN-MDSCs in mice bearing WT or $Arid1a$ KO xenografts. Based on tumor volume, PMN-MDSC neutralization reduced the growth of $Arid1a$ KO cells to a level comparable to that of WT cells (Supplementary Fig. 2g). PMN-MDSC depletion restored the number of total and IFNγ+ CD8+ T cells in $Arid1a$-depleted tumors to a level similar to that in WT tumors (Supplementary Fig. 2h). At 10 weeks of age, $Pten^{PC-/-}$; $Arid1a^{PC-/-}$ tumors started to progress to the early invasive carcinoma stage. At this timepoint, $Pten^{PC-/-}$; $Arid1a^{PC-/-}$ mice received a 4-week treatment of anti-Ly6G antibody. PMN-MDSC depletion caused weight reductions in $Arid1a$-deleted tumors (Supplementary Fig. 2i). Invasive adenocarcinoma developed in IgG-treated control mice, while anti-Ly6G antibody treatment arrested tumor progression (Fig. 2f). According to staining for Ki67, SMAα, CD8 and Ly6G and FACS analysis, tumor remnants in anti-Ly6G-treated $Pten^{PC-/-}$; $Arid1a^{PC-/-}$ mice exhibited reduced proliferation and stromal reactions, accompanied by increased total and IFNγ+ CD8+ T cells (Fig. 2g and Supplementary Fig. 2j, k). Thus, we conclude that PMN-MDSC enrichment is responsible for the acceleration of prostate tumorigenesis in $Pten^{PC-/-}$; $Arid1a^{PC-/-}$ mice.

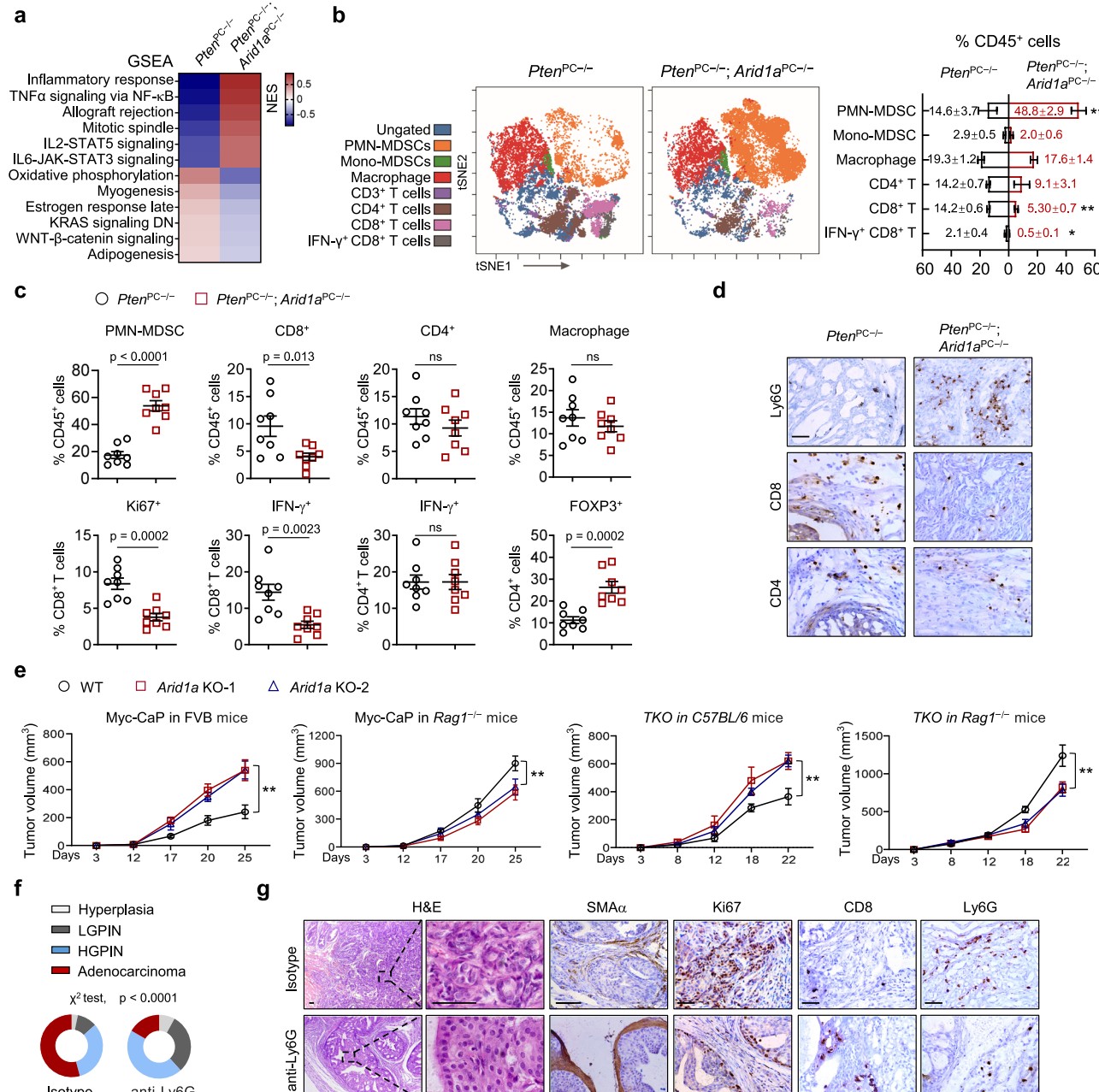

**Fig. 2 | Arid1a loss produces an immunosuppressive TME via the induction of PMN-MDSC recruitment. a** GSEA revealed the top 6 upregulated and down-regulated hallmark pathways in the prostate epithelium from 3-month-old $Pten^{PC-/-}$ and $Pten^{PC-/-}$; $Arid1a^{PC-/-}$ prostate tumors. **b** Immunoprofiling and quantification of 3-month-old $Pten^{PC-/-}$ and $Pten^{PC-/-}$; $Arid1a^{PC-/-}$ prostate tumors by CyTOF ($n = 3$). Exact $p$-values between the indicated tumors for PMN-MDSCs ($p = 0.002$), CD8+ T cells ($p = 0.0006$), IFN-γ+ CD8+ T cells ($p = 0.0188$). **c** Quantification of the indicated tumor-infiltrating immune cell populations by FACS analysis ($n = 8$). **d** IHC analysis for CD4+, CD8+, and PMN-MDSCs (Ly6G). Scale bars, 50 μm. **e** Tumor volume of WT and $Arid1a$ KO cells subcutaneously injected into immunocompetent mice (FVB or C57BL/6) or $Rag1^{-/-}$ mice. For Myc-CaP in FVB mice, WT or $Arid1a$ KO-1 ($n = 6$, each group), $Arid1a$ KO-2 ($n = 5$), WT vs $Arid1a$ KO-1 or KO-2 ($p < 0.0001$); for Myc-CaP in $Rag1^{-/-}$ mice, $n = 5$ for each group, WT vs $Arid1a$ KO-1 ($p < 0.0001$), WT vs

$Arid1a$ KO-2 ($p = 0.0007$); for $TKO$ in $C57BL/6$ mice, WT or $Arid1a$ KO-1 ($n = 6$, each group), $Arid1a$ KO-2 ($n = 7$), WT vs $Arid1a$ KO-1 ($p = 0.0004$), WT vs $Arid1a$ KO-2 ($p = 0.0002$); for $TKO$ in $Rag1^{-/-}$ mice, WT or $Arid1a$ KO-2 ($n = 7$, each group), $Arid1a$ KO-1 ($n = 6$), WT vs $Arid1a$ KO-1 or KO-2 ($p < 0.0001$). **f** Quantitation of the tumor histopathology in isotype- or Ly6G antibody-treated $Pten^{PC-/-}$; $Arid1a^{PC-/-}$ prostates starting when mice were 10-week-old and treatment was administered for 4 weeks ($n = 6$). **g** H&E and IHC staining for SMAα, Ki67, CD8 and Ly6G in mouse prostates with or without anti-Ly6G antibody treatment. Scale bars, 50 μm. **b, c, e,** Data represent the mean ± SEM. Statistical significance was determined by two-tailed unpaired $t$-test (**b, c**), two-way ANOVA followed by multiple comparisons (**e**) and $\chi^2$-test (**f**). **b, d, g** Representative data of triplicate experiments are shown. Source data are provided as a Source Data file. *$p < 0.05$, **$p < 0.01$, ns, no significance.

---

Castration drives the infiltration of PMN-MDSCs into the TME[30], which in turn confers anti-androgen resistance via secretion of inflammatory cytokines, such as IL-23 or inhibition of anti-tumor immunity[14]. Thus, we asked whether *Arid1a* deleted tumors are resistant to androgen deprivation therapy (ADT). To this end, 10-week-old

$Pten^{PC-/-}$; $Arid1a^{PC-/-}$ mice received castration followed by additional 2 months of enzalutamide (ENZ) treatment to mimic ADT (Supplementary Fig. 2l). Reflected by changes in tumor volumes and histology (Supplementary Fig. 2m-o), we found that *Arid1a* deleted tumors were largely refractory to ADT as compared to *Pten*-deleted lesions. Similar

as sham operated mice, ADT-treated *Arid1a* KO tumors also enriched in PMN-MDSCs and paralleled a significant reduction of CD8[+] T cells, whereas the expression of AR remained comparable between *Arid1a* WT and KO mice (Supplementary Fig. 2p). Together, our results indicate *Pten/Arid1a* DKO tumors are castration resistant.

### *Arid1a* loss enhances PMN-MDSC expansion through the activation of NF-κB signaling

To dissect the molecular mechanism by which *Arid1a* loss enhances PMN-MDSC expansion, we referenced our transcriptomic profiling data, and found a number of cytokine and chemokine transcripts to be significantly upregulated in *Arid1a* deleted tumors, including *Cxcl2*, *Cxcl3*, *Il23*, *Il1a*, *Ccl6* and among others (Fig. 3a). Differentially expressed genes were validated by RT-qPCR analysis. We found that the expression of NF-κB target genes, such as *Cxcl2*, *Cxcl3*, *Ccl2* and *Ccl3*, was upregulated in *Arid1a*-deleted tumors (Supplementary Fig. 3a). Analysis of 12-week-old *Pten*[PC−/−] and *Pten*[PC−/−]; *Arid1a*[PC−/−] mouse prostates indicated that *Arid1a* loss led to a significant increase in the levels of p-P65, suggesting that *Arid1a* deletion stimulates NF-κB activity (Fig. 3b). To determine whether this regulation is cell autonomous rather than based on the difference in tumor malignancy between *Pten*[PC−/−] and *Pten*[PC−/−]; *Arid1a*[PC−/−] mice, we silenced *ARID1A* in human and mouse PCa cells and verified that *ARID1A* depletion sensitized PCa cells to TNF-α stimulation, as evidenced by the increases in p-IKKα/β and p-P65 levels and the nuclear accumulation of p65 and p50 (Fig. 3c and Supplementary Fig. 3b). Conversely, ARID1A over-expression attenuated TNF-α-induced NF-κB activation (Fig. 3d). In contrast, IL-6-STAT3 or IFN-STAT1 signal activities did not differ significantly between WT and *Arid1a*-deleted cells (Supplementary Fig. 3c–e). *Arid1a*-depleted cells were more sensitive to TNF-α-induced NF-κB activation, which induced higher expression of target genes such as *Cxcl2*, *Cxcl3* in C4-2 and Myc-CaP cells (Fig. 3e and Supplementary Fig. 3f). Furthermore, GSEA analysis in Myc-CaP cells supported that *Arid1a*-deleted cells significantly enriched the gene signature linked to NF-κB signaling (Supplementary Fig. 3g). Recent studies have revealed that ARID1A loss impairs the expression of IFN signaling components, especially Th1-type chemokines (CXCL9 and CXCL10), to compromise effector T-cell tumor trafficking and anti-tumor immunity[31]. However, we found that ARID1A loss did not reduce *Cxcl9* and *Cxcl10* expression in PCa cells treated with IFN-γ or TNF-α and mouse prostate tumors (Supplementary Fig. 3f, h, i).

NF-κB-mediated chemokine secretion bridges tumor cells and cells in the TME, and CXCR2 ligands (CXCL2 and CXCL3) enhance MDSC chemotaxis in advanced PCa[9,16,18]. Quantified by ELISA, we detected that CXCL2 and CXCL3 protein levels were elevated in peripheral blood and prostate tumors of *Pten*[PC−/−]; *Arid1a*[PC−/−] mice relative to that in *Pten*[PC−/−] mice (Fig. 3f). We further demonstrated that treatment with JSH-23 to diminish NF-κB signaling compromised the growth of *Arid1a*-deleted xenografts to an extent similar as WT Myc-CaP tumors (Fig. 3g). RNA-seq analysis to compare the expression profiles of WT and *Arid1a* deleted tumors with or without NF-κB inhibition indicated that JSH-23 treatment led to the downregulated expressions of more than half of upregulated genes elicited by *Arid1a* loss (504/784; Fig. 3h). These genes included *Cxcl2*, *Cxcl3*, *Tnfα* and among others, indicating that *Arid1a* loss shapes immunosuppressive TME via NF-κB activation. We hypothesized that enhanced CXCR2 ligand expression by NF-κB signaling provided chemotaxis to attract CXCR2-expressing PMN-MDSCs to *Arid1a*-deleted tumors. We first analyzed the impacts of pharmacological inhibition of NF-κB signaling in tumor cells or CXCR2 in PMN-MDSCs using a transwell migration assay. Pretreatment with the P65 inhibitor JSH-23 or an anti-CXCL2 neutralizing antibody in *Arid1a* KO cells abrogated the attraction of PMN-MDSCs isolated from tumors (Fig. 3i). Moreover, pretreatment with the CXCR2 inhibitor SB255002 completely blocked the migration of PMN-MDSCs, while administration of the CCR2 inhibitor RS504393 had no effect (Fig. 3i).

More importantly, in vivo blockade of the CXCR2 axis using the SB255002 every other day dosing schedule for 14 days limited the growth of *Arid1a*-deleted tumors to a level similar to that in control tumors (Fig. 3j). Concomitant with a reduction in PMN-MDSCs, inhibition of CXCR2 restored the populations of total and IFNγ[+] CD8[+] T cells but not CD4[+] T cells in *Arid1a*-deleted tumors (Fig. 3k). Together, we conclude that the NF-κB/CXCL-CXCR2 axis stimulates the recruitment of PMN-MDSCs to shape an immunosuppressive TME.

### ARID1A ablation silences the enhancer of the *A20* gene to stimulate NF-κB signaling

We firstly assessed whether *Arid1a* loss affects the remaining SWI/SNF complex in mouse prostate tumors. However, qRT-PCR and immunoblotting analysis collectively pointed out that *Arid1a* loss did not significantly alter the remaining components, including ARID1B, BRG1, BAF155 and among others (Fig. 4a and Supplementary Fig. 4a). Moreover, we performed Co-IP analysis by using anti-BRG1 antibody to compare SWI/SNF composition in *Pten*[PC−/−] and *Pten*[PC−/−]; *Arid1a*[PC−/−] tumors. Regardless of *Arid1a* loss, BRG1 co-immunoprecipitated comparable amounts of SWI/SNF components, including ARID1B, BAF155, BAF53B and BAF45B (Supplementary Fig. 4b), suggesting that *Arid1a* loss did not significantly alter the SWI/SNF complexes that are remaining. Next, to compare chromatin accessibilities of PCa cells with or without *Arid1a* ablation, transposase-accessible chromatin using sequencing (ATAC-seq) of Myc-CaP cells revealed a total atlas of 65,188 peaks, with 32.43% of peaks found in intergenic regions, ~26.94% found in promoter regions and 32.98% in intron regions (Supplementary Fig. 4c). In contrast to the limited 2,505 peaks (3.8% among total peaks) gaining accessibility, *Arid1a* KO led to 54,964 peaks (84.3%) showing reduced accessibility (Fig. 4b). Further characterization of H3K27ac modifications to demarcate active transcription regions revealed that the sites losing chromatin accessibility after *Arid1a* loss simultaneously exhibited reduced H3K27ac modifications (Fig. 4b). We integrated data on ATAC-seq changes and messenger RNA expression level changes in the nearest genes, which revealed that more than half of genes downregulated after *Arid1a* KO (330 out of 645) concurrently displayed reduced chromatin accessibility (Supplementary Fig. 4d).

We directly assessed the consequences of *Arid1a* loss on the chromatin recruitment of the SWI/SNF complex. Due to the lack of a ChIP-grade ARID1A antibody, we performed ChIP-seq for core subunits of the BRG1 complex in control and *Arid1a* KO Myc-CaP cells. A similar approach was used in previous studies[32]. *Arid1a* ablation led to a global change in BRG1 occupancy on a genomic scale. Approximately 40% of BRG1-binding sites (7,402 out of 18,790) displayed reduced BRG1 binding after *Arid1a* ablation (Fig. 4c), while increased BRG1 occupancy in some regions was likely due to the secondary effects caused by *Arid1a* loss. Quantification revealed that most sites that exhibited decreased BRG1 targeting simultaneously exhibited significant reductions in chromatin accessibility and the intensities of H3K27ac modifications (Fig. 4d). Enhancers and promoters can be distinguished by the methylation status at H3K4. Histone H3K4me1 and H3K27ac are enhancer-specific modifications and are required for enhancers to activate transcription of target genes[33,34], whereas high levels of tri-methylation (H3K4me3) in combination with H3K27ac predominantly mark active or poised promoters[34–36]. Focusing on the peaks showing reduced BRG1 occupancy, we characterized histone modifications associated with active cis-regulatory elements (H3K4me3, H3K4me1, and H3K27ac). Approximately 50% of sites (3224 out of 7402 peaks) were defined as active enhancers marked with H3K4me1 and H3K27ac; in contrast, only a few (344 out of 7402 peaks) were categorized as active promoters enriched for H3K4me3 and H3K27ac modifications (Fig. 4e). These data suggest that *Arid1a* loss affects enhancer utilization in PCa cells.

To identify the putative targets of ARID1A that cause immune suppression, we aligned the genes showing reduced BRG1 peaks with

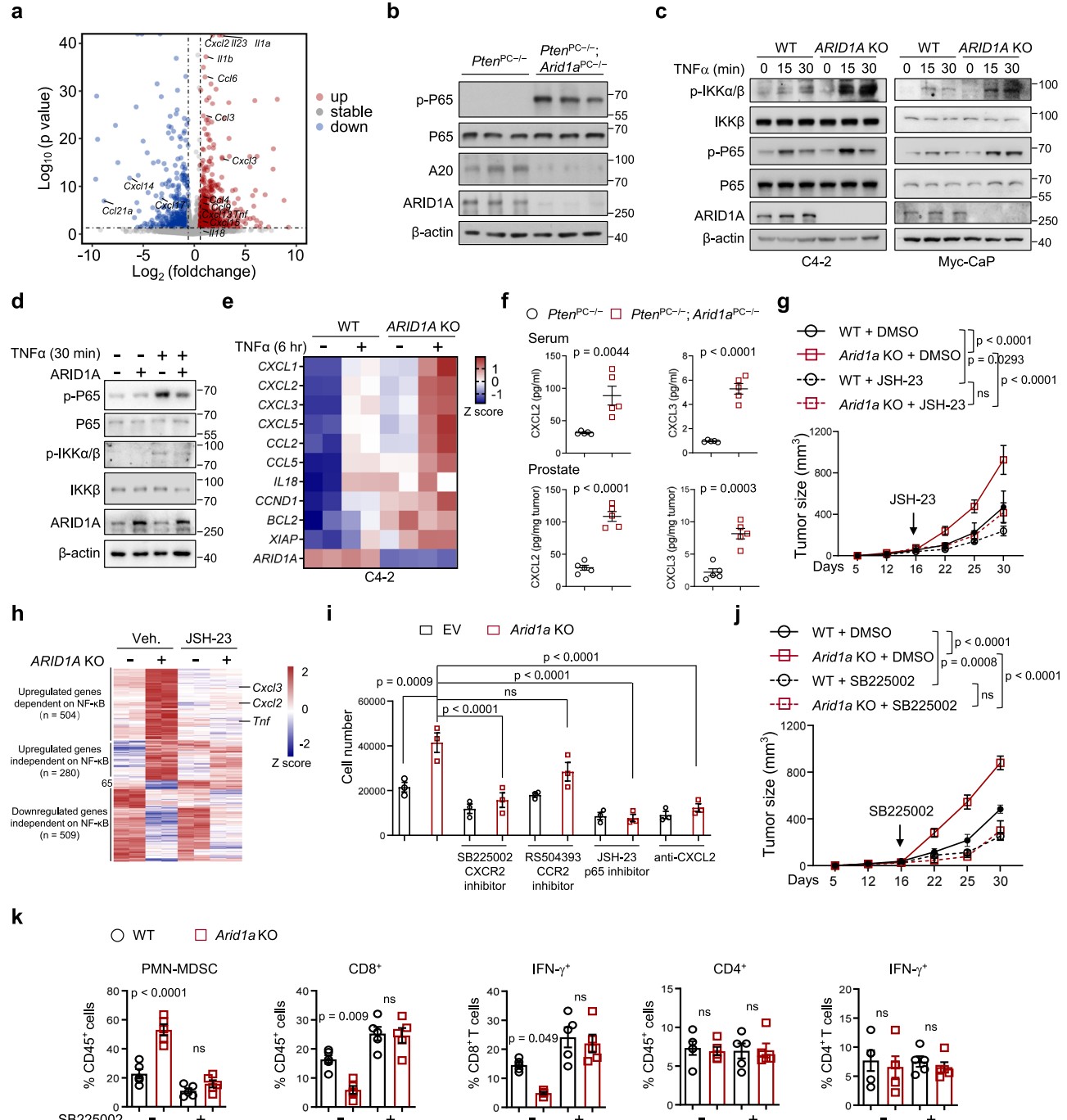

**Fig. 3 | Hyperactivation of NF-κB in ARID1A-depleted tumors causes the excess recruitment of MDSCs. a** Volcano plots showing the differentially expressed genes (DEGs) between epithelial cells of 3-month-old $Pten^{PC-/-}$; $Arid1a^{PC-/-}$ versus $Pten^{PC-/-}$ mouse prostates ($n = 3$). The upregulated and downregulated cytokines and chemokines are indicated. $p$-value was determined by DEGseq analysis. **b** IB analysis of the indicated protein in 3-month-old mouse prostates. **c** IB analysis of WT and $ARID1A$-depleted C4-2 and Myc-CaP cells treated with TNFα at the indicated time points. **d** IB analysis in WT and $ARID1A$-overexpression Myc-CaP cells with or without TNFα stimulation. **e** Heatmap summarizing the qRT-PCR results in WT and $Arid1a$ KO cells with or without TNFα stimulation. **f** ELISA of CXCL2 and CXCL3 in serum and prostate tumors of 3-month-old mice ($n = 5$). **g** Tumor volume of Myc-CaP expressing sg$ARID1A$ or control vector subcutaneously inoculated into FVB mice with or without JSH-23 treatment (WT + DMSO, $n = 5$; $Arid1a$ KO + DMSO,

$n = 7$; WT/$Arid1a$ KO + JSH-23, $n = 6$). **h** Epithelial cells in xenografts (**g**) were sorted for RNA-Seq and DEGs between WT and $Arid1a$ KO epithelium were shown in heatmap ($n = 2$). **i** Migration of PMN-MDSCs recruited by conditional mediums (CMs) with the indicated treatments ($n = 3$). **j, k** Mice were inoculated with WT or $Arid1a$ KO Myc-CaP cells and treated with or without SB225002. Tumor volume was monitored (**j**, WT + DMSO, $n = 5$; $Arid1a$ KO + DMSO, $n = 7$; WT/$Arid1a$ KO + JSH-23, $n = 6$) and quantification of each tumor-infiltrating immune cell population (**k**, $n = 5$) were measured by FACS analysis. **f, g, i–k** Data represent the mean ± SEM. Statistical significance was determined by two-tailed unpaired $t$-test (**f, i, k**) and two-way ANOVA followed by multiple comparisons (**g, j**). **b–d** Data were evaluated in triplicate, and representative data are shown. Source data are provided as a Source Data file. ns, no significance.

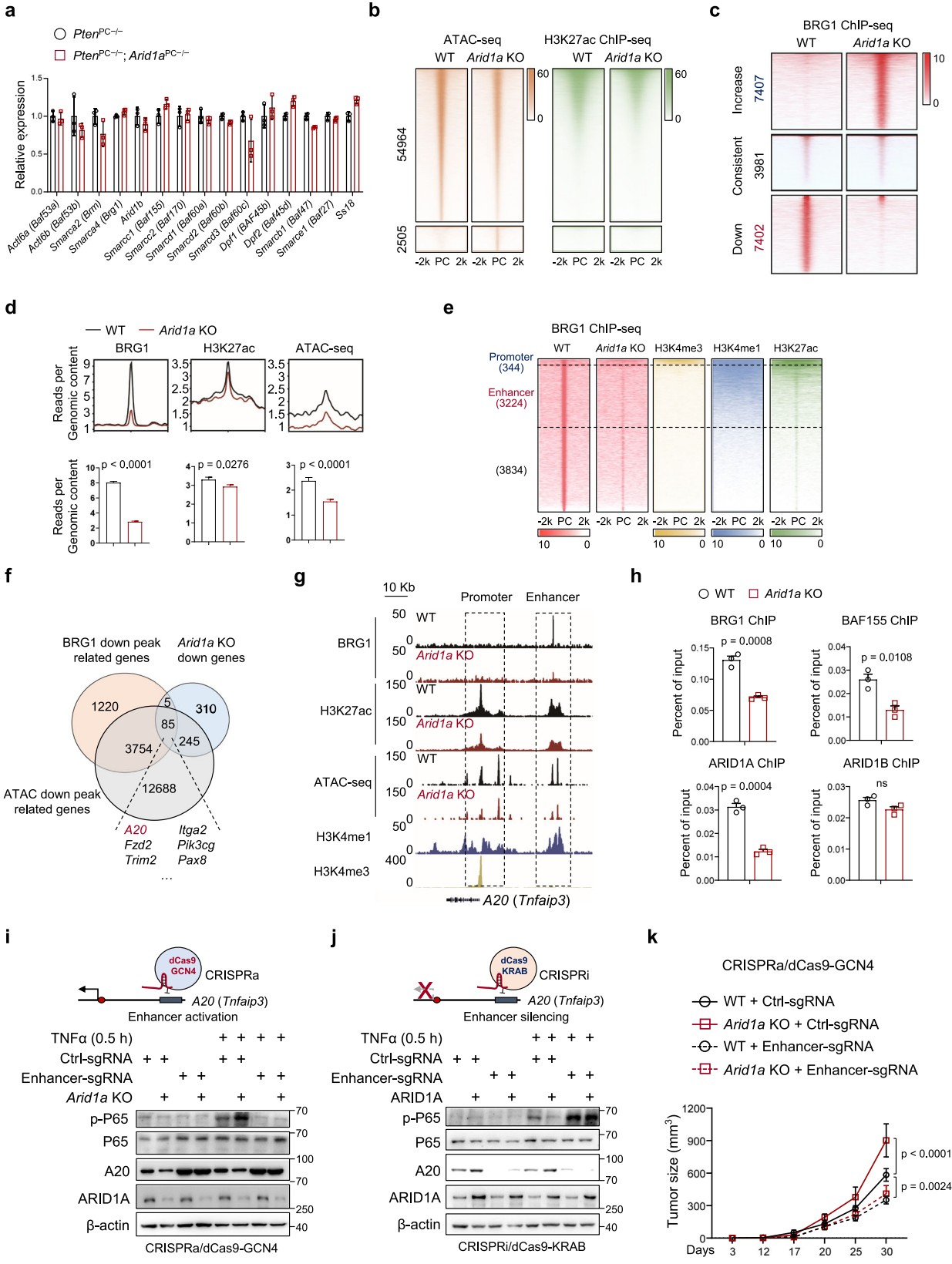

the decreases in chromatin accessibility and expression; 85 genes overlapped (Fig. 4f). Gene ontology (GO) biological process (BP) enrichment analysis revealed that these genes were related to pathways in cancer, NF-κB signaling and other pathways (Supplementary Fig. 4e). Interestingly, A20 deubiquitinase (also known as *Tnfaip3*), a critical negative regulator that restricts NF-κB activation, was

downregulated. A closer examination of BRG1 binding at *A20* gene loci revealed that *Arid1a* KO led to reduced BRG1 binding at enhancer but not promoter regions, which site was marked by H3K4me1 and H3K27ac but not H34K4me3 modifications (Fig. 4g). ChIP-qPCR assays verified that ARID1A, BAF155 and BRG1, but not ARID1B, were recruited to the enhancer region of the *A20* gene locus, and ARID1B was

**Fig. 4 | Arid1a ablation silences the enhancer of the A20 gene to stimulate NF-κB signaling. a** Relative expressions of SWI/SNF components in epithelium from 12-week-old $Pten^{PC-/-}$ and $Pten^{PC-/-}$; $Arid1a^{PC-/-}$ prostates ($n = 3$). **b** Heatmap of H3K27ac ChIP-seq in the differentially accessible sites obtained by ATAC-seq on Arid1a loss in Myc-CaP cells ($\pm 2$ kb regions centered at the peak summit). PC, peak center. **c** Heatmap of the ChIP-seq profiles of BRG1 peaks in control and Arid1a KO Myc-CaP cells shown in a horizontal window of $\pm 2$ kb from the peak center. **d** Profile and bar plot of BRG1 binding, H3K27ac modification and ATAC-seq intensities across BRG1 down peaks after Arid1a ablation. **e** ChIP-seq profiles for BRG1, H3K4me3, H3K4me1 and H3K27ac at the sites reducing BRG1 binding after Arid1a loss. **f** Venn diagram of the genes showing reduced BRG1 binding, accessibility and expression in Arid1a KO Myc-CaP cells compared to WT cells. **g** ChIP-seq tracks of BRG1, H3K27ac, H3K4me1, H3K4me3 and ATAC-seq signals in A20 gene loci as indicated. **h** ChIP-qPCR assays of BRG1, BAF155, ARID1A and ARID1B binding in the

A20 enhancer ($n = 3$), ns, no significance. **i** IB analysis of the indicated protein in WT and Arid1a KO Myc-CaP cells with or without A20 enhancer activation (CRISPRa/dCas9-based induction) and/or TNFα stimulation. **j** IB analysis in WT and ARID1A overexpressing Myc-CaP cells with or without A20 enhancer suppression (CRISPRi/dCas9-based suppression) and/or TNFα stimulation. **k** Tumor volume of the mice inoculated with Myc-CaP expressing sgArid1a and control vector with or without A20 enhancer activation (WT/Arid1a KO + Ctrl-sgRNA and WT + Enhancer-sgRNA, $n = 7$ for each group; Arid1a KO + Enhancer-sgRNA, $n = 5$). **a, d, h, k** Data represent the mean ± SEM. Statistical significance was determined by two-tailed unpaired t-test (**a, d, h**) and two-way ANOVA followed by multiple comparisons (**k**). **i, j** Experiments were repeated three times independently with similar results; data from one representative experiment are shown. ns, no significance. Source data are provided as a Source Data file.

incapable of compensating for ARID1A loss by binding to this site (Fig. 4h). A decrease in the transcript of enhancer RNA (eRNA) and H3K27ac modification in the absence of Arid1a indicated that Arid1a loss silences the enhancer activity of the A20 gene (Fig. 4g and Supplementary Fig. 4f). In contrast to previous reports[31], Arid1a ablation did not alter BRG1 targeting or H3K27ac modifications in the Cxcl9 and Cxcl10 gene loci (Supplementary Fig. 4g). Together, our results indicate that ARID1A loss silences the enhancer of the A20 gene.

Consistent with the role of A20 in restricting NF-κB signaling, Arid1a loss led to increases in the K63-linked polyubiquitin of RIP1 (Supplementary Fig. 4h), favoring the recruitment of the protein kinase TAK1 and the IκB kinase complex to ensure NF-κB signaling activation[37,38]. Furthermore, we employed dCas9-GCN4-mediated CRISPR activation (CRISPRa) with sgRNAs specific to the enhancer of the A20 gene locus to increase its expression. CRISPRa induced superacetylation in the enhancer regions and accordingly, the expression of A20. As a consequence, NF-κB signaling was attenuated in Arid1a KO cells to an extent similar to that in control cells (Fig. 4i and Supplementary Fig. 4i). Conversely, we targeted dCas9-KRAB (CRISPRi) to the enhancers of the A20 gene, leading to the formation of a heterochromatin-forming complex that causes enhancer inactivation. Enhancer silencing led to the transcriptional silencing of A20 (as evidenced by decreased expression and H3K27ac marks), and NF-κB activity was simultaneously restored in ARID1A-overexpressing cells to a level comparable to that in control cells (Fig. 4j and Supplementary Fig. 4j). More importantly, A20 enhancer activation abrogated the difference in tumor growth between mice bearing WT and Arid1a KO cells (Fig. 4k), and MDSCs, CD8$^+$ T cells and IFNγ$^+$ expressing CD8$^+$ T cells were present in similar proportions (Supplementary Fig. 4k). Altogether, these results indicate that ARID1A inhibits NF-κB signaling activity in PCa cells via the regulation of enhancer activity.

## IKKβ acts as the convergence point for inflammatory signals to promote ARID1A reduction in PCa cells

Low mutation frequency in ARID1A cannot account for its prominent downregulation in PCa, which prompted us to characterize the upstream signal to modulate ARID1A expression. Because ARID1A was reduced in Pten/Trp53 DKO tumors (Fig. 1a), we examined whether PTEN or P53 directly modulates ARID1A expression. Nevertheless, siRNA mediated deletion of P53 in C4-2 and 22RV-1 cells did not alter ARID1A expression (Supplementary Fig. 5a, b). Similarly, depletion of PTEN or concurrent loss of PTEN and P53 did not affect ARID1A level as well (Supplementary Fig. 5a, b). Thus, we conclude that PTEN and P53 are unlikely to directly control ARID1A level. Given the role of ARID1A in shaping the immunosuppressive TME, we explored whether inflammatory signals govern ARID1A expression. We found that TNF-α but not IL-6 or INF-γ treatments reduced ARID1A protein levels in PCa cells (Fig. 5a). In C4-2, PC3 and Myc-CaP cells, TNF-α-induced ARID1A downregulation was due to the reduction in protein expression but not mRNA expression (Supplementary Fig. 5c, d). Cotreatment of cells with

the proteasomal inhibitor MG132 prevented TNF-α-induced ARID1A degradation (Fig. 5b). In the context of TNF-α treatment, the most pronounced signaling event is IKK phosphorylation-dependent NF-κB activation. However, siRNA-based knockdown (KD) of the p65, p50 or p52 NF-κB subunit failed to inhibit TNF-α-induced ARID1A protein degradation (Fig. 5c). In contrast, silencing IKKβ but not IKKα abolished TNF-α-stimulated ARID1A degradation (Fig. 5d). Similarly, IKKβ depletion attenuated the TNF-α-induced polyubiquitination of ARID1A (Fig. 5e), indicating that TNF-α-induced ARID1A downregulation depends on IKKβ.

Consistent with this finding, ARID1A bound to IKKβ with higher affinity than to other IKK family members (IKKα and NEMO) in 293 T cells (Supplementary Fig. 5e). TNF-α treatment enhanced the association between endogenous ARID1A and IKKβ but not other IKK members (Fig. 5f). Domain mapping revealed that the N-terminus of the IKKβ-containing kinase domain is responsible for the association with ARID1A (Supplementary Fig. 5f), and the middle region of ARID1A interacts with IKKβ (Supplementary Fig. 5g). GST pulldown experiments supported a direct interaction between ARID1A and IKKβ (Supplementary Fig. 5h). The results showing that ARID1A protein stability was tightly regulated by IKKβ prompted us to perform a gel-filtration chromatography assay in PCa cells. As expected, the elution profile of the IKK core complex, including IKKα, IKKβ, and NEMO, was identical after TNF-α treatment (Fig. 5g). Notably, IKKβ and p-IKKβ expression outside the IKK complex was also detected, and both forms were found to be tightly colocalized with ARID1A (Fig. 5g). Because ARID1A is a nuclear protein, we also separated cells into cytoplasmic and nuclear fractions and examined the presence of IKKβ. The experiments demonstrated a nuclear subfraction of IKKβ, whose abundance was enhanced by TNF-α treatment (Supplementary Fig. 5i). This observation was consistent with previous reports[39,40] and was further supported by immunofluorescent staining (Supplementary Fig. 5j), indicating that IKKβ interacts with ARID1A in cells.

Activation of IKKβ depends on its phosphorylation at Ser177 and Ser181[41]; thus, we assessed whether IKKβ phosphorylation was required for its regulation of ARID1A. Compared to WT IKKβ and the phosphorylation mimic IKKβ-SD, phosphorylation-incompetent mutant (Ser177Ala and Ser181Ala, IKKβ-SA) was unable to promote ARID1A turnover (Fig. 5h), indicating that IKKβ phosphorylation is required for its role in ARID1A degradation. The IKK complex represents a converging point for transducing diverse NF-κB-activating stimuli, particularly TNF-α and IL-1[41]. We found that IL-1β or c-GAMP treatment also promoted ARID1A downregulation whenever IKKβ phosphorylation was induced (Fig. 5i). Next, we extended the analysis to address whether this regulation is a cell type specific. TNFα treatment stimulated ARID1A turnover in lung and colorectal cancer cells, A549 and HCT166 cells (Supplementary Fig. 5k). Similar results were obtained in Jurkat T cells, RAW264.7 macrophage cells and OCI-Ly10 B cells (Supplementary Fig. 5l), suggesting that this regulatory axis

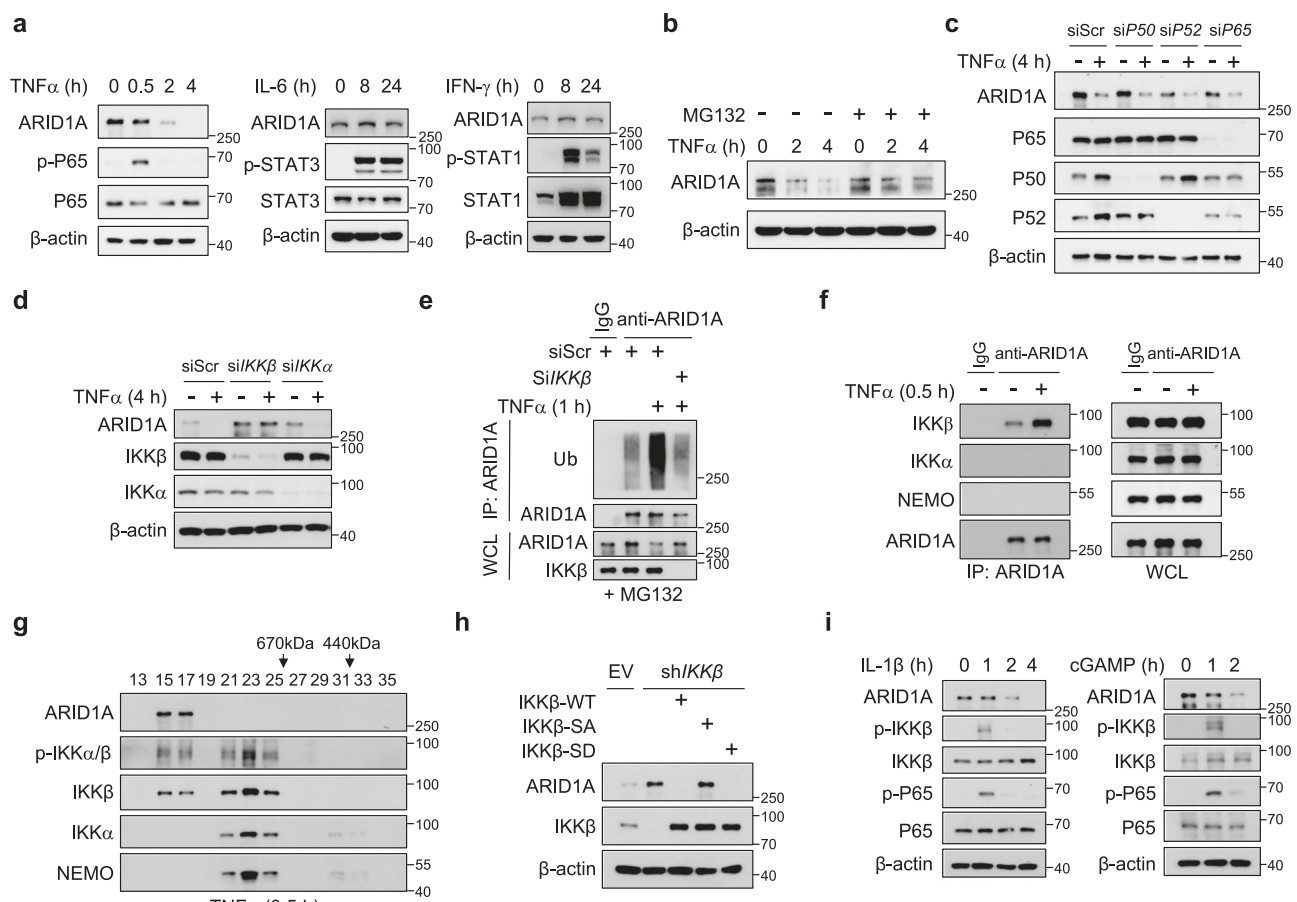

**Fig. 5 | IKKβ acts as the convergence point for inflammatory signals to cause ARID1A destruction. a** IB analysis of C4-2 cells treated with TNFα, IL-6 and IFN-γ for the indicated duration of time. **b** IB analysis of C4-2 cells stimulated with TNFα with or without MG132 treatment for the indicated duration of time. **c** IB analysis of C4-2 cells transfected with scramble or *P50*, *P52*, or *P65* oligonucleotides with or without TNFα treatment. **d** IB analysis of C4-2 cells transfected with scramble or *IKKβ* and *IKKα* oligonucleotides with or without TNFα treatment. **e** IB analysis of the WCL from WT and *IKKβ* KD C4-2 cells with or without TNFα treatment and anti-ARID1A

immunoprecipitates as indicated. **f** IB analyses of the WCL and anti-ARID1A immunoprecipitates of C4-2 cells. **g** IB analysis of nuclear extracts from C4-2 cells after gel-filtration fractionation. Cells were treated with 20 ng/ml TNFα for 30 min before harvesting. **h** IB analysis of the indicated proteins in C4-2 control and *IKKβ* KD cells with or without IKKβ-WT, IKKβ-SA or IKKβ-SD overexpression. **i** IB analysis of C4-2 cells treated with IL-1β or c-GAMP for the indicated duration of time. All experiments were repeated three times independently with similar results; data from one representative experiment are shown. Source data are provided as a Source Data file.

represents a general mechanism independent of cell types. Collectively, our results demonstrate that IKKβ acts as the convergence point for inflammatory signals, such as TNF-α and IL-1, to promote ARID1A downregulation.

## IKKβ phosphorylates ARID1A and promotes its destruction via β-TRCP

Phosphorylation of IKKβ at Ser177 and Ser181 residues enables IKKβ to phosphorylate substrates (e.g., β-catenin and IκB)[41,42]. Using an anti-phosphoserine antibody, we found that TNF-α treatment led to increased ARID1A phosphorylation, while IKKβ knockout or incubation with λ-PPase eliminated ARID1A phosphorylation, suggesting that ARID1A is a phosphorylation substrate of IKKβ (Fig. 6a). We further demonstrated that IKKβ self-phosphorylation is required for subsequent ARID1A phosphorylation. First, an in vitro kinase assay revealed that the IKKβ-SA mutant failed to phosphorylate ARID1A, whereas the phosphomimetic IKKβ-SD mutant enhanced phosphorylation (Fig. 6b). Second, forced expression of IKKβ-SD but not IKKβ-SA in IKKβ-depleted cells enhanced the phosphorylation and polyubiquitination of ARID1A (Supplementary Fig. 6a). To further define the phosphorylation site of ARID1A, we aligned the amino acid sequences of ARID1A across species and found potential phosphorylation sites (Ser1316 and 1320 residues) fitting the IKKβ

phosphorylation consensus motif (DpSGΨXpS/T) (Fig. 6c). We mutated this possible IKKβ phosphorylation residue to alanine (ARID1A-SA), and we found that WT but not SA mutant ARID1A protein was directly phosphorylated by IKKβ in vitro (Fig. 6d). Exogenously expressed ARID1A-SA protein displayed a reduced turnover rate compared with that of WT ARID1A in 293 T cells (Supplementary Fig. 6b). We therefore substituted Ser1316/Ser1320 in endogenous ARID1A for alanine in C4-2 cells by applying CRISPR–Cas9 technology and thereby generated *ARID1A*[Mut] cells (Supplementary Fig. 6c). Along with diminished ARID1A phosphorylation, *ARID1A*[Mut] cells exhibited reduced ARID1A polyubiquitination compared with WT cells (Fig. 6e). In addition, ARID1A-SA mutated protein was resistant to IKKβ-SD-induced destruction, while WT ARID1A was not (Fig. 6f). Similarly, we generated *Arid1a*[Mut] knock-in Myc-CaP cells and observed increased ARID1A protein expression but not increased mRNA expression (Supplementary Fig. 6d, e). More importantly, the mice engrafted with *Arid1a*[Mut] cells developed significantly smaller tumors and exhibited reduced MDSC infiltration and increased CD8+ T cell infiltration relative to that in WT tumors (Fig. 6g, h). Together, these results demonstrate that IKKβ-induced phosphorylation regulates ARID1A protein stability.

IKK influences substrate ubiquitination by altering interactions between substrate proteins and their E3 ubiquitin ligases in a

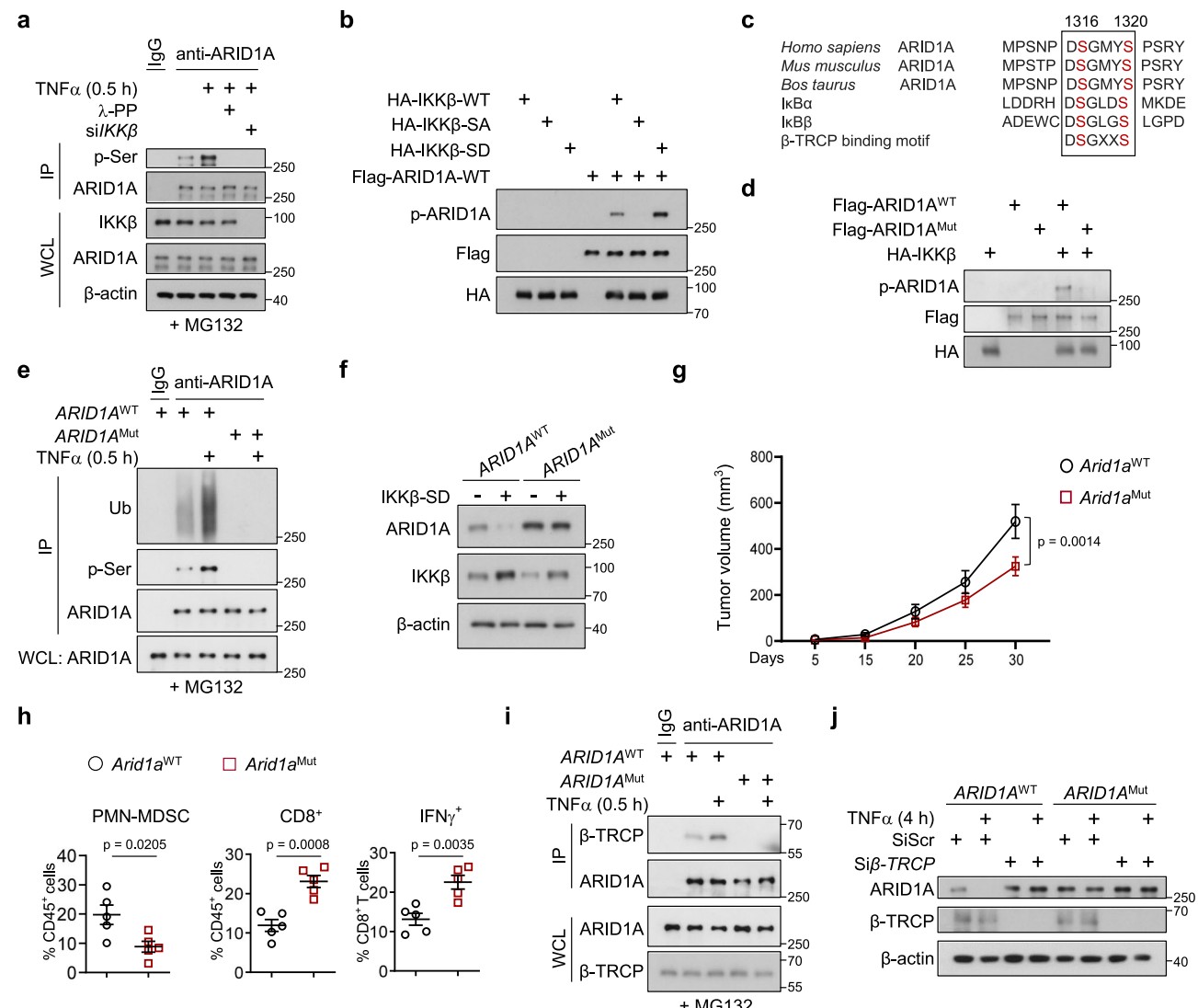

**Fig. 6 | IKKβ phosphorylates ARID1A and promotes ARID1A destruction via β-TRCP. a** IB analysis of the WCL and immunoprecipitates derived from C4-2 cells treated with TNFα with or without λ-phosphatase or *IKKβ* oligonucleotides. **b** In vitro kinase assays indicated that the phosphorylation of IKKβ was required for ARID1A phosphorylation. **c** Sequence alignment of the putative IKKβ phosphorylation sites and β-TRCP binding motif at S1316 and S1320 of ARID1A. **d** In vitro kinase assays to examine the phosphorylation of S1316 and S1320 sites in ARID1A by IKKβ. **e** WT and mutant ARID1A cell lysates were subjected to IP with anti-ARID1A antibody and IB with anti-ubiquitin (anti-Ub) and anti-p-Ser antibody. **f** IB analysis of WT and mutant ARID1A cells with or without IKKβ-SD overexpression. **g, h** FVB mice

were inoculated with WT or *Arid1a*^Mut Myc-CaP cells, and tumor volume (**g**, *n* = 5) and quantification of each tumor-infiltrating immune cell population (**h**, *n* = 5) were measured by FACS. **i** IB analysis of the WCL and immunoprecipitates derived from WT and *ARID1A*^Mut C4-2 cells treated with or without TNFα. **j** IB analysis of WT and *ARID1A*^Mut C4-2 cells transfected with scramble or *β-TRCP* oligonucleotides with or without TNFα stimulation. **g, h** Data represent the mean ± SEM. Statistical significance was determined by two-way ANOVA followed by multiple comparison (**g**) and two-tailed unpaired *t*-test (**h**). **a, b, d–f, i, j** Experiments were repeated three times independently with similar results; data from one representative experiment are shown. Source data are provided as a Source Data file.

phosphorylation-dependent manner[41–43]. Thus, we reasoned that IKKβ-induced phosphorylation of ARID1A alters the recruitment of E3 ligases. Interestingly, the phosphorylation sites identified (Ser1316/Ser1320) also lie in the conserved β-TrCP degradation motif (DSGXXS degron), and β-TRCP, an E3 ligase that preferentially binds to phosphorylated substrates, is known to degrade ARID1A in gastric cancer cells[44]. Therefore, we examined whether phosphorylation of Ser1316/Ser1320 in ARID1A enabled its recognition by β-TrCP E3 ligase. Coimmunoprecipitation assays revealed that phosphorylation-incompetent ARID1A lost the capacity to interact with β-TRCP in *ARID1A*^Mut cells relative to WT cells (Fig. 6i). β-TRCP depletion rendered WT ARID1A insensitive to TNFα treatment, and ARID1A protein in *ARID1A*^Mut cells was resistant to TNFα treatment or β-TRCP depletion (Fig. 6j). Thus, IKKβ promotes ARID1A degradation through the β-TrCP-mediated ubiquitin proteasome pathway.

## Inhibition of NF-κB signaling sensitizes ARID1A-deficient tumors to ICB therapy

We asked whether these findings were of clinical relevance. Quantitation of the western blot results in tumors with high GS (>7; *n* = 42; Supplementary Table 2) revealed inverse relationships between the levels of ARID1A versus p-P65 and p-IKKα/β (Fig. 7a). On the other hand, tumors with higher ARID1A expression tended to exhibit higher A20 expression (Fig. 7a). Quantitative ELISA results in the same cohort verified that tumors with lower ARID1A expression produced significantly more CXCL2 and CXCL3 (Fig. 7b). We validated these results in PCa datasets, showing that ARID1A and NF-κB signatures, as defined by expression profile changes, were negatively associated in primary PCa and lethal CRPC (Fig. 7c). We also compiled a list of 39 MDSC-related genes from the literature[16]. As indicated in Fig. 7c, the ARID1A signature was significantly downregulated in MDSC-high samples

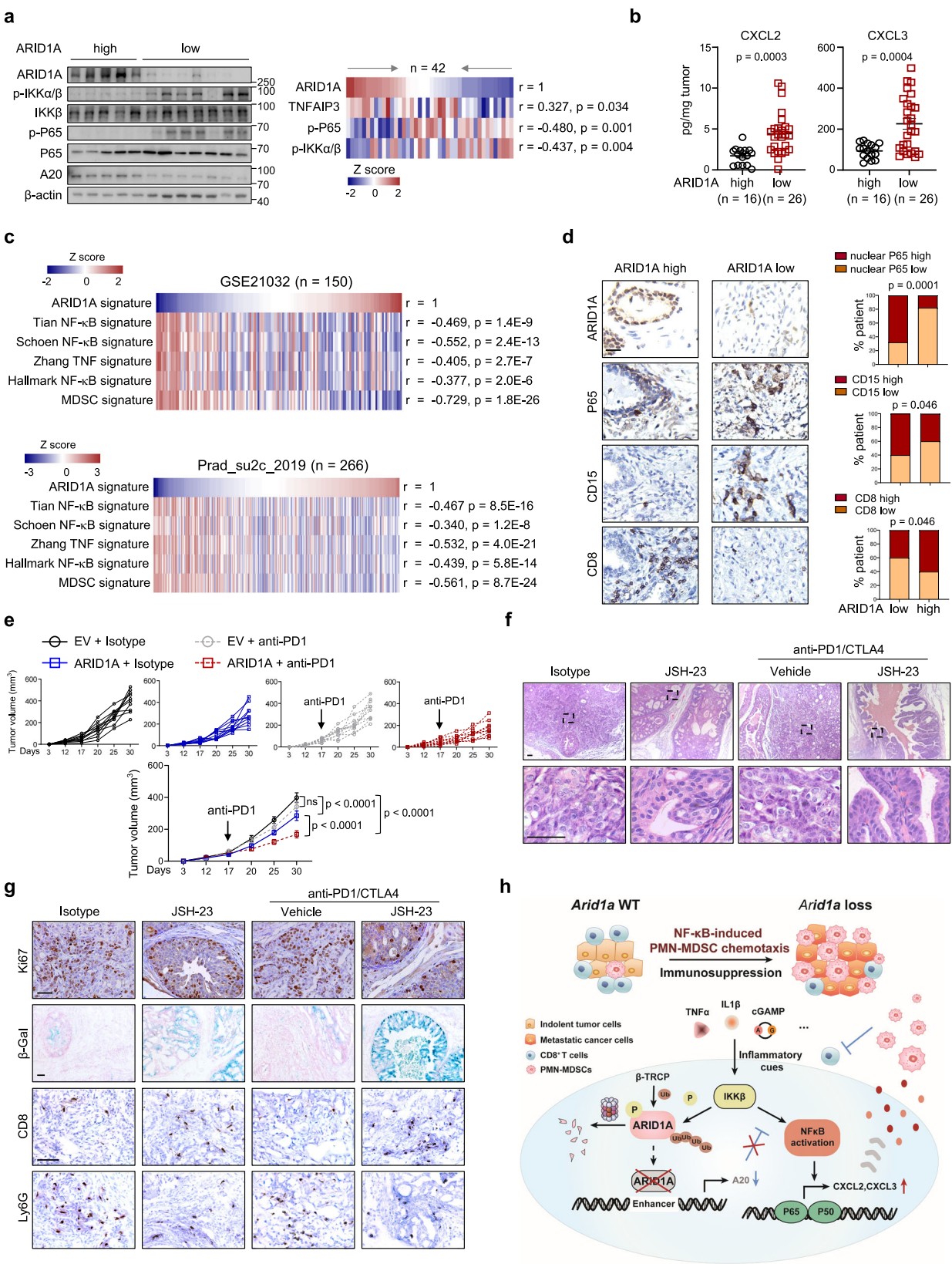

compared to MDSC-low samples. We extended the analysis to a tumor tissue microarray and found that the ARID1A level was positively and negatively associated with the infiltration of CD8[+] cells and the nuclear staining of P65 in PCa cells, respectively (Fig. 7d). In addition, tumors with low ARID1A expression possessed significantly more CD15[+] cells (dictation of human MDSCs[45]) than those with high ARID1A expression

(Fig. 7d). We quantified the abundances of CD15[+] cells and found that their abundances were positively associated with GS (Supplementary Fig. 7a), and the proportions of CD8[+] and CD15[+] cells in tumors were negatively and positively associated with biochemical recurrence, respectively (Supplementary Fig. 7b). Although PTEN did not directly regulate ARID1A expression, IHC analysis revealed a positive

**Fig. 7 | Inhibition of NF-κB signaling sensitizes ARID1A-deficient tumors to ICB therapy. a** Representative IB results of ARID1A, p-IKKβ, p-P65 and A20 expression in the lysates of human prostate tumors. Pearson's correlations among proteins indicated in PCa specimens are summarized in the heatmap ($n = 42$; GS > 7). **b** ELISA of CXCL2 and CXCL3 in PCa ($n = 42$). **c** Heatmap summary of the correlations of the indicated signature in PCa ($n = 150$, GSE21032; $n = 266$, Prad_SU2C_2019). **d** IHC analysis for ARID1A, P65, CD15 and CD8 markers. Scale bars, 50 μm. The correlations between ARID1A expression and nuclear P65 intensity and the abundance of CD15$^+$ and CD8$^+$ cells are shown as stacked columns ($n = 100$). **e** Volume of tumors derived from WT and *ARID1A*-overexpressing cells injected subcutaneously into FVB mice and treated with IgG and anti-PD1 antibody ($n = 10$). **f** Prostate tumor histology of *Pten*$^{PC-/-}$; *Arid1a*$^{PC-/}$ mice with or without NF-κB inhibition (JSH-23) in combination with anti-PD1/CTLA-4 treatment ($n = 10$), Scale bars, 50 μm. **g** IHC staining for Ki67, CD8 and Ly6G and β-Gal in sections by the indicated treatments. Scale bars, 50 μm. **h** ARID1A functions downstream of inflammation-induced IKKβ activation to shape the immunosuppressive TME through the regulation of NF-κB-mediated chemotaxis. **b** and **e** Data represent the mean ± SEM. Statistical significance was determined by two-tailed Pearson's correlations test (**a** and **c**), two-tailed unpaired *t*-test (**b**), two-tailed χ$^2$ (**d**) and two-way ANOVA followed by multiple comparisons (**e**). **g** Experiments were repeated at least three times independently with similar results; data from one representative experiment are shown. Source data are provided as a Source Data file.

correlation between PTEN and ARID1A level in PCa, highlighting our mouse model was clinically relevant (Supplementary Fig. 7c). Together, these results reinforce the link between MDSC-high prostate tumors and reduced ARID1A activity.

Advanced PCa is resistant to ICB therapy, which is attributed to intrinsic tumor cell mechanisms and low levels of immune infiltrates[46–48]. We showed that ARID1A overexpression slightly but significantly reduced Myc-CaP cells derived xenograft growth, and WT tumors were refractory to anti-PD1 antibody treatment (Fig. 7e). In contrast, ARID1A overexpression rendered the tumors sensitive to anti-PD1 antibody treatment, as reflected by the further reduction in tumor volumes as compared to isotype treatment (Fig. 7e). In contrast to the limited infiltration of CD8$^+$ T cells in WT tumors, ARID1A-overexpressing tumors treated with anti-PD1 antibody showed synergistic increases in the proportions of total and IFNγ$^+$ CD8$^+$ T cells (Supplementary Fig. 7d). Next, we evaluated the therapeutic effects of an NF-κB inhibitor and/or anti-PD1/CTLA4 antibodies in *Pten*$^{PC-/-}$; *Arid1a*$^{PC-/-}$ mice. To this end, we included *Pten*$^{PC-/-}$; *Arid1a*$^{PC-/-}$ tumors with NF-κB inhibition alone to compare with co-targeted treatment (the combinational regimen of NF-κB inhibitor, JSH-23 and anti-PD1/CTLA4 antibody). As previously reported[17,49], JSH-23 was administrated by gavage daily and/or anti-PD1/CTLA4 antibodies were simultaneously administrated twice a week. Judged by histology quantitation, IHC analysis and immunoprofiling data, NF-κB inhibition compromised *Arid1a*-deleted tumor progression along with the reduced PMN-MDSC infiltrations and expansion of CD8$^+$ T cells (Fig. 7f and Supplementary Fig. 7e, f). Accordingly, NF-κB inhibition attenuated the elevation of CXCL2 and CXCL3 in *Arid1a*-deleted tumors (Supplementary Fig. 7g). These results support that *Arid1a* loss expediates PCa progression via NF-κB-induced PMN-MDSC chemotaxis. Similar to other aggressive PCa models[17,19] (e.g., *Pten*$^{PC-/-}$; *Trp53*$^{PC-/-}$; *Smad4*$^{PC-/-}$), *Pten*$^{PC-/-}$; *Arid1a*$^{PC-/-}$ mice exhibited primary resistance to ICB treatment containing anti-CTLA4 and anti-PD1 antibodies (Fig. 7f and Supplementary Fig. 7e). We showed that NF-κB inhibition and ICB therapies acted in synergistical manner to restrict *Pten*$^{PC-/-}$; *Arid1a*$^{PC-/-}$ tumor progression. Combinational treatment of JSH-23 and anti-PD1/CTLA4 antibody further provoked the regression of *Pten*$^{PC-/-}$; *Arid1a*$^{PC-/-}$ tumors as compared to JSH-23 treatment alone (Fig. 7f and Supplementary Fig. 7e). Necropsy at the termination of 1.5 months treatment revealed that 2 of 10 anti-NF-κB/PD1/CTLA4-treated mice showed no disease, and 6 of 10 exhibited minimal residual invasive tumors. NF-κB blockade in combination with ICB inhibited PMN-MDSC infiltration and restored CD8$^+$ T cell level in *Arid1a*-deleted tumors (Supplementary Fig. 7f). Further IHC staining of Ki67, CD8, Ly6G, and β-Gal validated the pathology analysis results (Fig. 7g). Together, our results reinforce that the inhibition of NF-κB signaling sensitizes advanced PCa tumors, such as *Arid1a*-deficient tumors, to ICB therapy.

## Discussion

ARID1A is presumed to be a tumor suppressor based on loss-of-function mutational profiles in a broad variety of human cancers. Recent clinical specimen analyses reveal that ARID1A mutations or its

low expression are associated with antitumor immunity[31,50]. Nevertheless, little is known about its roles in TME remodeling and immune evasion. We showed that the ARID1A-mediated feedback axis promotes localized PMN-MDSC expansion and immune evasion (Fig. 7h), highlighting that ARID1A is a key epigenetic regulator that coordinates chronic inflammation and immune suppression to curtail PCa progression. In addition, our results indicated that *Arid1a* loss renders tumors refractory to anti-androgen therapy (Supplementary Fig. 2l–p). Expect for NF-κB-induced PMN-MDSC chemotaxis, whether other mechanism causing castration resistance awaits further investigations.

While targeting immune suppression mechanisms in the TME has revolutionized cancer treatment, clinical trials with anti-CTLA4 or anti-PD1 only show minimal benefit in PCa patients[46–48]. Immune evasion may be related to impaired tumor-specific antigen presentation and/or the presence of suppressive immunocytes, such as MDSCs, Tregs, and M2-type macrophages[51,52]. MDSCs are of particular interest, considering their pervasiveness in PCa and their profound inhibitory effect on T-cell functionality[52]. Our findings support the view that inhibition of PMN-MDSCs improves the efficiency of anti-PD1/CTLA4 treatment to provoke the regression of advanced PCa. We have noted that known toxicities falling within the spectrum of irAEs (immune related adverse events), which have already been described for CTLA-4 or PD1 pathway blocking agents[53]. Till now, great efforts have been made to target NF-κB signal as a therapeutic approach, which support or enhance the effectiveness of chemotherapeutic agents in clinical trial including proteosome inhibitors like bortezomib and carfilzomib. The anticancer effect of bortezomib and carfilzomib are partially attributed to inhibiting NF-κB activity by preventing IκB degradation[54]. For instance, bortezomib is a valuable treatment option in the management of relapsed multiple myeloma that improves survival and delays disease progression[55,56]. Our preclinical results emphasized the beneficial effects of co-targeting immune checkpoints and NF-κB or CXCR2 in advanced PCa, providing rationales for future clinical trials for patients. Interestingly, studies indicate that BET inhibitor (BETi) prevents the recruitment of BRD4 to p65-bound cis-regulatory elements to disrupt pro-survival NF-κB signaling, and thereby inducing unrestrained TNF-mediated activation of the extrinsic apoptotic cascade[57]. As a consequence, BETi sensitizes tumors to ICB therapy such as anti-PD1 treatment. Based on our results, it is reasonable to speculate that BETi might render *Arid1a* low or deleted tumors more sensitive to ICB, which remains to be further defined.

Further study is needed to determine whether immune cells other than MDSCs, such as macrophages, Tregs and NK cells, are affected by ARID1A loss. Recent studies have revealed that ARID1A loss impairs IFN signaling, especially Th1-type chemokine (CXCL9 and CXCL10) expression, to compromise effector T-cell tumor trafficking and antitumor immunity[31]. However, we found that ARID1A loss did not decrease *Cxcl9* and *Cxcl10* expression or chromatin configuration. Thus, ARID1A is unlikely to directly regulate the expression of Th1-type chemokines to cause immune evasion in our experimental models. Instead, NF-κB-induced PMN-MDSC chemotaxis was found to be the

culprit causing the metastatic outgrowth of *Arid1a*-deficient prostate tumors. The discrepancy is likely due to different cellular and genetic milieus, highlighting that ARID1A regulates the chromatin accessibility of target genes in a context-dependent manner.

ARID1A is described as a key tumor suppressor in a broad array of cancers, such as HCC and ovarian carcinoma[25,58,59]. Nevertheless, it also shows that ARID1A has context-dependent oncogenic role[58]. A study by Sun et al. demonstrates that ARID1A exerts tumor-promoting functions during the early phases of liver transformation[58]. Nevertheless, mice with liver-specific *Arid1a* loss in established tumors accelerates progression and metastasis[58]. Mechanistically, *Arid1a* promotes tumor initiation by increasing CYP450-mediated oxidative stress, while *Arid1a* loss after initiation decreased chromatin accessibility and reduced transcription of genes associated with metastasis[58]. In contrast to the observations that *Arid1a* loss promoted PCa progression in immunocompetent mice, we observed that *Arid1a* KO exhibited the inhibitory effects when tumor cells were engrafted in *Rag1* null mice (Fig. 2e), suggesting that tumor or innate immune cell mediated mechanisms might account for a negative role of ARID1A loss on tumor growth. Recent discoveries indicate that defective control of enhancer activity is a principal mechanism underlying tumor suppression caused by ARID1A inactivation[27,60]. In ARID1A-mutant colorectal cancer and ovarian clear cell carcinoma, many enhancers lose SWI/SNF binding and cannot stimulate gene expression[27]. We consistently found that approximately half of the sites showing reduced BRG1 targeting after *Arid1a* ablation were also cataloged as enhancers. These regions exhibited reduced H3K27ac modifications and chromatin accessibility, supporting the role of ARID1A in enhancer-mediated gene regulation. This notion is exemplified by the indispensable role of ARID1A in activating the enhancer of the *A20* gene. SWI/SNF cooperates with transcription factors or coregulators to activate target gene expression. For example, in estrogen-receptor-positive breast cancer, ARID1A loss impairs SWI/SNF binding at ER-FOXA1-GATA3 sites to determine luminal identity and therapeutic response[32]. Future studies need to clarify the cooperation between ARID1A-dependent SWI/SNF targeting and transcription factors to determine the specificity and extent of gene activation.

The ARID1A mutation frequency is low in PCa, and the regulatory mechanism downregulating ARID1A is not clear in PCa. Here, we revealed that the ARID1A protein is phosphorylated by IKKβ to trigger its proteasomal degradation in a β-TRCP-dependent manner. Signaling pathways that mediate the protumorigenic effects of inflammation are often subject to a feed-forward loop[61]. For example, activation of NF-κB in immune cells induces the production of cytokines that stimulate NF-κB in cancer cells to induce chemokines that attract more inflammatory cells into the tumor[61]. The ARID1A signal cascade identified here manifests a positive feedback loop to shape the TME in PCa. Inflammation-induced IKKβ activation promoted ARID1A downregulation, which in turn caused A20 reduction to trigger the secretion of NF-κB-dependent inflammatory cytokines. Beside transcriptional regulation, ARID1A is also found to be regulated by the ATM/β-TRCP axis and TRIM32/USP11 in other cancer types[44,62]. Thus, ARID1A might serve as a hub to sense various stimuli to shape the TME and impact tumorigenesis.

Together, our results demonstrate a critical role of ARID1A in MDSC recruitment and demonstrate that the IKKβ/ARID1A/NF-κB feedback axis integrates inflammation and immunosuppression to drive PCa progression. Targeting ARID1A or its coordinated signaling cascade might be an alternative approach to combat de novo resistance to ICB therapy in PCa.

## Methods
The study is performed with all relevant ethical regulations of the Institutional biomedical research ethics committee of Shanghai Institute of Nutrition and Health, Chinese Academy of Sciences.

### Cell culture
Cell lines were purchased from American Type Culture Collection (293 T, CRL-11268; C4-2, CRL-3314; PC-3, CRL-1435; 22RV-1, CRL-2505; DU145, HTB-81; HCT116, CCL-247; RAW 264.7, TIB-71; Myc-CaP, CRL-3255), Cell Bank, Shanghai Institute of Biochemistry and Cell Biology, Chinese Academy of Sciences (A549, SCSP-503; Jurkat, SCSP-513) or Nanjing Cobioer Biosciences Co., LTD, P. R. China (OCI-LY10, CBP60558). C4-2, PC-3, 22RV-1 and Jurkat cells were cultured in RPMI 1640 media supplemented with 10% FBS at 37 °C under 5% $CO_2$. 293 T, Myc-CaP, DU145, A549, HCT116, Raw264.7, and OCI-Ly10 cells were cultured in DMEM media at the same growth conditions. Cell lines were tested to confirm lack of mycoplasma contamination. The culture and experiments of organoid cells generated in this study were performed according to previously reported[63,64].

### Human tumor samples and TMA analysis
The use of pathological specimens, as well as the review of all pertinent patient records, was approved by the ethical standards of the institutional research committee and with the 1964 Declaration of Helsinki and its later amendments or comparable ethical standards. All patient samples were collected by the Department of Pathology with approval from the Research Ethics Committee of Daping Hospital, Army Medical University, and informed consent was obtained from the patients. To determine the correlation of ARID1A expression, NF-κB signaling activity and protein levels of CXCL2, CXCL3 in PCa tissues, a total of 42 prostate samples (GS > 7) were collected during surgical radical prostatectomy (Supplementary Table 2). Immunostaining of TMA (adjacent normal tissues: 40; tumor sample: 100; Supplementary Table 1) were performed by Department of Pathology at the Daping Hospital, Army Medical University by using anti-ARID1A (Cell Signaling Technology; 12354 S, clone: D2A8U, 1:500), CD8 (Abcam; ab189926, clone: EPR10640(2), 1:200), CD15 (Abcam; ab220182, clone: FUT4/815, 1:500), P65 (Cell Signaling Technology; 8242 S, clone: D14E12, 1:500). The tissue samples and clinical parameters of PCa patients who underwent radical prostatectomy were collected, including age at diagnosis, baseline serum PSA level, GS, adverse pathological features (extraprostatic extension, seminal vesicle invasion, lymph node invasion, and positive surgical margins), and follow-up PSA levels. No patients received adjuvant therapy until biochemical recurrence. For ARID1A and P65, staining intensity was scored following a three-tiered system, yielding a staining index ranging from 1 to 9 (extensive, strong staining). CD8+ and CD15+ cells were measured as the number of CD8 and CD15 positive cells in each TMA core compared to the total cells number. Low expression of ARID1A and P65 were defined by a staining index below 6, whereas staining scores if 6–9 were considered high expression. CD8+ and CD15+ cells were measured based on the number of CD8 and CD15 positive cells versus the total cells number. The median positive numbers were set as the cut offs to divide the patients. IHC results were quantified by pathologists blinded to the outcome. All the analyses were conducted or confirmed by two certified clinical pathologists independently (QL and JJ).

### Animal models and experiment assays
All mice were maintained in a specific-pathogen-free (SPF) facility, and all related protocols were performed in compliance with the Guide for the Care and Use of Laboratory Animals and were approved by the Institutional biomedical research ethics committee of Shanghai Institute of Nutrition and Health, Chinese Academy of Sciences. The maximal tumor size/burden permitted (2 cm³) by ethics committee was not exceeded in the study. Mice were housed under specific-pathogen-free conditions with standard food and water ad libitum in a 12 h light and 12 h dark cycle. Humidity and ambient temperature were maintained between 45–65% and 20–24 °C, respectively. Arid1a-floxed mice were provided by Dr. Hongbin Ji (Shanghai Institute of Biochemistry and Cell

Biology, Chinese Academy of Sciences). *Pten*-floxed mice were generated by Hong Wu[4]. The PB[Cre/+] transgenic mice were obtained from Fen Wang[65]. For GEMM studies, all mice are maintained in C57BL/6 background, and the histology analysis and molecular characterizations are described as below. Male mice were sacrificed for analysis at 3- and 4-month-old. For subcutaneous injection, 4–6-week-old male mice (C57BL/6, FVB or Rag1−/− mice; Shanghai SLAC Laboratory Animal Co., Ltd) were injected with $1 \times 10^6$ cells. The tumor size was measured every 4 days using calipers in two dimensions to generate a tumor volume using the following formula: $0.5 \times (length \times width^2)$. 10-week-old *Pten*[PC−/−] and *Pten*[PC−/−]; *Arid1a*[PC−/−] mice received castration. Two weeks later, Enzalutamide (MedKoo Biosciences, 201821) was dissolved in DMSO as 90 mg/ml stock solution, which was diluted in corn oil (Sigma) to make 2.5 mg/ml working solution. Mice were given 100 μl I.P. intraperitoneal injections three times a week for 2 months[66]. More details about the number and age of animals used in each experiment, see the corresponding figure legends.

### Histology quantifications
Histological features were graded using previously described nomenclature and criteria[67]. In brief, HGPIN was characterized by the intraglandular proliferation of crowding cells with atypia and cribriform formation or the development of multilayered, solid glandular structures. Adenocarcinoma displays an abnormal architectural glandular pattern with disturbance of benign epithelial-stromal relationships, indicated by the presence of atypical cells breaking the basal membrane. Quantitative histological results are derived from five random slides of each mouse in the entire animal cohort, as previously described[68,69]. In brief, for each animal, five random fields were captured, each of which was further divided into four quadrants. In each quadrant, the most advanced histological feature was recorded for quantification. Thus, the number of each lesion subtype in each experimental group was established, and the percentage of different lesion subtypes was compared.

### Expression plasmids, shRNA, sgRNA, CRISPRa, and CRISPRi
pLenti-puro-ARID1A was purchased from Addgene for overexpression. The full-length human IKKα, IKKβ, and NEMO cDNA were cloned into pcDNA5-Flag (Invitrogen), pcDNA3.1 (Myc or HA tag) (Invitrogen) or pLVX-IRES-puro (Clontech) to generate expression plasmids. ARID1A mutants were constructed as described previously. The *Arid1a* sense and antisense KD1 and KD2 oligonucleotides were annealed and cloned into pLKO.1-Puro (Addgene). pLentiCRISPRv2-sgRNAs targeting *Arid1a* gene locus were used to knock out the gene in cells. sgRNAs for CRISPRa or CRISPRi were designed by CRISPick (https://portals.broadinstitute.org/gppx/crispick/public). Briefly, dSV40-dCas9-10×GCN4-mCherry construct was generated from dSV40-dCas9-10×GCN4 (Addgene, #107310) by inserting a P2A-mCherry cassette for sorting purpose. Control and sgRNA targeting the enhancer of *A20* were cloned into sgRNA-SPH vector. The dCas9-GCN4 expressed Myc-CaP cells were infected with control and *Arid1a*-sgRNA lentivirus. CRISPR-mediated endogenous *A20* activation is achieved by SPH activator system as previously reported[70]. For CRISPRi, sgRNAs were cloned into lenti-dCas9-KRAB-puro vector (Addgene) and stably transfected into Myc-CaP cells. The siRNA, sgRNA, and shRNA sequences are listed in Supplementary Table 3.

### Infections and transfections
Lentivirus was used to establish individual stable cells, and empty vector was used as the controls for overexpression, shRNA-based knockdown and sgRNA-based knockout. $1 \times 10^5$ cells were plated into 6 well plates the day before infection. When the cells reached ~50% confluence, 2 mL of lentiviral supernatant with 10 μg/mL polybrene was added and removed 24 hr later. Infected cells were selected in 2 μg/mL puromycin or 5 μg/mL basticidin. For transfections, cells

were transfected with siRNA duplexes or plasmid by using Lipofectamine (Invitrogen) according to manufacturer's instructions. One day before transfection, cells were plated into 6 well plates. We diluted 100 μM oligonucleotide or 2 ug plasmid and 2 μl Lipofectamine respectively in 50 μl of Opti-MEM. While complexes were formed after 15–20 min, we added and incubated them with the cells at 37 °C for 4 h. Remove the medium from the cells and add 2 mL of fresh medium to each well until harvest.

### Immunostaining and IHC
ARID1A (Sigma-Aldrich; HPA005456, 1:500), CD4 (Abcam; ab183685, clone: EPR19514, 1:500), CD8 (Abcam; ab209775, clone: EPR20305, 1:200), Ly6G (BioLegend; 127601, clone: 1A8, 1:500), SMAα (Sigma-Aldrich; A2547, clone: 1A4, 1:4000), AR (Santa Cruz Biotechnology; SC816S, clone: N-20, 1:500), CK8 (Abcam; ab154301, 1:4000) and Ki67 (Cell Signaling Technology; 12202, clone: D3B5, 1:500) antibody were used for immunohistochemistry. Formalin-fixed paraffin-embedded (FFPE) tissue sections were de-paraffinized. The tissue sections were blocked first for 1 h in MOM Blocking reagent (Vector Labs; MKB-2213). Sections were then incubated sequentially with the primary antibody overnight and followed by 1 h incubation with biotinylated anti-mouse (Vector Labs; BA-9200-1.5) or rabbit (Vector Labs; BA-1000-1.5) secondary antibodies in a 1:500 dilution. The Streptavidin-HRP and DAB detection kit (Vector Labs) were used according to the manufacturer's instructions. For quantification, positive cells were counted from at least three random slides of each mouse for a total of 4–8 pairs.

### Reagent and immunotherapy
The chemicals used in vitro and in vivo assays are as follows: SB225002 (0.1 μM for the in vitro assays and 2 mg/kg for the mouse treatment, MedChemExpress; HY-16711), JSH-23 (5 μM for the in vitro assays and 2 mg/kg for the mouse treatment, MedChemExpress; HY-13982); RS504393 (0.2 μM for the in vitro assays, MedChemExpress; HY-15418); MG132 (Selleck Chemical; S2619s); cycloheximide (BioVision; 1041-1 G). SB225002 and JSH-23 in DMSO were diluted in corn oil for in vivo administration through intraperitoneal injection every other day or gavage daily, respectively. For immunotherapy, antibody intraperitoneal injection was started when subcutaneous tumor volume reached ~100 mm³ or *Pten*[PC−/−]; *Arid1a*[PC−/−] mice were 2.5-month-old. The following antibodies were injected alone or in combination: anti-mouse PD1 (BioXCell; Clone: RMP1-14, BE0146); anti-mouse CTLA4 (BioXCell; Clone: 9H10, BE0131); anti-mouse Ly6G (BioXCell; clone: 1A8, BE0075-1); and their respective isotype IgG controls. PD1/CTLA4 antibody were simultaneously administrated to the mice. Treatment was administered twice a week through intraperitoneal injections at a dosage of 200 μg/injection/antibody, and subcutaneous tumor volume was monitored every 4 days. CXCL2 was depleted from conditioned media by incubation with mouse antibody against Mip2/Cxcl2 (R&D Systems; MAB452).

### Cytokine measurement
Protein lysis of PCa specimen was used to determine CXCL2 and CXCL3 levels by single-plex sandwich ELISA kits (Elabscience; E-EL-H1904c and E-EL-H1905c). The amount of CXCL2 and CXCL3 protein in the serum and protein lysis of prostate tissues from *Pten*[PC−/−] and *Pten*[PC−/−]; *Arid1a*[PC−/−] mice was determined using mouse CXCL2 and CXCL3 specific ELISA kits (Elabscience; E-EL-M0019c and Beijing BioRab Technology Co. Ltd.; ZN2584). IFN-γ levels were determined using a single-plex sandwich ELISA (Senxiong Biotech). The assay was performed according to the manufacturer's instructions.

### PMN-MDSC isolation and transwell assay
PMN-MDSCs were isolated from prostate tumors of *Pten*[PC−/−]; *Arid1a*[PC−/−] mice sorted by FACS (CD45+ CD11b+ Ly6G+ Ly6C[low]) and plated in

RPMI1640 medium. PMN-MDSCs ($1 \times 10^5$ cells/well) were seeded in the top chamber of the transwell (Corning). Conditioned media from cultured Myc-CaP cell lines (with or without *Arid1a* knockout; with or without SB225002, JSH-23, RS504393 or CXCL2 antibody pretreatment) were collected and added to the bottom layer of the transwell. After 4 h incubation, cells that migrated to the bottom chamber were counted. These experiments were performed in triplicate, and statistical significance was assessed using two-tailed unpaired *t*-test.

## CyTOF

Tumor cells were isolated using Mouse Tumor Dissociation Kit (Miltenyi Biotec; 130-096-730) and were depleted of red blood cells using RBC Lysis Buffer (BioLegend; 420301). Cells were stimulated for 4 h at 37 °C with 10% FBS RPMI1640 medium supplemented with cell activation cocktail (BioLegend; 423303; 1:500 dilution). Cells were Fc-blocked by CD16/CD32 antibody (Biolegend; 156604) and incubated with CyTOF surface antibody cocktails for 30 min at 4 °C. For intracellular staining, cells were permeabilized using fixation/permeabilization buffer solution (BD Biosciences). Cells were washed twice and incubated with CyTOF intracellular antibody mix for 1 h at room temperature. For singlet discrimination, cells were washed and incubated with Cell-ID Intercalator-Ir (Fluidigm 201192 A) overnight at 4 °C. The samples were submitted to the Flow Cytometry and run using CyTOF Instrumentation (DVS Science), and were analyzed by FlowJo and Cytobank. Cell populations were identified as T cells (CD45$^+$ CD3e$^+$), CD4$^+$ T cells (CD45$^+$ CD3e$^+$ CD8a$^-$ CD4$^+$), CD8$^+$ T cells (CD45$^+$ CD3e$^+$ CD8a$^+$ CD4$^-$), PMN-MDSC (CD45$^+$ CD11b$^+$ F4/80$^{low}$ Ly6G$^+$ Ly6C$^{low}$), M-MDSC (CD45$^+$ CD11b$^+$ F4/80$^{low}$ Ly6G$^-$ Ly6C$^{high}$) and macrophage (CD45$^+$ CD11b$^+$ F4/80$^{high}$). CyTOF staining panels are detailed as follows: Ly6G (conjugated to 141Pr, DVS-Fluidigm; clone: 1A8, 3141008B), CD4 (conjugated to 145Nd, DVS-Fluidigm; clone: RM4-5, 3145002B), CD8a (conjugated to 146Nd, DVS-Fluidigm; clone: 53-6.7, 3146003B), CD45 (conjugated to 147Sm, DVS-Fluidigm; clone: 30-F11, 3147003B), CD11b (conjugated to 148Nd, DVS-Fluidigm; clone: M1/70, 3148003B), CD3e (conjugated to 152Sm, DVS-Fluidigm; clone: 145-2C11, 3152004B), F4/80 (conjugated to 159Tb, DVS-Fluidigm; clone: BM8, 3159009B), Ly6C (conjugated to 162Dy, DVS-Fluidigm; clone: HK1.4, 3162014B) IFNγ (conjugated to 165Ho, DVS-Fluidigm; clone: XMG1.2, 3165003B).

## Lymphocyte staining and flow cytometry

Tissues were dissected, minced into small pieces and further digested by 1 mg/ml Collagenase Type II (Thermo Fisher Scientific; 17101015), 1 mg/ml Collagenase Type IV (Thermo Fisher Scientific; 17104019) and 0.1 mg/ml DNase I recombinant (Sigma-Aldrich; 4536282001) at 37 °C for 30–60 min. One million cells were incubated with 1 μl CD16/CD32 antibody (Biolegend; 156604) to block the Fc receptor at 4 °C for 10 min. Cell suspension were incubated with cell surface antibodies at 4 °C for 30 min. For intracellular cytokine staining, cells were stimulated for 4 h at 37 °C with cell activation cocktail (BioLegend; 423303). After permeabilized with FOXP3 Fixation/Permeabilization Buffer (eBioscience; eBio 00-5523) according to the manufacturer's protocol, cell suspension was incubated with IFNγ, FOXP3, and Ki67 antibody at 4 °C for 30 min. Then, cells were analyzed on a Gallios analyzer (Beckman Coulter Life Sciences). Data were analyzed with FlowJo v.10 (FlowJo LLC). All FACS antibodies were used in a dilution of 1:100. F4/80 (BV510, BioLegend; clone: BM8, 123135), CD11b (BV605, BioLegend; clone: M1/70, 101237), Ly6G (PE-Cy7, Bio-Legend; clone: 1A8, 127617), Ly6C (APC, BioLegend; clone: HK1.4, 128016), CD45 (APC-Cy7, BioLegend; clone: 30-F11, 103116), CD4 (PE-Cy7, BioLegend; clone: GK1.5, 100422), CD8 (AF700, BioLegend; clone: 53-6.7, 100730), FOXP3(APC, eBioscience; clone: 236 A/E7, 17-4777), Ki67 (BV421, BioLegend; clone: 16A8, 652411) and IFNγ (FITC, BioLegend; clone: XMG1.2, 505806).

## T cell suppression assay

PMN-MDSCs were isolated from 3-month-old *Pten*$^{PC-/-}$; *Arid1a*$^{PC-/-}$ mice prostate tumors. CD8$^+$ T cells were isolated from spleen of wild-type C57BL/6 mice. A T cell suppression assay was performed using PMN-MDSCs sorted by FACS (CD45$^+$ CD11b$^+$ Ly6G$^+$ Ly6C$^{low}$) and CFSE (Invitrogen)-labeled MACS-sorted (Miltenyi Biotec; 130-104-075) CD8$^+$ T cells in anti-CD3- and anti-CD28-coated 96-well plates at an MDSC/ T cell ratio of 0:4, 1:4, with $1 \times 10^5$ PMN-MDSCs. CFSE intensity was quantified 96 h later with peaks identified by FACS. CFSE peaks indicated the division times. Division times 0–2 and 3–4 were defined as low proliferation and high proliferation, respectively.

## RNA isolation and real-time PCR

Total RNA was extracted using TRIzol (Invitrogen) according to the manufacturer's instructions. First-strand cDNA was synthesized by HiScript II Q RT SuperMix (Vazyme) for qPCR. Real-time PCR was performed with FastStart Universal SYBR Green Master (Roche) on QuantStudio 7 Flex Real Real-Time PCR System (Applied Biosystems). The primers used for real-time PCR are shown Supplementary Table 3. The $2^{-\Delta\Delta Ct}$ method was used to calculate relative expression changes.

## IP and IB analysis

For IP assays, cells were lysed and washed in HEPES lysis buffer (20 mM HEPES, pH 7.4, 200 mM NaCl, 1.5 mM MgCl$_2$, 2 mM EGTA, 0.5% NP-40, 1 mM NaF, 1 mM Na$_3$VO$_4$ and 1 mM PMSF) supplemented with protease-inhibitor cocktail (Roche). Cell lysates were incubated overnight at 4 °C with indicated primary antibody and protein A/G agarose beads (Roche) or anti-flag M2 agarose (Sigma). Beads were centrifuged at 1000 g for 5 min at 4 °C to remove the supernatant, washed four times with the IP buffer and boiled SDS-loading buffer for 10 min at 95 °C. Samples were run on SDS-PAGE gel analyzed by western blotting. Full scan blots, see the Source Data file.

## Chromatin-immunoprecipitation assays

The ChIP assays were performed using ChIP kit (Catalog no. 17-371; Millipore). The procedure was according to the kit instruction manual provided by the manufacturer. Briefly, $1 \times 10^7$ Myc-CaP cells were fixed by 1% formaldehyde, fragmented by sonication to shear the chromatin to 400–1000 bp. The sheared crosslinked chromatin was incubated with IgG, anti-BRG1 (Abcam, clone: EPNCIR111A, ab110641), anti-BAF155 (Santa Cruz Biotechnology; clone: G-7, sc-365543X), anti-ARID1A (Cell Signaling Technology; clone: D2A8U, 12354 S) and anti-ARID1B (Cell Signaling Technology; clone: E9J4T, 92964 S) antibodies (10 μg antibody for each ChIP reaction) overnight followed by Protein G conjugated agarose beads incubation. The precipitated DNA was amplified by primers and quantified by QuantStudio 7 Flex Real Real-Time PCR System (Applied Biosystems). ChIP primer sequences can be found in the Supplementary Table 3.

## Enhancer RNA (eRNA) assay

According to the manufacturer's guidelines (Cell-Light TM EU Nascent RNA Capture Kit; RiboBio), nascent RNA was labeled in WT and *Arid1a* KO Myc-CaP cells by ethynyl-labeled uridine (EU). Subsequently, the resulting EU-labeled RNA was detected via Cu (I)-catalyzed click chemistry that introduced a Biotin tag for RNA purification. At last, streptavidin-purified RNA was applied to reverse transcriptase-mediated cDNA synthesis and further qPCR analysis.

## In vitro kinase assay

Recombinant proteins of Flag-ARID1A (substrate) and HA-IKKβ (enzyme) were first prepared and purified from E. coli. In a typical phosphorylation reaction, 1 μg Flag-ARID1A protein was incubated with HA-IKKβ in a 50 μl kinase reaction buffer (50 mM Tris-HCl, 5 mM MgCl2, 30 μM ATP) at 37 °C for 1 h. Phosphorylation of ARID1A was analyzed by western blotting with a thiophosphate ester rabbit

monoclonal antibody (Abcam; ab92570). Full scan blots, see the Source Data file.

## Gel filtration

Gel-filtration chromatography analyses were performed with Superose 6 Increase 10/300 GL columns (GE Healthcare; 29-0915-96). Briefly, C4-2 cells were lysed in 1 mL Hypotonic Buffer containing protease inhibitors (Roche). Nuclear pellets were collected and washed twice with Hypotonic Buffer, and then lysed in HEPES lysis buffer. After lysing for 20 min on ice, cell debris were removed and the nuclear extraction was further filtered through a 0.45 mm syringe filter before loaded onto a Superose 6 Increase 10/300 GL columns. A total of 20 µl samples from each fraction were analyzed by IB using indicated antibodies. Full scan blots, see the Source Data file.

## GST pull-down assay

Full length HA-IKKβ protein was obtained from in-vitro translation kit (catalog no. L1170; Promega). BL21 E. coli transformed with empty vector or pGEX-GST-ARID1A-F2 truncated plasmid was induced by Isopropyl-1-thio-β-D-galactopyranoside (IPTG) (0.5 mM) at 16 °C for 5 h. After sonication, incubate the cell lysate with glutathione-Sepharose beads for 1 h at 4 °C. Recombinant proteins binding to beads were incubated with HA-IKKβ protein for 3 h at 4 °C. Beads were subsequently harvested through centrifugation and washed three times by 0.2% NP-40 buffer before boiled and loaded to SDS-PAGE. Full scan blots, see the Source Data file.

## Organoid generation

The mouse prostatic organoid formation assay was performed according to previously reported[63,64]. Mouse prostate tissue was minced and digested in 5 mg/ml Collagenase type II (Life Technologies;17101-015) with 10 µM Y-27632 dihydrochloride and incubated at 37 °C for 1 h. The dissociated tissue pellets were resuspended in TrypLE (Gibco; 12605-010) with 10 µM Y-27632 for further digestion for 15 min at 37 °C. The dissociated cell suspensions were stained for 30 min on ice with the following antibody: CD24-FITC (BioLegend; clone: M1/69, 101806), PE conjugated CD49f (eBioscience; clone: GoH3, 12-0495-82) and DAPI (1 ng/µl, Sigma; D8417). For organoids formation assay, cells were suspended using organoid medium, and mixed with Matrigel (1:1). The cell suspension was seeded into 24 well culture plate with the cell number of 2000 in a volume of 50 µl per well. The number and size of the organoids were determined on day 9.

## RNA-seq and analysis

Cells were isolated from 3-month-old $Pten^{PC-/-}$ and $Pten^{PC-/-}$; $Arid1a^{PC-/-}$ mice ($n = 3$) through isolating EpCAM$^+$; CD45$^-$ cells. For subcutaneous tumors, Myc-CaP expressing sgARID1A and control vectors inoculated subcutaneously into FVB mice and treated with or without JSH-23 and epithelial cells were sorted for RNA-seq. Total RNA was extracted using TRIzol reagent (Invitrogen), and then subjected for sequencing by RiboBio (Guangzhou, China). Samples were demultiplexed into paired-end reads using Illumina's bcl2fastq conversion software v2.20. The reference genome was indexed using bowtie2-build, and reads were aligned onto the GRCm38/mm10 mouse reference genome using TopHat2 with strand-specificity and allowing only for the best match for each read. The aligned file was used to calculate strand-specific read count for each gene using HTSeq-count (version 0.13.5). The significance was operated by setting fold changes threshold at level of 1.5 and $p < 0.05$. Heatmaps were generated using the pheatmap (1.0.12) package in R (4.2.0). For gene enrichment analysis (GSEA), The ZHANG_ RESPONSE_TO_IKK_INHIBITOR_AND_TNF_UP (223 genes) and HALLMARK_ TNFA_SIGNALING_VIA_NF-κB (200 genes) signature were derived from GSEA C2: curated gene sets and hallmark gene sets. Similar transcriptome sequencing and analysis were also performed by WT and Arid1a KO Myc-CaP cells. Weighted GSEA enrichment

statistic and Signal2Noise or Diff_of_Classes metric for ranking genes were used.

## ChIP-seq and data analysis

Chromatin-immunoprecipitation experiments were carried out by WT and Arid1a KO Myc-CaP cells using the ChIP kit (Catalog no. 17-371; Millipore). Chromatin from $3 \times 10^6$ cells was used for each ChIP reaction with 10 µg of the target protein antibodies: anti-BRG1 (Abcam, clone: EPNCIR111A, ab110641), anti-H3K27ac (Cell Signaling Technology; clone: D5E4, 8173 S), anti-H3K4me1 (Cell Signaling Technology; clone: D1A9, 5326) and anti-H3K4me3 (Cell Signaling Technology; clone: C42D8, 9751 S). Purified DNA was then prepared for sequencing (Illumina). Libraries were quantified and sequenced on the Illumina HiSeq 2500 Sequencer (125-nucleotide read length). Reads mapped to the same genomic positions were filtered by PICARD-MarkDuplicates (Galaxy Version 2.18.2.2), and the nonredundant reads were used for peak calling. MACS2-callpeak (Galaxy Version 2.1.1.20160309.6) was used for performing peak calling with the threshold, $p$-value $\leq 0.005$. ChIP peak profile plots and read-density heat maps were generated using deepTools, and cistrome overlap analyses were carried out using the ChIPseeker (Galaxy Version 1.18.0). Galaxy available pipelines were utilized for analysis (https://usegalaxy.org/).

## ATAC-seq and data analysis

50,000 Myc-CaP cells with or without Arid1a KO were suspended in cytoplasmic lysis buffer (CER-I from the NE-PER kit, Invitrogen; 78833). Nuclei were resuspended in 50 µl of 1× TD buffer, then incubated with 2–2.5 µl Tn5 enzyme for 30 min at 37 °C (Nextera DNA Library Preparation Kit; FC-121-1031). Samples were purified and PCR-amplified with the NEBNext High-Fidelity 2X PCR Master Mix (NEB; M0541L). ATAC-seq libraries were sequenced on the Illumina HiSeq 2500 (125-nucleotide read length, paired end). Paired-end fastq files were trimmed and uniquely aligned to the GRCm38/mm10 mouse genome assembly using BWA. Duplicated reads were removed by PICARD-MarkDuplicates (Galaxy Version 2.18.2.2). Filtered bam files were used for peak calling by MACS2 and an initial threshold q-value of 0.01 as cutoff. Bigwig files were then visualized using the UCSC genome browser, and the final figures were assembled using Adobe Illustrator.

## Analysis of ARID1A, NF-κB, and MDSC signature in human PCa patients

Analysis in human tumor datasets was carried out as previously described[7,68,69]. Differentially expressed genes between epithelial cells of 3-month-old $Pten^{PC-/-}$; $Arid1a^{PC-/-}$ versus $Pten^{PC-/-}$ mouse prostates were defined as the Arid1a signature. The NF-κB signature was derived from TIAN_TNF_SIGNALING_VIA_NFKB (26 genes), SCHOEN_SCHOENNFKB_SIGNALING (34 genes), ZHANG_RESPONSE_TO _IKK_INHIBITOR_AND_TNF_UP (223 genes) and HALLMARK_TNFA_SIGNALING_ VIA_NFKB (200 genes) from GSEA C2: curated gene sets (the former 3) and hallmark gene sets. MDSC signature (39 genes) genes were generated as previously described[16]. To define the degree of Arid1a signature manifestation within the profiles from an external human tumor dataset, we used the previously described t-score metric[7,69]. For example, the t-score was defined for each external profile as the two-sided t-statistic comparing the average of the Arid1a-induced genes with the average of the Arid1a-repressed genes (For computing gene signature scores based on expression profile data from human dataset, genes were first z-normalized to the SD from the median across the dataset). The t-score contrasted the patterns of the "Arid1a-induced" genes against those of the "Arid1a-repressed" genes which did not include Arid1a mRNA itself, to derive a single value denoting coordinate expression of the two gene sets. NF-κB and MDSC signature scores were calculated using the method as previously described[16]. Specifically, we used the ssGSEA algorithm (GSVA Ver_1.45.5) to assign an enrichment score of genes in each gene

list above for each sample. Higher ssGSEA scores correspond to more joint upregulation of genes in each signature.

## Statistical analysis

GraphPad Prism 8.0 was used for statistical calculations. For all comparisons between two groups of independent datasets, two-tailed unpaired *t*-test was performed, *p*-value and standard error of the mean (SEM) were reported. For comparisons among more than two groups (>2), one-way or two-way ANOVA followed by multiple comparison were performed, *p*-values and SEM were reported; and *p*-values were adjusted by multiple testing corrections (Bonferroni) when applicable. For quantification of tumor cell or immune cell density, images of tumor sections with IF or IHC staining were captured by using microscope. The positive cells were counted. Three fields in each were randomly selected for tumor cell or immune cell density analysis and statistical analysis was performed by using *t*-test. The two-tailed Pearson correlation between ARID1A expression/signature and NF-κB activity, MDSC signature were calculated using Graphpad Prism 8.0, and difference of proportion were determined by two-tailed Fisher's exact test. Patient recurrence was determined by Kaplan–Meier analysis. Statistical testing was performed with the log-rank test. For genetically engineered mouse modes (GEMMs) analysis, the examinations were performed dependent on animal available. All the experiments were independently repeated at least three times (with at least 5 biological repeats in total) with the similar time course and treatment. The xenograft assays were results of one-time experiment with sufficient animal number indicated in figure legend. In all figures, not significant (ns), $p < 0.05$ (∗) and $p < 0.01$ (∗∗).

## Reporting summary

Further information on research design is available in the Nature Portfolio Reporting Summary linked to this article.

# Data availability

All RNA-seq, ChIP-seq and ATAC-seq data generated during this study have been deposited in the Gene Expression Omnibus (GEO) database under accession numbers GSE197688. Published datasets used in this study are available through GEO (GSE21032 or cBioPortal database (SU2C/PCF Dream Team, PNAS 2019, https://www.cbioportal.org/)). The remaining data are available within the Article, Supplementary Information or Source Data file. Source data are provided with this paper.

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

## Acknowledgements

This study was supported by grants from the National Key Research and Development Program of China (2021YFA1300601, J.Q.; 2018YFA0902700, J.Q.), the National Natural Science Foundation of China Projects (81825018, J.Q.; 82130085, J.Q.; 81773121, J.Q.; 81802818, N.L.; 81974395, H.H. and 82173036, H.H.), Shanghai Pilot Program for Basic Research-Chinese Academy of Science, Shanghai Branch (JCYJ-SHFY-2022-007, J.Q.), and the Chinese Academy of Sciences (QYZDB-SSW-SMC052, J.Q.). J.Q. is sponsored by Program of Shanghai Academic/Technology Research Leader (19XD1424300). N.L. is supported by the Initiative Postdocs Supporting Program by MOHRSS and the National Postdoc Management Committee (bx201800247).

## Author contributions

J.Q., H.H., J.J. and N.L. designed the experiment and prepared the manuscript. N.L., Qiu.L., Y.H. and S.P. performed most experiments. N.L. and Q.L. contributed to the computational statistical analysis. Q.L. and J.J. performed the TMA and pathology analyses. B.C., J.X., X.M., Q.P., H.W., J.G., Xu.W., G.Z., Y.L., W.Z. and Y.Z. performed a specific subset of the experiments and analyses, which were supervised by M.T., Qin.L., Xi.W., Y.X., G.H., J.J., and H.H.

## Competing interests

The authors declare no competing interests.
