## [Peer Review File · Nature Communications]

ARID1A loss induces polymorphonuclear myeloid-derived suppressor cell chemotaxis and promotes prostate cancer progressionREVIEWER COMMENTS

Reviewer #1 (Remarks to the Author):

In the paper titled "ARID1A loss shapes the immunosuppressive tumor microenvironment via NF- κ B-induced MDSC chemotaxis to promote prostate cancer metastasis" the authors convincingly demonstrate that loss of ARID1A in a Pten genetic background leads to marked increase in tumour growth due to an immunosuppressive phenotype. The studies are well controlled and technically impressive.

Major comments:

- All in vivo work was done in a Pten null genetic background. Given the clear increase in tumour growth the Pten status in their clinical cohort (Fig 7) should be determined to see if there is a correlation in patient ARID1A protein expression. This is particularly important as it is unclear if ARID1A degradation alone is sufficient to drive transformation.
- Figure 2b: Using Cytof the authors demonstrate that there is an increase in polymorphonuclear-MDSCs. Greater information is needed how the authors separated neutrophils from MDSCs particularly given that anti-Gr-1 treatment will also impact neutrophils (Fig 2h).
- To identify potential drivers of ARID1A-mediated immunosuppressive activity (Figure 4f) the authors overlaps the open chromatin regions with reduced BRG1/SMARCA4 peaks and down-regulated genes. Given the proposed mechanism via enhancer reprogramming, greater explanation of how the specific enhancer-gene promoters were identified is needed.
- Figure 4i should be removed as it largely duplicates Figure 4g and visualization of eRNA in the track is difficult.
- Pg 18, line 2-4: It is stated that ARID1A-OE tumors exhibit increased sensitivity to anti-PD1 therapy. However, the overexpressing tumors also have reduced growth. More rigorous statistical testing is needed to confirm the proposed synergy.
- Pg 18, line 11-12: The authors claim that "cotargeted treatment (of NF- κ B inhibitor and/or anti-PD1/CTLA4) provoked the regression of PtenPC $^{-/-}$;12 Arid1aPC $^{-/-}$ tumors.". While an intriguing concept the data is somewhat misleading as the experiment (fig 7f) cannot separate the impact of NF- κ B inhibition alone. This is particularly important as the authors demonstrated that inhibiting this pathway has a profound impact on tumour growth (Fig 3i). It is therefore unclear if the reduced adenoma observed is due to a synergism with anti-PD1/CTLA4 or simply JSH-23 treatment alone.

Minor comments:

- Figure 2A - Needs an x-axis label
- Figure 3 I - Needs an x-axis label
- Figure 4D - Needs samples labels (red/black)
- Page 12, Line 8 /throughout - Please change the Open Chromatin Region (OCR) to the more common convention cis-regulatory element (CRE)
- Pg 13, line 18 - The extended Data Fig. 4e does not show a reduction in H3K27Ac.

Reviewer #2 (Remarks to the Author):

In the present manuscript, Yan et al. investigate the loss of ARID1A, a subunit of the SWI/SNF chromatin remodeling complex, as a new mechanism by which the prostate tumour microenvironment recruits immunosuppressive MDSCs. The authors generated and employed a new transgenic mouse model (PtenPC $^{-/-}$; Arid1aPC $^{-/-}$) to prove their hypothesis. They showed that loss of ARID1A in a Pten loss background promotes metastasis by enhancing the recruitment of MDSCs via NF κ B in the tumour microenvironment. In line with their hypothesis, the authors demonstrate by using available datasets (TGCA) that ARID1A signature opposite correlates with MDSC and NF κ B signature and has a worst prognosis in prostate cancer patients.

This is a timely study: immunotherapy has not reached a satisfactory degree of clinical efficacy in prostate cancer despite many years of research and clinical development. One of the reasons is that the tumour microenvironment of prostate cancers is highly immunosuppressive due to the infiltration of MDSCs.

Many groups worldwide are now exploring new therapies aimed to block the tumour recruitment of these cells. Linking tumour epigenetic changes with a higher immune suppressive microenvironment that drive metastasis is interesting, even if not fully novel. The effects of ARID1A loss in SWI/SNF binding and on the epigenetic machinery (H3K4me3/H3K27ac) (10.1038/ng.3744) and the effect of NFkB activation and MDSC recruitment have already been characterized. Additionally, the frequency of ARID1A mutation in prostate cancer patients is very low, thus the overall clinical relevance remains limited.

Major concerns:

- The manuscript seeks accurate histopathological analysis and tumour measure of the Pten; Arid1a-ko tumours. Measurements of all the prostate lobes (Volume) and clearer representative immunohistochemistry images with higher resolution and magnification are required. Evidence of metastasis needs to be better demonstrated. Are the mice also collecting metastasis at earlier times, such as 12 weeks old? Different time-points and markers (e.g. PanCK) need to be included.
- The characterization of the tumour development in PtenPC^{-/-};Arid1aPC^{-/-} mice has been performed at 16 weeks old (Figure 1). However, the characterization of the tumour microenvironment (Figure 2) reports data collected from mice 12 weeks old. What is the disease stage at that time point? The analyses need to be coherent.
- The manuscript does not clarify which factors are differentially expressed by the prostate epithelium of Pten; Arid1a-ko vs Pten-ko tumours. This is a significant limitation of the study. The authors should perform a gene expression profiling of cytokines and cytokine-related factors in Pten; Arid1a-ko vs Pten-ko tumours to demonstrate changes in the recruiters of MDSCs. Reporting results of a few cytokines by qPCR (Figure 3a) is limited to having unbiased transcriptomic profiling of the mouse models (Figure 2a). Additionally, evidence regarding the soluble factors produced by the ARID1A-deficient epithelial tissues is required. Are they accumulating also in the serum? What about the cytokine profile of tumour cells upon NFkB inhibition? Is it equal to Arid1a genetic inhibition?
- A big limitation of the study is the absence of a clear demonstration of the endocrine status of the Pten; Arid1a-ko tumours. Are they hormone-dependent or independent, meaning do they respond or not to castration?

Minor Points:

- The manuscript needs to be carefully checked for typos, and not all the panels are cited in the text.
- In figure 2b, the reported graph is not clear. The author should display the quantification of the immune population identified by the tSNE analysis more comprehensibly.

Reviewer #3 (Remarks to the Author):

Li et al. propose an intriguing hypothesis that ARID1A loss in conjunction with PTEN loss plays a role in prostate cancer immunosuppression via polymorphonuclear myeloid-derived suppressor cell (MDSC). They suggest this is mediated through CXCR2 ligand-mediated MDSC chemotaxis. At the outset, the authors propose a major role for inflammation and TME that promote prostate cancer (PCa) metastasis and diminished response to checkpoint therapy. There have been several high-profile papers implicating the role of MDSC in advanced PCa.

Comments

1) ARID1A mutations are suggested to act as tumor suppressors given that the majority are missense or frameshift mutations. The low percentage of mutations in localized PCa (TCGA) and advanced PCA (Cytra et al., Nature Communications), suggest that there would need to be other mechanisms for regulation of ARID1A if this plays a significant role in human PCa. What evidence do the authors have for dysregulation in human prostate cancer?

2) The authors demonstrate ARID1a expression as decreasing in higher Gleason grade localized PCa. Is this consistent with a model that leads to increased MDSC? MDSC are not seen at high levels in localized PCa?

3) The authors show that depletion of PTEN alone and in concert with depletion of TP53 correlates with downregulation with ARID1A in vivo. From this data (Figure 1a and the following) it remains unclear if the depletion of these genes is inducing ARID1A downregulation or if these tumors are generally more inflamed? Looking at total IKKb, it seems that this increases in the same manner as the p-IKKb (Fig 1A), therefore quantifying these bands might help to understand if phosphorylation is really increased and involved in the downregulation of ARID1A. It would be helpful to understand this phenomenon if this could be elaborated a bit more, is there a higher burden of cytokines and/or MDSCs in the PTEN depleted and, especially PTEN/TP53 depleted setting vs WT? Why is the depletion of ARID1A necessary to show results in Fig.2? Fig1. Showed complete downregulation of ARID1A in the PTEN/p53 depleted setting.

4) Especially, the effect with TP53 depletion seems to strongly affect ARID1A expression, however, from there onwards the authors seem to ignore their double knockout model. For example, experiments performed in Fig 3. C would be interesting in the context of p53 deletion or using cell lines that harbor p53 mutations such as DU145.

5) This study focuses highly on the microenvironmental aspects of PCa. How does the loss of ARID1A affect SWI/SNF composition in the functional mouse model PCa components? Ding et al. (PMID: 3049614) have shown that PTEN stabilizes BRG1 expression via the AKT/GSK3b/FBXW7 axis, could this be true for ARID1A also?

6) One intriguing question that comes up when knocking out ARID1A is what happens to the SWI/SNF complexes that are remaining? Is 1B elevated? What happens to the other complex members. BRG1 has been shown to increase in advanced PCa. As the authors perform BRG1 ChIP, it would be important to know if BRG1 and perhaps other markers of lineage plasticity are altered with ARID1A KO including BAF53b.

7) A recent study by Wellinger et al. (PMID: 34782346) has shown that TNF-mediated apoptosis can be potentiated by BET inhibition in cancer cells. Can this be exploited in the context of PCa as well? The effect on cancer cells and immune cells would be very interesting, especially in the ARID1a depleted setting.

8) Some of the ARID1A low tumor specimens do not show the postulated trend of increased IKKb and p65 expression (Fig 7. A). What could be the possible explanation that ARID1A is downregulated here?

9) Does the inflammatory environment also influence ARID1A expression of the immune compartment/surrounding tissue or is ARID1A safeguarded by intact PTEN/TP53 (coming back to the question: is the loss of tumor suppressors directly promoting this phenomenon or is it promoting the inflammation? Is this a PCa-specific phenomenon? If yes, why do we not see downregulation or mutations of ARID1A more often in PCa, since PTEN is among the most frequently mutated genes in PCa?

Minor Comments

1) The authors use the term "adenoma" throughout the manuscript. What is meant by this? This term is not usually used for PCa.

Reviewer #4 (Remarks to the Author):

In this manuscript the authors identify a new molecular network that accounts for increased recruitment of MDSC and consequent immune suppression in prostate cancer, with implications for tumor progression and resistance to immune checkpoint therapies. The authors adopt a series of in vitro and in vivo experiments, in mouse models, and analyses on patient-derived tumor specimens and data sets. They elegantly demonstrate that ARID1A, a member of the SWI/SNF1

complex, silences the expression of chemokines, including CXCL2 that promote the recruitment of MDSC in the tumor. Indeed, downregulation of ARID1A results in increased CXCL2 production via NF- κ B mediated signaling. Investigating deeply on the molecular mechanisms of this process they found that inflammatory cytokines, mainly TNF α , are responsible for ARID1A downregulation via IKK β activity. Clinical relevance is given by the observation that ARID1A expression is low in human prostate cancer and correlates with prognosis. However, the manuscript has several inconsistencies that need to be fixed:

1) Page 5 line 20: "The results showing that patients bearing low ARID1A expression had adverse disease outcomes". Please specify that this conclusion arises from results show in the manuscript.

2) Figure 1b shows correlation of ARID1A expression with clinical outcome in the cohort of patients of the tissue microarray used in the work. Translational relevance of results would benefit if this kind of analyses would be extended to deposited patient-derived data sets, as those used in figure 7 for correlation of ARID1A expression with NF- κ B signaling and MDSC infiltration.

3) The combination of immune checkpoint inhibition and a drug against NF- κ B is shown effective in prostate cancer bearing mice in figure 7f/g. However, NF- κ B signaling is involved in a plethora of different pathways, therefore the shown effect could be explained by other mechanism rather than the one described in the manuscript. The authors should provide evidence that the levels of CXCL2 are changed in prostates upon the treatment.

4) As said above, targeting of NF- κ B in vivo could potentially impact on several cellular pathways, thus leading to potential off targeting or side effects. Are there any NF- κ B inhibiting drugs used in the clinic? These issues should be at least discussed.

5) Figure 7f. Are anti PD1/CTLA4 given simultaneously in these mice? Please justify this treatment regimen, also in light of the known toxicity induced by this combination in patients.

6) Also, please clearly state in the text commenting Figure 7f that ICB is given in combination with a drug targeting NF- κ B. This is not mentioned, except for an unclear referring to "cotargeted treatment" (page 17 line 11).

7) In Figures 1b, 7b and 7d, how was the cut off of IHC signal intensity determined to divide the patients on the basis of ARID1A, P65, CD15 or CD8 staining?

8) Ly6G and Ly6C suffice to distinguish PMN-MDSC and M-MDSC. Why did authors include also Gr1 antibody in the stainings (both CyTOF and conventional flow cytometry)? As Gr1 recognizes both Ly6G and Ly6 do the three antibodies compete for binding to their ligands? It is reported in the Biolegend website (vendor of the antibodies used in this manuscript) that "Clone RB6-8C5 impairs the binding of anti-mouse Ly-6G clone 1A8" (<https://www.biolegend.com/en-gb/products/pe-cyanine7-anti-mouse-ly-6g-antibody-6139>)

9) According to authors' results showing a specific increase of PMN-MDSC in tumors lacking ARID1A (Figure 2b), IHC and in vivo experiments with neutralizing antibody should be performed with the 1A8 clone (specifically targeting the PMN-MDSC marker Ly6G).

10) Figure 2c, 3J, 6h, and extended data figure 2a, 2e, 2h, 4g and 6c: are these MDSC PMN- or M-MDSC? Please also show the manual gating strategies used to identify all the different sub-populations in flow cytometry experiments.

11) Figure 1i: "tumor cells with a luminal origin (AR+ cells, arrow) were found to invade the stromal  compartment". As also stroma cells can express AR, to be sure that they are infiltrating tumor cells a counterstaining with CK8 should be performed.

12) Page 8 line 16. "led to profound reduction in total and cytotoxic T cells". IFN γ is not a marker for cytotoxicity in CD8 T cells. Also, in the histogram of panel 2b CD8+IFN γ + cells seem negligible in both groups of mice.

13) Figure 2e. Results in Rag^{-/-} mice are paradoxically as they suggest that in absence of T cells, tumor or innate immune cell mediated mechanisms account for a negative role of ARID1A loss on tumor growth, which is the opposite of what shown in full competent mice or patients. This paradoxical effect should at least be discussed.

14) Figure 3f: evaluation of Cxcl2 and Cxcl3 levels should be performed also on murine (MyC-CaP) and not only human cell lines. Extended figure 3e: Cxcl9 and Cxcl10 should be tested also after TNF α stimulation, because IFN does not alter ARID1A levels in tumor cells.

15) Extended data figure 5f is supposed to show localization of IKKB and ARID1A by immunofluorescence, but only the green IKK β staining is visible. Please reconcile. Add separate channels if necessary to better appreciate the staining.

16) I would suggest removing the word "metastasis" from the tile and similarly toning down conclusions on metastases in the manuscript. Indeed, apart from results shown figure 1J, all the reported experiments in mice and patient-derived specimens analyze primary tumors.

17) MDSC usually are very rare in WT mice, especially in the BM. Also, the authors do not show purification results, nor a proof that those cells are true MDSC (expression of suppressive genes or suppressive assays). Indeed, as they are from WT mice they would be more likely been neutrophils or monocytes. The best would have been to purify MDSC from tumor bearing mice.

Minor points:

18) It is not clear how many times in vivo experiments were repeated.

19) Please reconcile the number of mice used in the experiment show in Figure 1c and 1d.

20) Panel 1g is not cited in the text.

21) Figure 2b the histogram is not clear. Please clearly indicate the two types of mice displayed in the two sides of the graph.

22) Figure 2b-2c and extended figure 2a. It is not clear how many mice/group were used for CyTOF or for conventional flow cytometry. Does panel 2c show CyTOF or flow cytometry data?

23) Page 16 line 19. In commenting figure 7c, AIRD1A is reported to be negatively correlated with NF-kB signatures. Please correct.

24) Figure 7e. Which cells were used for the in vivo experiment?

25) Insufficient methods are reported for infections and transfections.

Point-by-point Response to the reviewers' comments

Reviewer #1 (Remarks to the Author):

In the paper titled "ARID1A loss shapes the immunosuppressive tumor microenvironment via NF- κ B-induced MDSC chemotaxis to promote prostate cancer metastasis" the authors convincingly demonstrate that loss of ARID1A in a Pten genetic background leads to marked increase in tumour growth due to an immunosuppressive phenotype. The studies are well controlled and technically impressive.

Response: We thank the reviewer for the positive comments, and the insightful suggestions to strengthen our studies. As such, we performed additional experiments and revised the manuscript to improve the clarity.

Major comments:

1) All in vivo work was done in a Pten null genetic background. Given the clear increase in tumour growth the Pten status in their clinical cohort (Fig 7) should be determined to see if there is a correlation in patient ARID1A protein expression. This is particularly important as it is unclear if ARID1A degradation alone is sufficient to drive transformation.

Response: We thank the reviewer for the constructive comment. Loss of PTEN and activation of the PI3K/AKT signaling pathway are frequently observed in advanced PCa^{1,2,3}. Our results indicated *Arid1a* deleted mice developed hyperplasia or multifocal low-grade PIN (LGPIN) with low penetrance (**Fig. S1e**). While *Arid1a* loss cooperated with *Pten* deletion to produce metastasis-prone tumors (**Fig. 1c-j and Fig. S1g-h**). Per your comment, we investigated the correlation between ARID1A and PTEN level in a tumor tissue microarray. A positive correlation between PTEN and ARID1A level supported our mouse model was clinically relevant (**Fig. S7c**). Nevertheless, we ruled out the possibility that PTEN or P53 directly regulate ARID1A level, showing that neither knockdown *PTEN* or *P53* in PCa cells affected ARID1A expression (**Fig. S5a, b**). We have integrated the clinical association between PTEN and ARID1A in this revision.

S7c, IHC analysis for ARID1A and PTEN (left). Scale bars, 50 μ m. The correlation between PTEN and ARID1A expression is shown as stacked columns (right, n = 100). Statistical significance was determined by Fisher's exact test. **S5a**, IB analysis of C4-2 and DU145 cells with *TP53* or *Pten* KD respectively. **S5b**, IB analysis of 22RV-1 cells with *TP53* an/or *Pten* KD.

Experiments were repeated three times independently with similar results; data from one representative experiment are shown.

2) *Figure 2b*: Using *Cytof* the authors demonstrate that there is an increase in polymorphonuclear-MDSCs. Greater information is needed how the authors separated neutrophils from MDSCs particularly given that anti-*Gr-1* treatment will also impact neutrophils (*Fig 2h*).

Response: We thank the reviewer for pointing out this important issue. Neutrophils and PMN-MDSC share origin and many morphological and phenotypic features⁴. However, they have different biological roles. Neutrophils are one of the major mechanisms of protection against invading pathogens, whereas PMN-MDSCs have immune suppressive activity and restrict immune responses in cancer^{5, 6}. In mice, both neutrophils and PMN-MDSC are characterized as CD11b⁺ Ly6G⁺ Ly6C^{low}. However, expression profile analysis revealed a clear difference between PMN-MDSC from tumor-bearing mice and neutrophils from tumor-free mice^{5, 7}. Thus, we compared gene expressions of Ly6G⁺ cells in peripheral blood of WT mice and *Pten*^{PC-/-}; *Arid1a*^{PC-/-} tumors. We showed that Ly6G⁺ cells in tumors exhibited higher level of immunosuppressive genes such as *Nos2*, *Arg1*, *Nox2*, *S100a9* manifested the features of PMN-MDSCs, whereas the Ly6G⁺ cells in peripheral blood predominantly expressed the inflammatory related genes including *Tnf*, *Il6*, *Cxcl4* and among others (**Fig. S2c**). In addition, standard T cell proliferation co-culture assay showed that Ly6G⁺ cells isolated from tumors strongly suppressed CD3 and CD28 antibody-induced T cell proliferation and activation, establishing that they are functional PMN-MDSCs (**Fig. S2d, e**). Likewise, treatment with anti-Ly6G antibody depleted PMN-MDSCs and increased CD8⁺ T cells in tumors, consistent with

alleviation of MDSC suppression of T cells (Fig. S2h, k). We concluded that *Arid1a* loss shapes an immune-suppressive microenvironment in PCa via infiltrating PMN-MDSCs.

S2c, Gene expression were quantified by real-time PCR in Ly6G⁺ cells from 3-month-old *Pten*^{PC-/-}; *Arid1a*^{PC-/-} mice prostate tumors and WT mice peripheral blood (n = 5). **S2d**, Summary of T cell proliferation assessed by CFSE flow cytometry after 4 days. High and low proliferation were defined as T cell division ≥ 2 and ≤ 1 , respectively (n = 5). **S2e**, IFN- γ secretion by CD8⁺ T cells in the assay in **S2d**, measured by ELISA (n = 5). **S2h**, Quantification of each tumor-infiltrating immune cell type in WT and *Arid1a* KO Myc-CaP subcutaneous tumors with or without Ly6G antibody treatment (n = 6). **S2k**, Flow cytometry analysis of PMN-MDSCs, CD8⁺ T cells and IFN γ ⁺ CD8⁺ T cells (n = 6). **S2c**, **S2e**, **S2h**, **S2k**, Data represent the mean \pm SEM. Statistical significance was determined by two-tailed unpaired *t* test (**S2c**, **S2e**, **S2k**), Fisher's exact test (**S2d**) and two-way ANOVA followed by multiple comparison (**S2h**). *** *p* < 0.01, ns, no significance.

3) To identify potential drivers of ARID1A-mediated immunosuppressive activity (Figure 4f) the authors overlaps the open chromatin regions with reduced BRG1/SMARCA4 peaks and down-regulated genes. Given the proposed mechanism via enhancer reprogramming, greater explanation of how the specific enhancer-gene promoters were identified is needed.

Response: Studies have clarified that chromatin modifications can be employed for a more accurate discrimination between promoters and enhancers^{8,9}. Enhancers and promoters can be distinguished by the methylation status at H3K4. Histone H3K4me1 and H3K27ac are enhancer-specific modifications and are required for enhancers to activate transcription of target genes^{9,10}, whereas high levels of trimethylation (H3K4me3) in combination with H3K27ac predominantly mark active or poised promoters^{8,9,11}. Here, we integrated H3K4me1, H3K4me3 and H3K27ac ChIP-seq results to distinguish active enhancer and promoters. In this revision, we have clarified how the enhancers and promoters were categorized with the references

cited. For *A20* gene, a closer examination revealed only one BRG1 binding site, which occupancy was decreased upon *Arid1a* KO. This BRG1 binding site was simultaneously marked by H3K27ac and H3K4me1, but not H3K4me3 (**Fig. 4g**); therefore, it was termed as enhancer. *Arid1a* KO led this site exhibiting the reduced H3K27ac, a histone marker to demark active transcription (**Fig. 4g**). In addition, we showed enhancer RNA (eRNA) production was decreased upon *Arid1a* deletion (**Fig. S4f**). Thus, these results indicated that *Arid1a* loss silences the enhancer of *A20* gene.

4) Figure 4i should be removed as it largely duplicates Figure 4g and visualization of eRNA in the track is difficult.

Response Fig. 4i were removed as suggested. Due to new data added in this revision, Fig. 4g in original submission was moved to Fig. S4f.

5) Pg 18, line 2-4: It is stated that ARID1A-OE tumors exhibit increased sensitivity to anti-PD1 therapy. However, the overexpressing tumors also have reduced growth. More rigorous statistical testing is needed to confirm the proposed synergy.

Response : We thank the reviewer for this important question. To clarify this issue, we performed a similar assay via increase of animal number (n = 10). *ARID1A* overexpression slightly but significantly reduced xenograft growth. WT tumors were refractory to anti-PD1 antibody treatment, which did not show significant effect on tumor growth (**Fig. 7e**). In contrast, *ARID1A* overexpression rendered the tumors to be sensitive to anti-PD1 antibody treatment, as reflected by the further reduction in tumor volumes upon anti-PD1 antibody treatment as compared to isotype treatment (**Fig. 7e**). Comparison in each group was calculated by two-way ANOVA followed by multiple comparisons. We have incorporated this new data in the revised manuscript.

7e

7e, Volume of tumors derived from WT and *ARID1A*-overexpressing cells injected subcutaneously into FVB mice and treated with isotype and anti-PD1 as indicated (n = 10).

Data represent the mean \pm SEM. Statistical significance was determined by two-way ANOVA followed by multiple comparisons.

6) Pg 18, line 11-12: The authors claim that "cotargeted treatment (of NF- κ B inhibitor and/or anti-PD1/CTLA4) provoked the regression of *Pten*^{PC-/-}; *Arid1a*^{PC-/-} tumors.". While an intriguing concept the data is somewhat misleading as the experiment (fig 7f) cannot separate the impact of NF- κ B inhibition alone. This is particularly important as the authors demonstrated that inhibiting this pathway has a profound impact on tumour growth (Fig 3i). It is therefore unclear if the reduced adenoma observed is due to a synergism with anti-PD1/CTLA4 or simply

Response: We thank the reviewer for the insightful suggestion to improve our studies. Thus, we included *Pten*^{PC-/-}; *Arid1a*^{PC-/-} tumors with NF- κ B inhibition alone to compare with co-targeted treatment (the combinational regimen of NF- κ B inhibitor and anti-PD1/CTLA4 antibody treatment). Judged by histology quantitation, IHC analysis and immunoprofiling data, NF- κ B inhibition compromised *Arid1a*-deleted tumor progression along with the reduced PMN-MDSC infiltrations and expansion of CD8⁺ T cells (Fig. 7f, g and Fig. S7e, f). NF- κ B inhibition attenuated the elevation of CXCL2 and CXCL3 in *Arid1a*-deleted tumors (Fig. S7g). These results support that *Arid1a* loss expedites PCa progression via NF- κ B-induced PMN-MDSC chemotaxis.

Importantly, when we combined the NF- κ B inhibition with anti-PD1/CTLA4 treatment, we observed further inhibition of prostate tumorigenesis and enhanced anti-tumor CD8 T cell response compared with NF- κ B inhibition alone (Fig. 7f, g and Fig. S7e, f). With this new data

incorporated, we concluded that NF- κ B inhibition and immune checkpoint blockade (ICB) therapies acted in synergistical manner to impede *Pten*^{PC-/-}; *Arid1a*^{PC-/-} tumor progression.

7f, Prostate tumor histology of *Pten*^{PC-/-}; *Arid1a*^{PC-/-} mice with or without NF- κ B inhibition (JSH-23) in combination with anti-PD1/CTLA-4 treatment (n = 10). **7g**, IHC staining for Ki67, CD8 and Ly6G and β -Gal in sections by the indicated treatments. Scale bars, 50 μ m. **S7e**, Prostate tumor histology of *Pten*^{PC-/-}; *Arid1a*^{PC-/-} mice with or without NF- κ B inhibition (JSH-23) in combination with anti-PD1/CTLA-4 treatment (n = 10). **S7f**, Quantification of the indicated tumor-infiltrating immune cell population of *Pten*^{PC-/-}; *Arid1a*^{PC-/-} mice prostates with or without anti-NF- κ B treatment in combination with anti-PD1/CTLA-4 therapy (n = 5). **S7g**, ELISA of CXCL2 and CXCL3 in prostate tumors of 4-month-old *Pten*^{PC-/-}; *Arid1a*^{PC-/-} mice treated as indicated (n = 5).

S7e-g, Data represent the mean \pm SEM. Statistical significance was determined by χ^2 test (**S7e**) and two-way ANOVA followed by multiple comparisons (**S7f**, **S7g**). Representative data of triplicate experiments are shown. ns, no significance.

Minor comments:

- Figure 2A - Needs an x-axis label

- Figure 3 I - Needs an x-axis label

- Figure 4D - Needs samples labels (red/black)

Response : We apologize for the mistakes during preparation of the manuscript. The inaccuracies were revised accordingly.

Page 12, Line 8 /throughout - Please change the Open Chromatin Region (OCR) to the more

common convention cis-regulatory element (CRE)

Response: Open chromatin regions (OCRs) are nucleosome-free regions that can be accessed by protein factors. It is often used to assess chromatin accessibility. CREs are collections of transcription factor binding sites and other non-coding DNA that are sufficient to activate transcription in a defined spatial and/or temporal expression domain, composed of DNA (typically, non-coding DNA) containing binding sites for TFs and/or other regulatory molecules. After careful consideration of your insightful suggestion, we changed “OCR region” to “region exhibiting the reduced chromatin accessibility” throughout the manuscript.

Pg 13, line 18 - The extended Data Fig. 4e does not show a reduction in H3K27Ac.

Response: We apologize for not making it clear in the initial submission (Fig. S4e in the original submission). Now we showed that *Arid1a* loss led to *A20* enhancer (BRG1 binding site) exhibiting the reduced H3K27ac modifications by browser tracker (**Fig. 4g**) and ChIP-qPCR analysis (the first two columns in **Fig. S4i**), which was reverted by dCas9-GCN4-mediated CRISPR activation (CRISPRa) (**Fig. S4i**).

Reviewer #2 (Remarks to the Author):

In the present manuscript, Yan et al. investigate the loss of ARID1A, a subunit of the SWI/SNF chromatin remodeling complex, as a new mechanism by which the prostate tumour microenvironment recruits immunosuppressive MDSCs. The authors generated and employed a new transgenic mouse model (PtenPC^{-/-}; Arid1aPC^{-/-}) to prove their hypothesis. They showed that loss of ARID1A in a Pten loss background promotes metastasis by enhancing the recruitment of MDSCs via NFκB in the tumour microenvironment. In line with their hypothesis, the authors demonstrate by using available datasets (TCGA) that ARID1A signature opposite correlates with MDSC and NFκB signature and has a worst prognosis in prostate cancer patients. This is a timely study: immunotherapy has not reached a satisfactory degree of clinical efficacy in prostate cancer despite many years of research and clinical development. One of the reasons is that the tumour microenvironment of prostate cancers is highly immunosuppressive due to the infiltration of MDSCs. any groups worldwide are now exploring new therapies aimed to block the tumour recruitment of these cells. Linking tumour epigenetic changes with a higher immune suppressive microenvironment that drive metastasis is interesting, even if not fully novel. The effects of ARID1A loss in SWI/SNF binding and on the epigenetic machinery (H3K4me3/H3K27ac) (10.1038/ng.3744) and the effect of NFκB activation and MDSC recruitment have already been characterized. Additionally, the frequency of ARID1A mutation in prostate cancer patients is very low, thus the overall clinical relevance remains limited.

Response: We appreciate the reviewer's interest and positive comments on our study. Point-to-point responses are addressed below.

Major concerns:

1) The manuscript seeks accurate histopathological analysis and tumour measure of the immunohistochemistry images with higher resolution and magnification are required. Evidence of metastasis needs to be better demonstrated. Are the mice also collecting metastasis at earlier times, such as 12 weeks old? Different time-points and markers (e.g., PanCK) need to be included.

Response: Thanks for your constructive comment to strengthen our studies. As suggested, we incorporated the quantitation of prostatic volumes and replaced higher magnification of H&E

and IHC images to substantiate our conclusion (**Fig. 1d, f, g, h and j**). Quantitation of 3- or 4-month-old prostatic volumes suggested that *Arid1a* deletion accelerated *Pten* loss-induced PCa progression (**Fig. 1d**). In addition, we showed that 12-week-old *Arid1a* deleted mice already developed locally invasive cancer (**Fig. 1e** and **Fig. S1g**). The tumor malignance progressed over time, showing that the percent of adenocarcinoma further increased in 4-month-old *Arid1a*-deleted prostates relative to that in 3-month-old mice (**Fig. 1e, f** and **Fig. S1g**). Metastatic incidences in 12-week-old *Arid1a* KO mice are less obvious, in which less lymph node and minimal lung metastasis were developed (**Fig. 1i, j** and **Fig. S1h**). We also performed CK8 staining to show that luminal origin tumor cells indeed invaded to stromal compartment (**Fig. 1h**). With the characterizations of the additional time point in mice, our results clearly indicated that *Arid1a* deletion cooperates with *Pten* loss to accelerate PCa progression.

1d, Quantitation of 3- or 4-month-old prostatic volumes of indicated mice (n = 10). **1e**, Quantification of histological grade at the indicated time points (n = 15). **1f**, H&E-stained sections in the anterior (AP), dorsolateral (DLP) and ventral (VP) prostates of 16-week-old *Pten*^{PC-/-} and *Pten*^{PC-/-}; *Arid1a*^{PC-/-} prostates (n = 15, representative data are shown). Scale bars, 50 μ m. **1g**, Immunohistochemical analysis of ARID1A and Ki67 expression and β -Gal staining in prostate sections. Scale bar, 50 μ m. **1h**, IHC for SMA α , AR and CK8 in prostate sections. Scale bars, 50 μ m. **i**, Quantification of the metastatic incidence as indicated (n = 15). **1j**, AR and CK8 staining of lymph nodes and lungs from 16-week-old mice. Scale bar, 50 μ m. **S1g**, H&E-stained sections in the AP, DLP and VP sections of 12-week-old *Pten*^{PC-/-} and *Pten*^{PC-/-}; *Arid1a*^{PC-/-} mice (n = 15, representative data are shown). Scale bars, 50 μ m. **S1h**, AR and CK8 staining of lymph nodes from 12-week-old mice. Scale bar, 50 μ m.

1d, **1e**, Data represent the mean \pm SEM. Statistical significance was determined by two-tailed unpaired *t* test (**1d**), χ^2 test (**1e**) and Fisher's exact test (**1i**). * *p* < 0.05. ** *p* < 0.01.

2) *The characterization of the tumour development in PtenPC-/-; Arid1aPC-/- mice has been performed at 16 weeks old (Figure 1). However, the characterization of the tumour microenvironment (Figure 2) reports data collected from mice 12 weeks old. What is the disease stage at that time point? The analyses need to be coherent.*

Response: Following your last comment, now we showed that 12-week-old *Arid1a* deleted mice already developed adenocarcinoma with less metastatic incidences (**Fig. 1e, i and Fig. S1g, h**). We chose the earlier time point to characterize expression changes; therefore, the genes altered might be more directly related to earlier events influenced by *Arid1a* loss rather than secondary to malignance differences. We apologize for not making it clear in the initial submission, and the related issue was clarified in the revised manuscript.

3) *The manuscript does not clarify which factors are differentially expressed by the prostate epithelium of Pten; Arid1a-ko vs Pten-ko tumours. This is a significant limitation of the study. The authors should perform a gene expression profiling of cytokines and cytokine-related factors in Pten; Arid1a-ko vs Pten-ko tumours to demonstrate changes in the recruiters of MDSCs. Reporting results of a few cytokines by qPCR (Figure 3a) is limited to having unbiased transcriptomic profiling of the mouse models (Figure 2a). Additionally, evidence regarding the soluble factors produced by the ARID1A-deficient epithelial tissues is required. Are they accumulating also in the serum? What about the cytokine profile of tumour cells upon NFkB inhibition? Is it equal to Arid1a genetic inhibition?*

Response: We thank the reviewer for pointing out this important issue. As suggested, we presented expression profiling-based volcano plot to indicate the changes of cytokine and

chemokine related factors upon *Arid1a* loss (**Fig. 3a**), whereas qPCR validations were moved to supplemental figure (**Fig. S3a**). Per your comments, we demonstrated that CXCL2 and CXCL3 level were indeed elevated in the serum and prostate tissues of *Arid1a* deleted mice (**Fig. 3f**). We further proved that treatment with JSH-23 to diminish NF- κ B signaling compromised *Arid1a*-deleted tumor growth to an extent similar as WT xenografts (**Fig. 3g**). RNA-seq analysis to compare the expression profile of WT and *Arid1a* deleted tumor with or without NF- κ B inhibition indicated that JSH-23 treatment led to the downregulated expression of more than half of upregulated genes elicited by *Arid1a* loss (504/784; **Fig. 3h**). These genes included *Cxcl2*, *Cxcl3*, *Tnf* and among others. In *Pten^{PC-/-}; Arid1a^{PC-/-}* mouse models, we proved that NF- κ B inhibition attenuated the inductions of CXCL2 and CXCL3 caused by *Arid1a* loss, along with the reduced recruitment of PMN-MDSCs and expansion of CD8⁺ T cells (**Fig. S7g**). Together, these results supported that ARID1A loss shapes the immunosuppressive TME via NF- κ B-induced MDSC chemotaxis.

3a, Volcano plots showing the differentially expressed genes (DEGs) between epithelial cells of 3-month-old *Pten^{PC-/-}; Arid1a^{PC-/-}* versus *Pten^{PC-/-}* mice prostates (n = 3). The upregulated and downregulated cytokines and chemokines are indicated. **3f**, ELISA of CXCL2 and CXCL3 in serum and prostate tumors of 3-month-old mice (n = 5). **3g**, Tumor volume of Myc-CaP expressing sgARID1A and control vector subcutaneously inoculated into FVB mice with or without JSH-23 treatment. **3h**, Epithelial cells in xenografts (g) were sorted for RNA-Seq and DEGs between WT and *Arid1a* KO epithelium were shown in heatmap (n = 2). **S7g**, ELISA of CXCL2 and CXCL3 in prostate tumors of 4-month-old *Pten^{PC-/-}; Arid1a^{PC-/-}* mice treated as indicated (n = 5). **3f**, **3g**, **S7g**, Data represent the mean \pm SEM. Statistical significance was determined by two-tailed unpaired *t* test (**3f**) and two-way ANOVA followed by multiple comparisons (**3g**, **S7g**). **p* < 0.05. ***p* < 0.01, ns, no significance.

4) A big limitation of the study is the absence of a clear demonstration of the endocrine status of the *Pten*; *Arid1a*-ko tumours. Are they hormone-dependent or independent, meaning do they respond or not to castration?

Response: We agree with reviewer that we should assess whether *Arid1a* is implicated in castration resistant. Thus, 10-week-old *Pten*^{PC-/-}; *Arid1a*^{PC-/-} mice received castration followed by additional 2 months of enzalutamide (ENZ) treatment to mimic androgen deprivation therapy (ADT, **Fig. S2i**). Reflected by changes in tumor volume and histology quantitation (**Fig. S2m-o**), we showed that *Arid1a* deleted tumors were refractory to ADT as compared to *Pten*-deleted lesions. Similar as sham operated mice, ADT-treated *Arid1a* KO tumors also enriched in PMN-MDSCs and paralleled a significant reduction of CD8⁺ T cells, whereas AR remained comparable between *Arid1a* WT and KO mice (**Fig. S2p**). Whether other mechanism implicated in resistance is an important question, but we feel it falls beyond the scope of the present studies. We have incorporated castration resistance data in the revised manuscript, and discussed that the mechanisms await further investigations.

S2i, Graphical scheme describing the ADT strategy. **S2m**, Gross photographs and volume of prostate gland from indicated mice with ADT (n = 10). Scale bar, 1 cm. **S2n**, H&E-stained sections in 5-month-old *Pten*^{PC-/-} and *Pten*^{PC-/-}; *Arid1a*^{PC-/-} prostates with ADT (n = 10, representative data are shown). Scale bars, 50 μ m. **S2o**, Quantification of histological grade from indicated mice with ADT (n = 10). **S2p**, Immunohistochemical analysis in castrated prostate sections. Scale bar, 50 μ m.

2m, Data represent the mean \pm SEM. Statistical significance was determined by two-tailed unpaired *t* test (**S2m**) and χ^2 test (**S2o**). Representative data of triplicate experiments are shown. * *p* < 0.05. ** *p* < 0.01.

Minor Points:

- The manuscript needs to be carefully checked for typos, and not all the panels are cited in the text.

Response: We apologize for the mistakes during preparation of the manuscript. The inaccurate presentation and descriptions are revised accordingly.

In figure 2b, the reported graph is not clear. The author should display the quantification of the immune population identified by the tSNE analysis more comprehensibly.

Response: We added the missing genotype and quantitation in Fig. 2b.

Reviewer #3 (Remarks to the Author):

Li et al. propose an intriguing hypothesis that ARID1A loss in conjunction with PTEN loss plays a role in prostate cancer immunosuppression via polymorphonuclear myeloid-derived suppressor cell (MDSC). They suggest this is mediated through CXCR2 ligand-mediated MDSC chemotaxis. At the outset, the authors propose a major role for inflammation and TME that promote prostate cancer (PCa) metastasis and diminished response to checkpoint therapy. There have been several high-profile papers implicating the role of MDSC in advanced PCa.

Response: We thank the reviewer for the insightful suggestions to improve our studies. Point-to-point responses are addressed below.

Comments

1) ARID1A mutations are suggested to act as tumor suppressors given that the majority are missense or frameshift mutations. The low percentage of mutations in localized PCa (TCGA) and advanced PCA (Cytra et al., Nature Communications), suggest that there would need to be other mechanisms for regulation of ARID1A if this plays a significant role in human PCa. What evidence do the authors have for dysregulation in human prostate cancer?

Response: We fully agree with reviewer that *ARID1A* is less frequently mutated in PCa (approximately 2% in The Cancer Genome Atlas [TCGA] dataset). However, when examination of its protein level, we detected lower *ARID1A* expression in tumors than that in adjacent normal tissues (**Fig. S1a**). *ARID1A* immunostaining intensity was negatively associated with the Gleason score (GS) and PSA levels in prostate tumors (**Fig. S1b, c**). More importantly, patients with low *ARID1A* expression exhibited a higher risk of biochemical recurrence (BCR) than those with high *ARID1A* levels (**Fig. 1b**). These results prompted us to investigate its possible function in prostate tumorigenesis. Further mechanistical studies identified that inflammation induced *IKK β* promoted *ARID1A* downregulation in PCa cells. Per your comments, we have revised the manuscript to clarify the rational for present study.

2) The authors demonstrate ARID1a expression as decreasing in higher Gleason grade localized PCa. Is this consistent with a model that leads to increased MDSC? MDSC are not

seen at high levels in localized PCa?

Response: Per your comments, we investigated the association between the abundance of MDSCs and Gleason score (GS) in PCa specimens. A positive association between MDSC abundance and GS was observed (**Fig. S7a**), highlighting the important role of MDSC in prostate progression.

S7a

S7a, CD15 expression levels stratified by GS in TMA (GS < 7, n = 14, GS = 7, n = 53, GS > 7, n = 33). Statistical significance was determined by one-way ANOVA followed by multiple comparisons.

3) The authors show that depletion of *PTEN* alone and in concert with depletion of *TP53* correlates with downregulation with *ARID1A* in vivo. From this data (Figure 1a and the following) it remains unclear if the depletion of these genes is inducing *ARID1A* downregulation or if these tumors are generally more inflamed? Looking at total *IKKb*, it seems that this increases in the same manner as the *p-IKKb* (Fig 1A), therefore quantifying these bands might help to understand if phosphorylation is really increased and involved in the downregulation of *ARID1A*. It would be helpful to understand this phenomenon if this could be elaborated a bit more, is there a higher burden of cytokines and/or MDSCs in the *PTEN* depleted and, especially *PTEN/TP53* depleted setting vs *WT*? Why is the depletion of *ARID1A* necessary to show results in Fig.2? Fig1. Showed complete downregulation of *ARID1A* in the *PTEN/p53* depleted setting.

Response: We thank the reviewer for pointing out this important issue. We also noticed this phenomenon and investigated this possibility. However, siRNA mediated deletion of *TP53* in C4-2 and 22RV-1 cells did not alter *ARID1A* expression (**Fig. S5a, b**). Similarly, depletion of *PTEN* or concurrent loss of *PTEN* and *TP53* in DU145 and 22RV-1 cells did not affect *ARID1A* level as well (**Fig. S5a, b**). Thus, we concluded that *PTEN* and *P53* are unlikely to directly control *ARID1A* level in PCa cells. As suggested, we quantified *ARID1A* level and the ratio of *p-IKKβ* versus total *IKKβ* and found that they were inversely correlated in prostate genetically engineered mouse models (GEMMs), in which a lowest *ARID1A* expression and higher

proportion of p-IKK β /IKK β was detected in *Pten/Trp53* DKO tumors (**Fig. 1a**). Thus, it is reasonable to speculate that prominently reduced ARID1A in *Pten/Trp53* DKO tumors was due to inflammatory signal induced IKK β activation. To clarify this issue, we have incorporated these results in the revised manuscript.

S5a, IB analysis of C4-2 and DU145 cells with *TP53* or *Pten* KD respectively. **S5b**, IB analysis of 22RV-1 cells with *TP53* an/or *Pten* KD. **1a**, IB analysis of the indicated proteins in mice prostate tissues with the indicated genotypes (n = 3).

Experiments were repeated three times independently with similar results; data from one representative experiment are shown.

Following your comment, we showed that *PTEN/Trp53* DKO tumors produced significantly higher cytokine and their related factors (**Please see figures attached below for your reference, Fig. a**), reminiscent of *Pten/Arid1a* deleting setting. Similar as the previous studies¹², we detected the enrichment of PMN-MDSCs in *Pten/Trp53* DKO tumors related to WT mice (**Fig. b**). Therefore, it is reasonable to speculate that *Arid1a* downregulation might contribute to form an immunosuppressive TME in *Pten/Trp53* KO tumors. Notably, our results are carried out in *Pten* null context rather than *Pten/Trp53* DKO setting, in which ARID1A expression is still present. To improve the narration and avoid the confusion, the related issues have been clarified with the additional data incorporated.

a, Heatmap summarizing the qRT-PCR results in epithelial cells of 3-month-old WT, *Pten^{PC-/-}*, *Arid1a^{PC-/-}* and *Pten^{PC-/-}; Trp53^{PC-/-}* mice prostates (n = 3). **b**, quantification of PMN-MDSC in 3-month-old WT, *Pten^{PC-/-}*, *Arid1a^{PC-/-}* and *Pten^{PC-/-}; Trp53^{PC-/-}* mice prostates by FACS (n = 6). Statistical significance was determined by one-way ANOVA followed by multiple comparisons.

4) Especially, the effect with TP53 depletion seems to strongly affect ARID1A expression, however, from there onwards the authors seem to ignore their double knockout model. For example, experiments performed in Fig 3. C would be interesting in the context of p53 deletion or using cell lines that harbor p53 mutations such as DU145.

Response: As suggested, we proved that P53 deletion (22RV-1 cells; **Fig. a**) or mutation (DU145 cells; **Fig. b**) did not alter ARID1A loss effects to stimulate NF-κB signaling (**Please see figures attached below for your reference**).

a, IB analysis of WT and ARID1A-depleted 22RV-1 cells with or without TP53 KD stimulated with TNF α treatment at the indicated time points. **b**, IB analysis of WT and ARID1A-depleted DU145 cells stimulated with TNF α at the indicated time point.

Experiments were repeated three times independently with similar results; data from one representative experiment are shown.

5) This study focuses highly on the microenvironmental aspects of PCa. How does the loss of ARID1A affect SWI/SNF composition in the functional mouse model PCa components? Ding et al. (PMID: 3049614) have shown that PTEN stabilizes BRG1 expression via the AKT/GSK3b/FBXW7 axis, could this be true for ARID1A also?

Response: Thanks for your constructive comment. As mentioned above, we excluded the possibility that PTEN directly regulate ARID1A expression (**Fig. S5a, b**). As suggested, we assessed whether Arid1a loss affects the remaining SWI/SNF complex in mouse prostate tumors. However, qRT-PCR and immunoblotting analysis collectively pointed out that Arid1a loss did not significantly alter the key components of SWI/SNF complex, including ARID1B, BRG1, BAF155, BAF53B and among others (**Fig. 4a** and **Fig. S4a**). Moreover, we also performed Co-IP analysis by using anti-BRG1 antibody to compare SWI/SNF composition in *Pten*^{PC-/-} and *Pten*^{PC-/-}; *Trp53*^{PC-/-} tumors. Regardless of *Arid1a* loss, BRG1 co-

immunoprecipitated the comparable amount of SWI/SNF components, including ARID1B, BAF155, BAF53B and BAF45B (**Fig. S4b**), indicating that *Arid1a* loss did not alter the SWI/SNF complexes that are remaining.

4a, Relative expression of SWI/SNF components in epithelium from 12-week-old *Pten^{PC-/-}* and *Pten^{PC-/-}; Arid1a^{PC-/-}* prostates (n = 3). **S4a**, IB analysis of the indicated protein in 3-month-old *Pten^{PC +/-}* and *Pten^{PC +/-}; Arid1a^{PC-/-}* mouse prostates. Data represent the mean \pm SEM. Statistical significance was determined by two-tailed unpaired *t* test, NS, no significance. **S4b**, IB analysis of the WCL from 12-week-old *Pten^{PC +/-}* and *Pten^{PC +/-}; Arid1a^{PC-/-}* prostates and anti-BRG1 immunoprecipitates as indicated.

Experiments were repeated three times independently with similar results; data from one representative experiment are shown. ns, no significance.

6) One intriguing question that comes up when knocking out ARID1A is what happens to the SWI/SNF complexes that are remaining? Is 1B elevated? What happens to the other complex members. BRG1 has been shown to increase in advanced PCa. As the authors perform BRG1 ChIP, it would be important to know if BRG1 and perhaps other markers of lineage plasticity are altered with ARID1A KO including BAF53b.

Response: As mentioned above, *Arid1a* loss did not markedly alter the expression of SWI/SNF components, including BRG1, ARID1B, BAF53B, BAF155 and etc. (**Fig. 4a**). Reviewer raised an excellent question whether *Arid1a* loss alters lineage plasticity, as previous studies indicate that specialized SWI/SNF complexes are associated with neuroendocrine or small cell prostate cancer and may play a role in therapy resistance^{13, 14}. Nevertheless, we found that *Arid1a* loss did not alter the markers of neuroendocrine lineage (*Chag*, *Eno2* and *Syp*), basal (*Trp63*, *Ck5*) and luminal cells (*Ck8* and *Ck18*) (**Fig. S1i**). In addition, neuroendocrine lineage regulators *Ezh2*, *Sox2*, *Ascl1*, and androgen signal (*Ar* as well as its target, *Tmprss2*) remained unaffected upon *Arid1a* loss (**Fig. S1i**). These results suggest that *Arid1a* loss is not implicated in lineage

plasticity in our experimental model, highlighting specialized assemblies of the SWI/SNF complex with distinct functions in PCa.

S1i

S1i, Relative expression of lineage markers in epithelium from 12-week-old $Pten^{PC-/-}$ and $Pten^{PC-/-}; Arid1a^{PC-/-}$ prostates (n = 3). Neuroendocrine lineage (*Chga*, *Eno2* and *Syp*), basal (*Trp63*, *Ck5*), luminal (*Ck8*, *Ck18*), neuroendocrine lineage regulators (*Ezh2*, *Sox2* and *Ascl1*), and *Ar* target *Tmprs2*. Data represent as the mean \pm SEM. Statistical significance was determined by two-tailed unpaired *t* test, ns, no significance.

7) A recent study by Wellinger et al. (PMID: 34782346) has shown that TNF-mediated apoptosis can be potentiated by BET inhibition in cancer cells. Can this be exploited in the context of PCa as well? The effect on cancer cells and immune cells would be very interesting, especially in the ARID1a depleted setting.

Response: We thank the reviewer for pointing out this interesting issue. Studies indicate that BETi prevented the recruitment of BRD4 to p65-bound cis-regulatory elements to disrupt pro-survival NF- κ B signaling, and thereby inducing unrestrained TNF-mediated activation of the extrinsic apoptotic cascade and tumor cell death¹⁵. As a consequence, BETi sensitizes tumors to ICB therapy such as PD-1 treatment. Based on our results, *Arid1a* deletion resulted in the cells harboring higher NF- κ B activity; thus, the tumors might be also sensitive to BETi. Hence, we treated *Arid1a* WT or KO Myc-CaP cell with BRD4 inhibitor, JQ-1 to examine its response to TNF α -induced cell death (**Please see figures attached below for your reference**). Interestingly, we detected that treatment with JQ-1 sensitized *Arid1a*-deleted cells to TNF α induced death as compared to *Arid1a* intact cells (**Fig. a**). Likewise, we inoculated *Arid1a* depleted cells in a syngeneic model, and treated the tumors with anti-PD1 antibody with or without JQ-1 co-treatment. We noticed that JQ-1 and PD1 treatment acted in a synergistical manner to inhibit tumor growth (**Fig. b**), along with the increase of total CD8⁺ T cells (**Fig. c**). These results suggested that BETi might render *Arid1a* low or deleted tumors more sensitive to

ICB. We have discussed this possibility in the discussion section of revised manuscript.

a, WT and *ARID1A*-depleted Myc-CaP cells were co-cultured in 1 μ M BET inhibitor JQ-1 or DMSO in the presence or absence of recombinant murine TNF α (10 ng/mL) for 18 hours prior to assessment of tumor cell death by flow cytometry. **b**, Growth curves showing average *ARID1A* KO Myc-CaP subcutaneous tumor growth in each treatment condition (n = 6). 50 mg/kg JQ-1 (HY-13030, MedChemExpress) was given intraperitoneally once daily (14 days). Anti-mouse PD-1 (BioXCell, Clone RMP1-14) treatment was administered twice a week through intraperitoneal injections at a dosage of 200 μ g/injection. **c**, Flow cytometry analysis of tumor infiltrating CD8⁺ cells from Myc-CaP tumor-bearing mice treated with JQ1 alone or in combination with anti-PD1 on day 30 (n = 6). Data represent as mean \pm SEM. Statistical significance was determined by two-way ANOVA followed by multiple comparisons.

8) Some of the *ARID1A* low tumor specimens do not show the postulated trend of increased *IKKb* and *p65* expression (Fig 7. A). What could be the possible explanation that *ARID1A* is downregulated here?

Response: We apologize for not making it clear in the initial submission. As a key subunit of SWI/SNF complex, *ARID1A* is showed to be regulated at transcriptional and post-transcriptional level by various mechanisms. For instance, *ARID1A* is transcriptionally downregulated by their promoter methylation and repressive histone modifications¹⁶⁻¹⁸. In addition, the ubiquitin-proteasome system controls *ARID1A* stability¹⁹⁻²³. It was reported that phosphorylation of *ARID1A* is catalyzed by nuclear kinase ATM, and β -TRCP recognizes the phosphorylated *ARID1A* in response to DNA damage treatment^{21, 22}. In squamous cell carcinoma, TRIM32 directly ubiquitinates *ARID1A* to promote its degradation, while USP11 deubiquitinates *ARID1A* to stabilize *ARID1A*²³. Thus, *ARID1A* might serve as a hub to sense various stimuli to shape the TME and impact tumorigenesis. We detected an inverse correlation between *ARID1A* and p-IKK β or nuclear p65 expression, indicating that IKK β activation mediated *ARID1A* downregulation is important, but not necessary predominates in all prostate tumors. Other mechanism discussed above might contribute to its alternation dependent on different genetic milieu or context. We have clarified this issue in the revised manuscript.

9) Does the inflammatory environment also influence ARID1A expression of the immune compartment/surrounding tissue or is ARID1A safeguarded by intact PTEN/TP53 (coming back to the question: is the loss of tumor suppressors directly promoting this phenomenon or is it promoting the inflammation? Is this a PCa-specific phenomenon? If yes, why do we not see downregulation or mutations of ARID1A more often in PCa, since PTEN is among the most frequently mutated genes in PCa?

Response: Following your last comment, we have demonstrated that PTEN and P53 did not directly regulate ARID1A expression. The prominent ARID1A reduction in *Pten/Trp53* DKO tumours is likely due to hyperactivation of IKK β , and thereby promoting ARID1A destruction. As suggested, we further examined whether this mechanism is cell type specific. Firstly, we observed that TNF α treatment stimulated ARID1A turnover in lung and colorectal cancer cells, A549 and HCT116 cells (**Fig. S5k**), suggesting that TNF α -IKK β axis promotes ARID1A turnover in a cancer cell type independent manner. In addition, similar results were obtained in Jurkat T cells, RAW264.7 macrophage cells and OCI-Ly10 B cells (**Fig. S5l**), suggesting that it might also control ARID1A level in immune cells. Together, we proposed that IKK β acts as the convergence point for inflammatory signals to promote ARID1A downregulation. This regulatory mechanism might represent a general mechanism. Based on these results, we have noted that this issue awaits further exploration.

S5k, IB analysis of A549, and HCT116 cells treated with TNF α for the indicated duration of time. **S5l**, IB analysis of Jurkat, RAW264.7 and OCI-Ly10 cells treated with TNF α for the indicated duration of time.

Experiments were repeated three times independently with similar results; data from one representative experiment are shown.

Minor Comments

1) *The authors use the term “adenoma” throughout the manuscript. What is meant by this? This term is not usually used for PCa.*

Response: We apologize for the inaccuracy, which has been changed to “adenocarcinoma”. In the method section, we have described how we define mouse adenocarcinoma similar as the previous reports^{24, 25}.

Reviewer #4 (Remarks to the Author):

In this manuscript the authors identify a new molecular network that accounts for increased recruitment of MDSC and consequent immune suppression in prostate cancer, with implications for tumor progression and resistance to immune checkpoint therapies. The authors adopt a series of in vitro and in vivo experiments, in mouse models, and analyses on patient-derived tumor specimens and data sets. They elegantly demonstrate that ARID1A, a member of the SWI/SNF1 complex, silences the expression of chemokines, including CXCL2 that promote the recruitment of MDSC in the tumor. Indeed, downregulation of ARID1A results in increased CXCL2 production via NF- κ B mediated signaling. Investigating deeply on the molecular mechanisms of this process they found that inflammatory cytokines, mainly TNF α , are responsible for ARID1A downregulation via IKK β activity. Clinical relevance is given by the observation that ARID1A expression is low in human prostate cancer and correlates with prognosis.

However, the manuscript has several inconsistencies that need to be fixed:

Response: We thank the reviewer for the positive comments, and the insightful suggestions to strengthen our studies. As such, we performed additional experiments and revised the manuscript.

1) Page 5 line 20: *“The results showing that patients bearing low ARID1A expression had adverse disease outcomes”*. Please specify that this conclusion arises from results show in the manuscript.

Response: We have indicated it is related to figure 1b.

2) *Figure 1b shows correlation of ARID1A expression with clinical outcome in the cohort of patients of the tissue microarray used in the work. Translational relevance of results would benefit if this kind of analyses would be extended to deposited patient-derived data sets, as those used in figure 7 for correlation of ARID1A expression with NF- κ B signaling and MDSC infiltration.*

Response: Thanks for the reviewer's suggestion. Given ARID1A expression in PCa is largely

regulated by protein stability rather than mRNA level, we cannot use public dataset to analyze its mRNA to correlate with disease outcome. Per reviewer comment, we used ARID1A signature as defined by differentially expressed genes between epithelial cells of 3-month-old *Pten*^{PC-/-}; *Arid1a*^{PC-/-} versus *Pten*^{PC-/-} mice prostates ($p < 0.05$, foldchange ≥ 1.5), and observed a trend correlating lower ARID1A signature and the probability of biochemical recurrence in public dataset (GSE21034), albeit it did not reach statistical significance (**Please see figures attached below for your reference**). Signature-based analysis is likely to be confounded by different genetic milieu or context; however, it at least suggested a possible clinical implication. In any event, our TMA results support its clinical relevance.

Kaplan–Meier plot of recurrence after radical prostatectomy based on the t-score of ARID1A signature (Low, n = 35, Middle, n = 70, High, n = 35). ARID1A signature was defined by the differentially expressed genes between epithelial cells of 3-month-old *Pten*^{PC-/-}; *Arid1a*^{PC-/-} versus *Pten*^{PC-/-} mice prostates ($p < 0.05$, foldchange ≥ 1.5). The t-score was defined as the two-sided t-statistic comparing the average of the *Arid1a*-induced genes with the average of the *Arid1a*-repressed genes (For computing gene signature scores based on expression profile data from human dataset, genes were first z-normalized to the SD from the median across the dataset). t-score below lower quartile or above upper quartile are defined as low or high ARID1A signature respectively and the others are middle. Statistics by log-rank test shown is the difference between low and high group.

3) *The combination of immune checkpoint inhibition and a drug against NF-κB is shown effective in prostate cancer bearing mice in figure 7f/g. However, NF-KB signaling is involved in a plethora of different pathways, therefore the shown effect could be explained by other mechanism rather than the one described in the manuscript. The authors should provide evidence that the levels of CXCL2 are changed in prostates upon the treatment.*

Response: We fully agree with the reviewer that NF-κB signal exerts diverse functions. As suggested, several experiments were conducted to validate our findings. We showed that treatment with JSH-23 to diminish NF-κB signaling compromised *Arid1a*-deleted tumor growth to an extent similar as WT xenografts (**Fig. 3g**). RNA-seq analysis to compare the expression profile of WT and *Arid1a* deleted tumor with or without NF-κB inhibition indicated

that JSH-23 treatment led to the downregulated expression of more than half of upregulated genes elicited by *Arid1a* loss (504/784; **Fig. 3h**). These genes included *Cxcl2*, *Cxcl3*, *Tnf* and among others. In *Pten*^{PC-/-}; *Arid1a*^{PC-/-} mouse models, we proved that NF-κB inhibition attenuated the inductions of CXCL2 and CXCL3 caused by *Arid1a* loss (**Fig. S7g**), along with the contraction of PMN-MDSCs and expansion of CD8⁺ T cells. Together, these results supported that ARID1A loss shapes the immunosuppressive TME via NF-κB-induced PMN-MDSC chemotaxis.

3g, Tumor volume of Myc-CaP expressing *sgARID1A* and control vector subcutaneously inoculated into FVB mice with or without JSH-23 treatment. **3h**, Epithelial cells in xenografts (**g**) were sorted for RNA-Seq and DEGs between WT and *Arid1a* KO epithelium were shown in heatmap (n = 2). **S7g**, ELISA of CXCL2 and CXCL3 in prostate tumors of 4-month-old *Pten*^{PC-/-}; *Arid1a*^{PC-/-} mice treated as indicated (n = 5).

3g, **S7g**. Data represent the mean ± SEM. Statistical significance was determined by two-way ANOVA followed by multiple comparisons (**3g**, **S7g**). **p* < 0.05. ***p* < 0.01, ns, no significance.

4) As said above, targeting of NF-κB in vivo could potentially impact on several cellular pathways, thus leading to potential off targeting or side effects. Are there any NF-κB inhibiting drugs used in the clinic? These issues should be at least discussed.

Response: We fully agree with the reviewer that NF-κB signaling exerts diverse functions in a context dependent manner. Canonical NF-κB promotes proliferation, angiogenesis as well as the remodelling of TME²⁶. Thus, inactivation of NF-κB signal reduced tumor burden in *Kras* induced mouse lung cancer; similar observations were made in melanoma among other cancers²⁷⁻²⁹. However, opposite results were described in murine liver cancer or squamous cell carcinoma, thus highlighting the cell type-specific functions of this pathway and a possible role for NF-κB as a tumor suppressor in certain settings^{30, 31}. In addition to these cell-intrinsic

functions, tumor cells produce a number of NF- κ B-dependent cytokines and chemokines that affect the recruitment and phenotype of immune cells, and the outcome of cancer³²⁻³⁴. Moreover, NF- κ B controls the mRNA expression and protein stability of PD-L1 in tumor cells, thereby inhibiting cytotoxic CD8+ T cells^{35,36}. Till now, great efforts have been made to inhibit NF- κ B signal as a therapeutic approach, which enhance the effectiveness of chemotherapeutic agents in clinical trial including proteasome inhibitors like bortezomib and carfilzomib. The anticancer effect of bortezomib and carfilzomib are partially attributed to inhibit NF- κ B activity by preventing I κ B degradation³⁷. For instance, Bortezomib is a valuable treatment option in the management of relapsed multiple myeloma that improves survival and delays disease progression, albeit with an increased incidence of some adverse events such as thrombocytopenia and neutropenia^{38, 39}. In the revised manuscript, we have discussed the current progress targeting NF- κ B in the clinical setting. Our preclinical results emphasized the beneficial effects of co-targeting immune checkpoints and NF- κ B in PCa, providing rationales for future clinical trials for patients with advanced PCa,

5) *Figure 7f. Are anti PD1/CTLA4 given simultaneously in these mice? Please justify this treatment regimen, also in light of the known toxicity induced by this combination in patients.*

Response: We apology that we did not describe it clearly in the initial submission. Immune checkpoint blockade using antibodies against CTLA4 and/or PD1/PD-L1 generates durable therapeutic responses in a significant subset of patients across a variety of cancer types⁴⁰. However, advanced prostate cancer showed overwhelming de novo resistance to ICB. In the revised manuscript, we have clarified that animals are simultaneously subjected to intraperitoneal injections of PD1 and CTLA4 antibodies twice a week at a dosage of 200 μ g PD1 and 200 μ g CTLA4 antibodies according to Dr. Lu' s work⁴¹. We have noted known toxicities fall within the spectrum of irAEs (immune related adverse events) already described for CTLA-4 or PD-1 pathway blocking agents. The aim of present studies is to prove in a principle whether inhibition of NF- κ B signaling sensitizes advanced PCa tumors, such as *Arid1a*-deficient tumors, to ICB therapy.

6) Also, please clearly state in the text commenting Figure 7f that ICB is given in combination with a drug targeting NF- κ B. This is not mentioned, except for an unclear referring to “cotargeted treatment” (page 17 line 11).

Response: We have clarified that co-targeted treatment means the combinational regimen of NF- κ B inhibitor, JSH-23 and anti-PD1/CTLA4 antibody treatment in the revised manuscript.

7) In Figures 1b, 7b and 7d, how was the cut off of IHC signal intensity determined to divide the patients on the basis of ARID1A, P65, CD15 or CD8 staining?

Response: We have stated how we quantified in the revised manuscript. For ARID1A and nuclear P65, the quantification was based on a multiplicative index of the average staining intensity (1 to 3) and extent of staining (1 to 3), yielding a 10-point staining index ranging from 1 to 9^{24, 25}. Low expression of ARID1A and P65 were defined by a staining index below 6, whereas staining scores of 6 to 9 were considered high expression. CD8⁺ and CD15⁺ cells were measured based on the number of CD8 and CD15 positive cells versus the total cells number in one field. The median positive numbers were set as the cut offs to divide the patients.

8) Ly6G and Ly6C suffice to distinguish PMN-MDSC and M-MDSC. Why did authors include also Gr1 antibody in the stainings (both CyTOF and conventional flow cytometry)? As Gr1 recognizes both Ly6G and Ly6C do the three antibodies compete for binding to their ligands? It is reported in the Biolegend website (vendor of the antibodies used in this manuscript) that “Clone RB6-8C5 impairs the binding of anti-mouse Ly-6G clone 1A8” (<https://www.biolegend.com/en-gb/products/pe-cyanine7-anti-mouse-ly-6g-antibody-6139>)

Response: Reviewer raised the concern regarding to “clone RB6-8C5 impairs the binding of anti-mouse Ly6G clone 1A8”. We apologize that we did not describe accurately in the initial submission. For CyTOF assay, PMN-MDSCs were characterized as CD45⁺ CD11b⁺ F4/80^{low} Ly6G⁺ Ly6C^{low}, whereas M-MDSCs were CD45⁺ CD11b⁺ F4/80^{low} Ly6G⁻ Ly6C^{high}. We made a mistake to include Gr1 in the initial submission, which were not used in CyTOF experiment. In this revision, we applied anti-Ly6G (PE-Cy7, Bio-Legend; 127617) rather than anti-Gr1

antibody to validate the changes of PMN-MDSCs (**Fig. 2c, 3k, 6h** and **Fig. S2b, h, k, S4k, S7d, f**). All the new figures were incorporated into the revised manuscript.

9) According to authors' results showing a specific increase of PMN-MDSC in tumors lacking ARID1A (Figure 2b), IHC and in vivo experiments with neutralizing antibody should be performed with the 1A8 clone (specifically targeting the PMN-MDSC marker Ly6G).

Response: Following your previous comment, we used anti-Ly6G antibody (1A8 clone) rather than anti-Gr1 antibody to specifically evaluate PMN-MDSCs. Both IHC, FACS and in vivo experiments with neutralizing antibody (anti-Ly6G antibody) indicated that PMN-MDSC enrichment is responsible for the acceleration of prostate tumorigenesis in *Pten*^{PC-/-}; *Arid1a*^{PC-/-} mice (**Fig. 2c, d, g, 3k, 6h, 7g** and **Fig. S2b, g-k, S4k, S7d, f**).

10) Figure 2c, 3J, 6h, and extended data figure 2a, 2e, 2h, 4g and 6c: are these MDSC PMN- or M- MDSC? Please also show the manual gating strategies used to identify all the different sub-populations in flow cytometry experiments.

Response: Now we have replaced the figures to show PMN-MDSCs (CD45⁺ CD11b⁺ F4/80^{low} Ly6G⁺ Ly6C^{low}) rather than MDSCs (CD45⁺ CD11b⁺ F4/80^{low} Gr1⁺) (**Fig. 2c, 3k, 6h** and **Fig. S2h, k, S4k, S7d, f**). Per your comment, we presented the representative gating strategies for detecting the immune cells, including PMN-MDSCs, macrophage, CD8⁺ or CD4⁺ T cells, Treg, Ki67⁺ CD8⁺ T cells, IFN γ ⁺ CD8⁺ T cells and among others (**Fig. S2b**).

11) Figure 1i: "tumor cells with a luminal origin (AR+ cells, arrow) were found to invade the stromal compartment". As also stroma cells can express AR, to be sure that they are infiltrating tumor cells a counterstaining with CK8 should be performed.

Response: As suggested, we incorporated CK8 staining to demonstrate that luminal origin tumor cells indeed invade to stromal compartment (**Fig. 1h**).

12) Page 8 line 16. "led to profound reduction in total and cytotoxic T cells". IFN γ is not a

marker for cytotoxicity in CD8 T cells. Also, in the histogram of panel 2b CD8+IFN γ + cells seem negligible in both groups of mice.

Response: We agree with reviewer and rephrased it to “CD8⁺ T cells expressing IFN γ ” or “IFN γ ⁺ CD8⁺ T cells”. We added the quantitation number for each cell type to clearly indicate their differences in Fig. 2b. The histogram of IFN γ ⁺ CD8⁺ cells (**Fig. 2b**) seem negligible because it calculated as the proportion of IFN γ ⁺ CD8⁺ cells within CD45⁺ cells. The ratio of IFN γ ⁺ CD8⁺ cells in CD8⁺ cells rose to about 15% (**Fig. 2c**) in *Pten*^{PC-/-} tumors.

13) Figure 2e. Results in *Rag*^{-/-} mice are paradoxically as they suggest that in absence of T cells, tumor or innate immune cell mediated mechanisms account for a negative role of ARID1A loss on tumor growth, which is the opposite of what shown in full competent mice or patients.

Response: ARID1A is described as a key tumor suppressor in a broad array of cancers, such as HCC and ovarian carcinoma⁴²⁻⁴⁴. Nevertheless, it also shows that ARID1A has context-dependent oncogenic role⁴³. A study by Sun et al. demonstrates that ARID1A exerts tumor-promoting functions during the early phases of liver transformation⁴³. In contrast, mice with liver-specific *Arid1a* loss in established tumors accelerated progression and metastasis⁴³. Mechanistically, *Arid1a* promotes tumor initiation by increasing CYP450-mediated oxidative stress, while *Arid1a* loss after tumor initiation reduced transcription of genes associated with metastasis⁴³. Additional studies also imply that *Arid1a* seems to be oncogenic under specific conditions *in vivo*. For example, under the condition of inactivation of *Apc* and *Pten*, *Arid1a* loss impairs the formation of ovarian cancer for promoting striking epithelial differentiation⁴⁵. *Arid1a* is required for *Apc* mutated intestinal tumorigenesis, which is in contrast to its tumor suppressive function in AOM-DSS induced colorectal cancer models⁴⁶. In contrast to the observations that *Arid1a* loss promoted PCa progression in immune competent mice, we observed that *Arid1a* KO exhibited inhibitory effects when tumor cells were engrafted in *Rag1* null mice (**Fig. 2e**), suggesting that tumor or innate immune cell mediated mechanisms might account for a negative role of ARID1A loss on tumor growth. *Arid1a* loss causing the defects in oncogenic functions or accumulation of ROS in some degree might lead to a negative outcome of *Arid1a* loss on tumor growth. We have noted this issue in the revised manuscript.

14) Figure 3f: evaluation of *Cxcl2* and *Cxcl3* levels should be performed also on murine (Myc-CaP) and not only human cell lines. Extended figure 3e: *Cxcl9* and *Cxcl10* should be tested also after TNF α stimulation, because IFN does not alter ARID1A levels in tumor cells.

Response: We showed that *Arid1a* loss upregulated *Cxcl2* and *Cxcl3* levels in Myc-CaP cells (Fig. S3f). In addition, neither *Arid1a* loss or TNF α treatment decreased *Cxcl9* and *Cxcl10* levels in Myc-CaP cells (Fig. S3f).

15) Extended data figure 5f is supposed to show localization of IKK β and ARID1A by immunofluorescence, but only the green IKK β staining is visible. Please reconcile. Add separate channels if necessary to better appreciate the staining.

Response: Per reviewer comment, we performed double staining of ARID1A and IKK β . TNF α treatment enhanced nuclear subfractionation of IKK β , which were tightly co-localized with ARID1A (Fig. S5j).

S5j

S5j, IKK β and ARID1A immunostaining as indicated in Myc-CaP cells. Scale bar, 100 μ m. Experiments were repeated three times independently with similar results; data from one representative experiment are shown.

16) I would suggest removing the word “metastasis” from the title and similarly toning down conclusions on metastases in the manuscript. Indeed, apart from results shown figure 1J, all the reported experiments in mice and patient-derived specimens analyze primary tumors.

Response: We fully agree reviewer comments. Thus, we tone down the conclusion and changed

it to “prostate cancer progression” or “prostate tumorigenesis”.

17) MDSC usually are very rare in WT mice, especially in the BM. Also, the authors do not show purification results, nor a proof that those cells are true MDSC (expression of suppressive genes or suppressive assays). Indeed, as they are from WT mice they would be more likely been neutrophils or monocytes. The best would have been to purify MDSC from tumor bearing mice.

Response: We appreciate for this important question. Thus, we replaced the results of transwell migration assay by using CD45⁺ CD11b⁺ Ly6G⁺ Ly6C^{low} to isolated PMN-MDSCs from *Pten*^{PC-/-}; *Arid1a*^{PC-/-} tumors. The similar results obtained showing that enhanced CXCR2 ligand expression by NF-κB signaling provided chemotaxis to attract CXCR2-expressing MDSCs to *Arid1a*-deleted tumors (**Fig. 3i**). In addition, we also performed expression and functional assay to show they are indeed PMN-MDSCs. Firstly, Ly6G⁺ cells in tumors exhibited high level of immunosuppressive genes such as *Nos2*, *Arg1*, *Nox2*, *S100a9* manifested the features of MDSC, whereas the Ly6G⁺ cells in peripheral blood of WT mice expressed neutrophil related genes including *Tnf*, *Il6*, *Cxcl4* and among others (**Fig. S2c**). Secondly, standard T cell proliferation co-culture assay showed that these CD11b⁺; Ly6G⁺ cells in tumors strongly suppressed CD3 and CD28 antibody-induced T cell proliferation and activation (**Fig. S2d, e**), establishing that they are functional PMN-MDSCs. Thanks again for your constructive comment to improve our manuscript.

3i, Migration of PMN-MDSCs recruited by conditional mediums (CMs) with the indicated treatments (n = 3). **S2c**, Gene expression were quantified by real-time PCR in Ly6G⁺ cells from 3-month-old *Pten*^{PC /} ; *Arid1a*^{PC /} mice prostate tumors and WT mouse peripheral blood (n = 5). **S2d**, Summary of T cell proliferation assessed by CFSE flow cytometry after 4 days. High and low proliferation were defined as T cell division ≥ 2 and ≤ 1 , respectively (n = 5). **S2e**, IFN- γ secretion by CD8⁺ T cells in the assay in **S2d**, measured by ELISA (n = 5). **3i, S2c, S2e**, Data represent the mean \pm SEM. Statistical significance was determined by two-tailed unpaired *t* test (**3i, S2c, S2e**) and Fisher's exact test (**S2d**). * *p* < 0.05. ** *p* < 0.01, ns, no significance.

Minor points:

18) *It is not clear how many times in vivo experiments were repeated.*

Response: For genetically engineered mouse models (GEMMs) analysis, the examinations were performed dependent on animal available. Thus, all the experiments were independently repeated at least three times with the similar time course and treatment; data presented in figures included Fig 1f, g, h, j, Fig. S1d, e, g, h, Fig. 2d, g, Fig. S2n, p, Fig. 7f, g (at least 5 biological repeats). The xenograft assays were results of one-time experiment with sufficient animal number indicated in figure legend. We have clarified it in the method section and figure legend of the revised manuscript.

19) *Please reconcile the number of mice used in the experiment show in Figure 1c and 1d.*

Response: Now, Fig. S1f (Fig. 1d in last version) included ten animals, which is consistent with Fig. 1c.

20) *Panel 1g is not cited in the text.*

Response: We have replaced this figure with Figure 1e and cited it in the revised manuscript

21) *Figure 2b the histogram is not clear. Please clearly indicate the two types of mice displayed in the two sides of the graph.*

Response: The missing genotype was added in Figure 2b.

22) *Figure 2b-2c and extended figure 2a. It is not clear how many mice/group were used for CyTOF or for conventional flow cytometry. Does panel 2c show CyTOF or flow cytometry data?*

Response: Sorry for the incomplete descriptions. Figure 2b is CyTOF analysis, whereas Fig. 2c is FACS examination. In the revised manuscript, the number for each assay was indicated.

23) *Page 16 line 19. In commenting figure 7c, AIRDIA is reported to be negatively correlated with NF-kB signatures. Please correct.*

Response: The mistake has been corrected.

24) *Figure 7e. Which cells were used for the in vivo experiment?*

Response: We have indicated it is Myc-CaP cells.

25) *Insufficient methods are reported for infections and transfections.*

Response: As suggested, we have revised the methods to describe how we performed infections and transfections in detail.

REFERENCES

1. Rebello, R.J. et al. Prostate cancer. *Nat. Rev. Dis. Primers* **7**, 9-10 (2021).
2. Robinson, D. et al. Integrative Clinical Genomics of Advanced Prostate Cancer. *Cell* **161**, 1215-1228 (2015).
3. Wang, G., Zhao, D., Spring, D.J. & DePinho, R.A. Genetics and biology of prostate cancer. *Genes Dev.* **32**, 1105-1140 (2018).
4. Ibáñez-Vea, M. et al. Myeloid-derived suppressor cells in the tumor microenvironment: expect the unexpected. *J. Clin. Invest.* **125**, 3356-3364 (2018).
5. Veglia, F., Sanseviero, E. & Gabrilovich, D.I. Myeloid-derived suppressor cells in the era of increasing myeloid cell diversity. *Nat. Rev. Immunol.* **21**, 485-498 (2021).
6. Raskov, H., Orhan, A., Gaggar, S. & Gögenur, I. Neutrophils and polymorphonuclear myeloid-derived suppressor cells: an emerging battleground in cancer therapy. *Oncogenesis* **11**, e22 (2022).
7. Zhou, J., Nefedova, Y., Lei, A. & Gabrilovich, D. Neutrophils and PMN-MDSC: Their biological role and interaction with stromal cells. *Semin. Immunol.* **35**, 19-28 (2018).
8. Andersson, R. & Sandelin, A. Determinants of enhancer and promoter activities of regulatory elements. *Nat. Rev. Genet.* **21**, 71-87 (2020).
9. Allis, C.D. & Jenuwein, T. The molecular hallmarks of epigenetic control. *Nat. Rev. Genet.* **17**, 487-500 (2016).
10. Banerji, J., Olson, L. & Schaffner, W. A lymphocyte-specific cellular enhancer is located downstream of the joining region in immunoglobulin heavy chain genes. *Cell* **33**, 729-740 (1983).
11. Calo, E. & Wysocka, J. Modification of Enhancer Chromatin: What, How, and Why? *Mol. Cell* **49**, 825-837 (2013).
12. Bezzi, M. et al. Diverse genetic-driven immune landscapes dictate tumor progression through distinct mechanisms. *Nat. Med.* **24**, 165-175 (2018).
13. Alver, B.H. et al. The SWI/SNF chromatin remodelling complex is required for maintenance of lineage specific enhancers. *Nat. Commun.* **8**, e14648 (2017).
14. Cyrta, J. et al. Role of specialized composition of SWI/SNF complexes in prostate cancer lineage plasticity. *Nat. Commun.* **11**, e5549 (2020).

15. Wellinger, L.C. et al. BET Inhibition Enhances TNF-Mediated Antitumor Immunity. *Cancer Immunol. Res.* **10**, 87-107 (2022).
16. Zhang, X. et al. Promoter hypermethylation of ARID1A gene is responsible for its low mRNA expression in many invasive breast cancers. *PLoS ONE* **8**, e53931 (2013).
17. Wu, J. et al. Insertional Mutagenesis Identifies a STAT3/Arid1b/ β -catenin Pathway Driving Neurofibroma Initiation. *Cell Rep.* **14**, 1979-1990 (2016).
18. Luo, Q. et al. ARID1A Hypermethylation Disrupts Transcriptional Homeostasis to Promote Squamous Cell Carcinoma Progression. *Cancer Res.* **80**, 406–417 (2020).
19. Guan, B., Gao, M., Wu, C.-H., Wang, T.-L. & Shih, I.-M. Functional analysis of in-frame indel ARID1A mutations reveals new regulatory mechanisms of its tumor suppressor functions. *Neoplasia* **14**, 986–993 (2012).
20. Li, X.S., Trojer, P., Matsumura, T., Treisman, J.E. & Tanese, N. Mammalian SWI/SNF-a subunit BAF250/ARID1 is an E3 ubiquitin ligase that targets histone H2B. *Mol. Cell. Biol.* **30**, 1673-1688 (2010).
21. Jiang, Z.H. et al. DNA damage regulates ARID1A stability via SCF ubiquitin ligase in gastric cancer cells. *Eur. Rev. Med. Pharmacol. Sci.* **19**, 3194-3200 (2015).
22. Jiang, Z.H. et al. DNA damage-induced activation of ATM promotes β -TRCP-mediated ARID1A ubiquitination and destruction in gastric cancer cells. *Cancer Cell Int.* **19**, 1475-1482 (2019).
23. Luo, Q. et al. TRIM32/USP11 Balances ARID1A Stability and the Oncogenic/Tumor-Suppressive Status of Squamous Cell Carcinoma. *Cell Rep.* **30**, 98-111 (2020).
24. Li, N. et al. AKT-mediated stabilization of histone methyltransferase WHSC1 promotes prostate cancer metastasis. *J. Clin. Invest.* **127**, 1284-1302 (2017).
25. Qin, J. et al. COUP-TFII inhibits TGF- β -induced growth barrier to promote prostate tumorigenesis. *Nature* **493**, 236-240 (2013).
26. Taniguchi, K. & Karin, M. NF- κ B, inflammation, immunity and cancer: coming of age. *Nat. Rev. Immunol.* **18**, 309-324 (2018).
27. Bassères, D.S., Ebbs, A., Levantini, E. & Baldwin, A.S. Requirement of the NF- κ B Subunit p65/RelA for K-Ras–Induced Lung Tumorigenesis. *Cancer Res.* **70**, 3537-

- 3546 (2010).
28. Meylan, E. et al. Requirement for NF- κ B signalling in a mouse model of lung adenocarcinoma. *Nature* **462**, 104-107 (2009).
 29. Yang, J. et al. Conditional ablation of Ikkb inhibits melanoma tumor development in mice. *J. Clin. Invest.* **120**, 2563–2574 (2010).
 30. He, G. et al. Hepatocyte IKK β /NF- κ B Inhibits Tumor Promotion and Progression by Preventing Oxidative Stress-Driven STAT3 Activation. *Cancer Cell* **17**, 286-297 (2010).
 31. van Hogerlinden, M., Rozell, B.L., Toftgård, R. & Sundberg, J.P. Characterization of the Progressive Skin Disease and Inflammatory Cell Infiltrate in Mice with Inhibited NF- κ B Signaling. *J. Invest. Dermatol.* **123**, 101-108 (2004).
 32. Wang, David J., Ratnam, Nivedita M., Byrd, John C. & Guttridge, Denis C. NF- κ B Functions in Tumor Initiation by Suppressing the Surveillance of Both Innate and Adaptive Immune Cells. *Cell Rep.* **9**, 90-103 (2014).
 33. Hopewell, E.L. et al. Lung tumor NF- κ B signaling promotes T cell-mediated immune surveillance. *J. Clin. Invest.* **123**, 2509–2522 (2013).
 34. Ji, Z., He, L., Regev, A. & Struhl, K. Inflammatory regulatory network mediated by the joint action of NF- κ B, STAT3, and AP-1 factors is involved in many human cancers. *Proc. Natl. Acad. Sci. U S A* **116**, 9453-9462 (2019).
 35. Lim, S.-O. et al. Deubiquitination and Stabilization of PD-L1 by CSN5. *Cancer Cell* **30**, 925-939 (2016).
 36. Gowrishankar, K. et al. Inducible but not constitutive expression of PD-L1 in human melanoma cells is dependent on activation of NF- κ B. *PLoS ONE* **10**, e0123410 (2015).
 37. Manasanch, E.E. & Orłowski, R.Z. Proteasome inhibitors in cancer therapy. *Nat. Rev. Clin. Oncol.* **14**, 417-433 (2017).
 38. Robak, T. et al. Bortezomib-based therapy for newly diagnosed mantle-cell lymphoma. *N. Engl. J. Med.* **372**, 944-953 (2015).
 39. Richardson, P.G. et al. Bortezomib or High-Dose Dexamethasone for Relapsed Multiple Myeloma. *N. Engl. J. Med.* **352**, 2487-2498 (2005).

40. Sharma, P. & Allison, J.P. The future of immune checkpoint therapy. *Science* **348**, 56-61 (2015).
41. Lu, X. et al. Effective combinatorial immunotherapy for castration-resistant prostate cancer. *Nature* **543**, 728-732 (2017).
42. Wu, J.N. & Roberts, C.W.M. ARID1A mutations in cancer: another epigenetic tumor suppressor? *Cancer Discov.* **3**, 35-43 (2013).
43. Guichard, C. et al. Integrated analysis of somatic mutations and focal copy-number changes identifies key genes and pathways in hepatocellular carcinoma. *Nat. Genet.* **44**, 694-698 (2012).
44. Jones, S. et al. Frequent Mutations of Chromatin Remodeling Gene ARID1A in Ovarian Clear Cell Carcinoma. *Science* **330**, 228-231 (2010).
45. Zhai, Y. et al. Arid1a inactivation in an Apc- and Pten-defective mouse ovarian cancer model enhances epithelial differentiation and prolongs survival. *J Pathol* **238**, 21-30 (2015).
46. Mathur, R. et al. ARID1A loss impairs enhancer-mediated gene regulation and drives colon cancer in mice. *Nat. Genet.* **49**, 296-302 (2017).

REVIEWERS' COMMENTS

Reviewer #1 (Remarks to the Author):

The authors have addressed the issues raised in my initial review. I am happy with this revision and have no additional concerns with the work.

Reviewer #2 (Remarks to the Author):

The authors need to be fully congratulated for the amount of work done to further validate their hypothesis.

I fully recommend the publication of this interesting manuscript.

Reviewer #3 (Remarks to the Author):

The authors have rigorously addressed all of my concerns.

Reviewer #4 (Remarks to the Author):

The authors have addressed all my concerns

Response to the reviewers' comments

REVIEWERS' COMMENTS

Reviewer #1 (Remarks to the Author):

The authors have addressed the issues raised in my initial review. I am happy with this revision and have no additional concerns with the work.

Reviewer #2 (Remarks to the Author):

The authors need to be fully congratulated for the amount of work done to further validate their hypothesis.

I fully recommend the publication of this interesting manuscript.

Reviewer #3 (Remarks to the Author):

The authors have rigorously addressed all of my concerns.

Reviewer #4 (Remarks to the Author):

The authors have addressed all my concerns

Response: All the reviewers' concerns have been adequately addressed. We thank the reviewer for his or her comments to improve and strengthen our manuscript.